

# Simulating sedimentary burial cycles – Part 2: Elemental-based multikinetic apatite fission-track interpretation and modelling techniques illustrated using examples from northern Yukon

Dale R. Issler[1], Kalin T. McDannell[2], Paul B. O'Sullivan[3], and Larry S. Lane[1]

[1]Natural Resources Canada, Geological Survey of Canada, Calgary, AB, T2L 2A7, Canada
[2]Department of Earth Sciences, Dartmouth College, Hanover NH, 03755, United States
[3]GeoSep Services, Moscow, ID, 83843, United States

*Correspondence to*: Dale R. Issler (dale.issler@canada.ca)

**Abstract**

Compositionally dependent apatite fission track (AFT) annealing is a common but underappreciated cause for age dispersion in detrital AFT samples. We present an interpretation and modelling strategy that exploits multikinetic AFT annealing to obtain thermal histories that can provide more detail and better resolution compared to conventional methods. We illustrate our method using a Permian and a Devonian sample from the Yukon, Canada, both with complicated geological histories and long residence times in the AFT partial annealing zone. Effective Cl values (eCl; converted from $r_{mr0}$ values), derived from detailed apatite elemental data, are used to define AFT statistical kinetic populations with significantly different total annealing temperatures (~110–245 °C) and ages that agree closely with the results of age mixture modelling. These AFT populations are well-resolved using eCl values but exhibit significant overlap with respect to the conventional parameters, Cl content or $D_{par}$. Elemental analyses and measured $D_{par}$ for Phanerozoic samples from the Yukon and Northwest Territories confirm that $D_{par}$ has low precision and that Cl content alone cannot account for the compositional and associated kinetic variability observed in natural samples. An inverse multikinetic AFT model, AFTINV, is used to obtain thermal history information by simultaneously modelling multiple kinetic populations as distinct thermochronometers with different temperature sensitivities. A nondirected Monte Carlo scheme generates a set of statistically acceptable solutions at the 0.05 significance level and then these solutions are updated to the 0.5 level using a controlled random search (CRS) learning algorithm. The smoother, closer-fitting CRS solutions allow for a more consistent assessment of the eCl values and thermal history styles that are needed to satisfy the AFT data. The high-quality Devonian sample (39 single grain ages and 202 track lengths) has two kinetic populations that require three cycles of heating and cooling (each subsequent event of lower intensity) to obtain close-fitting solutions. The younger and more westerly Permian sample with three kinetic populations only records the latter two heating events. These results are compatible with known stratigraphic and thermal maturity constraints and the QTQt software produces similar results. Model results for these and other samples suggest that elemental-derived eCl values are accurate within the range, 0–0.25 apfu ($r_{mr0}$ values of 0.73–0.84), which encompasses most of the data



from annealing experiments. Outside of this range, eCl values for more exotic compositions may require adjustment relative to better constrained apatite compositions when trying to fit multiple kinetic populations. Our results for natural and synthetic samples suggest that an element-based multikinetic approach has great potential to increase the temperature range and resolution of thermal histories dramatically relative to conventional AFT thermochronology.

## 1. Introduction

Apatite fission track (AFT) thermochronology is a well-established method for constraining low-temperature (< 150 ºC) thermal histories for a broad range of rock types in a wide variety of geological settings (e.g., Gallagher et al., 1998; Gleadow et al., 2002; Lisker et al., 2009; Malusà and Fitzgerald, 2019; Wagner and Van den Haute, 1992). Fission tracks (FTs) are linear damage zones within apatite crystals that form continuously through time by the spontaneous fission decay
of $^{238}$U. An AFT 'age' is a function of the measured track density, which in turn depends on the uranium concentration and the orientation and length of the observed tracks. AFTs form with an initial length of ~16 μm but undergo temperature-dependent length reduction (thermal annealing), yielding a track length distribution that reflects the style and rate of heating and cooling of the apatite-bearing sample (Gleadow et al., 1983). For typical fluorapatite, FTs show variable partial annealing between ~20 ºC and ~110 ºC (Green et al., 1989; Ketcham et al., 1999) and are totally annealed at higher
temperatures. AFT age and length data can be used to constrain the time-temperature history of a sample over a broad temperature range using a thermal model with appropriate annealing kinetics (e.g., Gallagher, 1995, 2012; Green et al., 1989; Issler et al., 2005; Ketcham, 2005; Ketcham et al., 2000, 2007; Willett, 1997).

In general, application of the AFT method can be straightforward for crystalline rocks with apatite grains of similar
composition and a common formation age. In contrast, sedimentary rocks can pose special challenges for the interpretation and modelling of AFT data because they may contain mixed populations of detrital apatite of variable composition from source areas with significantly different thermal histories. This can lead to discordant AFT grain-age distributions where the variance of the grain ages is greater than expected for the analytical error and therefore the grains cannot be treated as a single age population (Brandon, 2002; Galbraith and Laslett, 1993; Vermeesch, 2019). Overdispersed AFT grain ages are
quite common for sedimentary rocks and may result from mixed provenance (e.g., Carter and Gallagher, 2004; Coutand et al., 2006; Garver et al., 1999) and/or variable composition-dependent annealing of fission tracks (Barbarand et al., 2003; Carlson et al., 1999; Crowley et al., 1991; Green et al., 1986; Ketcham et al., 1999; Ravenhurst et al., 2003). The sedimentary samples used in this study have discordant AFT grain-age distributions that we attribute to differential thermal annealing related to apatite composition.




Chlorine content (e.g., Green, 1995; Green et al., 1985, 1986) and $D_{par}$, the mean length of fission track etch figures on the polished mineral surface parallel to the crystallographic c-axis (Burtner et al., 1994; Donelick, 1993), are two widely used parameters for constraining AFT annealing kinetics (e.g., Barbarand et al., 2003; Donelick et al., 2005; Ketcham et al., 1999, 2007). Unfortunately, these single-parameter methods may only work for simple cases, which may be one of the reasons why there are relatively few published multikinetic AFT studies. The empirical $r_{mr0}$ parameter (Carlson et al., 1999; Ketcham et al., 1999, 2007) is potentially more useful for characterizing multikinetic AFT annealing behaviour because it can account for the effects of variable elemental composition, but it has largely been overlooked in the scientific literature. Individual $r_{mr0}$ values can be calculated for each analysed grain using a multivariate equation with elemental data obtained from electron probe microanalysis (Carlson et al., 1999; Ketcham et al., 2007). The $r_{mr0}$ parameter was calibrated based on the results of AFT annealing experiments for a range of apatite compositions and it is a measure of the relative resistance to annealing for a given apatite compared with the most retentive apatite used in the experiments (Ketcham et al., 1999, 2007). It has been applied successfully to constrain multikinetic AFT thermal histories for a number of areas in northern Canada (Issler, 2011; Issler and Grist, 2008a, 2008b, 2014; Powell et al., 2018, 2020; Schneider and Issler, 2019) where the conventional parameters, $D_{par}$ and Cl content, have failed.

In a previous paper (McDannell and Issler, 2021), we used synthetic data to show in principle that it is possible to recover a thermal history with multiple heating events from a single high-quality multikinetic AFT sample using inverse modelling techniques. The obvious implication, discussed in McDannell and Issler (2021), is that multikinetic AFT interpretation can provide more thermal history information than conventional approaches. The purpose of this paper is to demonstrate how compositionally variable detrital apatite grains within natural samples can be grouped into separate statistical kinetic populations that behave as distinct thermochronometers using elemental data and other supporting information. Our method allows for virtually all the sample AFT data to be used; only grains with obviously poor quality AFT or chemical analyses or, more rarely, single grains with unique chemistry or a different provenance may need to be excluded in some cases. The AFT kinetic populations of this study have a wider range of total annealing temperatures (~110 °C to 245 °C) than typical fluorapatite (~110 °C) and therefore they can resolve details of the thermal history beyond the range of a single fluorapatite population. We use an inverse thermal-history model (AFTINV; Issler, 1996; Issler et al., 2005; Schneider and Issler, 2019) to show that it is possible to extract records of multiple heating events preserved in natural multikinetic AFT samples with long residence times in the partial annealing zone. We conclude that age dispersion is a desirable feature of multikinetic AFT samples that can be exploited to resolve thermal histories in unprecedented detail compared with conventional approaches.



## 2. Multikinetic AFT methodology

We use two samples from northern Yukon, Canada, to illustrate how we interpret and model multikinetic AFT data. Table 1 summarizes basic sample location and stratigraphic information. The geological setting and implications of model results for these samples will be discussed in more detail elsewhere as part of a larger regional study of northern Yukon. The early Permian (~295–285 Ma) sandstone sample is from well cuttings collected approximately 70 m below a pre-Cretaceous unconformity in the Eagle Plain west of the Richardson Mountains. Approximately 1 km of Cretaceous strata overlie the unconformity, which represents approximately 170 million years of missing geological record. The second sandstone sample is from an Upper Devonian (~375–365 Ma) outcrop northeast of the first sample on the western flank of the Richardson Mountains. These samples provide an opportunity to investigate whether multikinetic samples can retain important details of the thermal history in areas with a complicated tectonic history and much of the stratigraphic record missing.

**Table 1**. Sample location information

| Sample Name | Curation No. | Lab No. | NTS Map | Location | Latitude (°N) | Longitude (°W) | Unit | Age |
|---|---|---|---|---|---|---|---|---|
| McParlon A-25 (1050-1100 m) | C-604308 | GSS P013-12 | 116-I-04 | Eagle Plain | 66.06947 | 137.31389 | Jungle Creek Fm. | Permian (Sakmarian-Artinskian) |
| 2009LHA003C1 | C-486271 | AtoZ 1148-06 | 116-I-09 | Dempster Hwy quarry | 66.55274 | 136.338984 | Imperial Fm. | Upper Devonian (early Famennian?) |

Figure 1 is a flowchart outlining the key steps for multikinetic AFT analysis and data interpretation. Steps 1 and 2 involve the acquisition of AFT and apatite elemental data. Steps 3 and 4 concern the processing of data to obtain kinetic parameter values that are associated with AFT age and length measurements. Steps 5 and 6 deal with visual displays of the data to aid in the interpretation of kinetic populations. Step 7 involves assessing the interpretation by considering all available data in the context of measurement uncertainty and missing information. The goal here is to try to use all the available data except for obviously poor analyses. Exclusion of any outlier data should be justified on reasonable analytical or geological grounds. Inverse modelling of the interpreted data is carried out in step 8. The objective is to obtain thermal solutions that provide close fits to the data within acceptable statistical precision while satisfying available geological constraints. In general, this requires iterative modelling to refine implicit model parameters, explicit boundary conditions, and investigate different thermal history styles.

### 2.1 AFT and elemental data acquisition

This section discusses the type of data required for multikinetic AFT thermochronology; more details on sample analysis are in Issler et al. (2021). Our AFT data were acquired using the LA-ICP-MS method (Chew and Donelick, 2012; Cogné et al., 2020; Donelick et al., 2005; Hasebe et al., 2004) although the technique works equally well using the older external detector method (EDM; Hurford and Green, 1982). The LA-ICP-MS method has some distinct advantages compared with EDM.



Sample throughput is significantly greater and costs are lower because there is no need to irradiate the sample with thermal neutrons to determine U concentration by proxy — thus avoiding the need for cool down waiting periods and the counting of

induced tracks in mica detectors. Instead, U is measured directly via laser analysis within spontaneous track count areas. The LA-ICP-MS method is well suited for the routine collection of AFT data from a large number of apatite grains because it involves less counting and faster processing than EDM. Typically, 40 single grain AFT ages and 100–200 track lengths are obtained per sample, depending on apatite yield. Generally, this amount of data is sufficient for most multikinetic samples with two or three kinetic populations, but more data may be required for samples with unevenly distributed populations or

with more than three populations.

It has been suggested that the LA-ICP-MS method may be less reproducible in certain circumstances than EDM because it tends to yield AFT data with larger age dispersion and more $\chi^2$ failures (Ketcham et al., 2018; Vermeesch, 2017). This has been more recently attributed to U zonation (i.e., Cogné and Gallagher, 2021) but we do not see evidence for this in our

samples. Unlike many of the examples shown in Cogné and Gallagher (2021) where grain count areas were large and multiple U spots were measured in high, low, and zero Ns regions, our samples typically have U ablation spots located within smaller count areas to mitigate potentially unrecognizable zoning. We think that both the LA-ICP-MS and EDM methods can yield equally good results for the age grains that are measured, which has been repeatedly demonstrated in published results when EDM and ICP-MS methods were compared or ICP-MS AFT dates were referenced to samples with

well-determined absolute ages (e.g., Ansberque et al., 2021; Cogné et al., 2020; Hasebe et al., 2004; Iwano et al., 2019; Seiler et al., 2013; Soares et al., 2014). Arguably, the separate collection of spontaneous count data and U concentration make the LA-ICP-MS method more objective than EDM because there is less opportunity for observer bias to influence results (Donelick et al., 2005). In the former method, spontaneous track areas are counted prior to obtaining U data so there is no way to assess relative age differences among grains during counting. For EDM, there are two issues to consider. First,

for many studies, the number of age grains counted is generally much less than 40 (usually ≤ 20). The greater number of counted age grains naturally increases the statistical probability of $\chi^2$ failure that may complicate mixture model interpretation (McDannell, 2020; Vermeesch, 2019). Second, extra information is available from induced tracks to influence (consciously or unconsciously) how areas are selected for counting (e.g., O'Sullivan, 2018). If only 20 grains are counted and detailed kinetic parameter data are lacking, it makes sense to avoid grains with strongly divergent spontaneous to

induced track ratios. The question becomes, "what is not being counted and how does that factor into sample age dispersion?"







**Figure 1**: Flowchart outlining the key steps involved in the acquisition, interpretation, and modelling of multikinetic AFT data.




Figure 1 illustrates the key steps in our multikinetic workflow. Step 1 summarizes the procedures needed for acquiring AFT and related data using the LA-ICP-MS method. Following standard mineral separation and grain mounting and etching procedures, spontaneous tracks are counted, $D_{par}$ is measured for individual apatite age grains, and grain x–y coordinates are recorded so that subsequent measurements can be linked to the age grains. The sample is analysed using LA-ICP-MS to

obtain U, Th, Sm and U–Pb age data for the AFT age grains, ensuring that the laser spot coincides with the track count area to minimize any potential problems with inhomogeneous U distributions. Normally the sample mount is then irradiated with $^{252}$Cf and re-etched to increase the number of confined tracks for length measurement, but this may not be necessary for samples with high track densities such as Precambrian samples. Track lengths, their angles with respect to the mineral c-axis, and $D_{par}$ are measured and x–y coordinates are recorded for the measured grains. As an option, U, Th, Sm and U–Pb age data

can be acquired for the length grains as well. Although apatite U–Pb ages and $D_{par}$ measurements are low precision, the data can be useful for the qualitative assessment of AFT kinetic populations because it is not always possible to obtain elemental data for every grain. In particular, U–Pb ages can be very useful for distinguishing between detrital and volcanic components in a sample. Routine application of the above procedures means that ages and lengths are measured separately and only a subset of grains may have both age and length measurements. There are significant advantages to obtaining length

measurements from all the age grains, but the analysis is more time-consuming and is not done routinely.

Step 2 (Fig. 1) involves the acquisition of detailed elemental data for sample apatite grains having AFT age and length measurements. These data are critical for constraining the annealing parameters that are required to recognize and group the data into different statistical kinetic populations. We recommend that elemental data be acquired using electron probe

microanalysis (EPMA) rather than by LA-ICP-MS even if the latter method may be more convenient to integrate in the workflow. We have used both methods and they give accurate information for the elements analysed. However, at present, F content cannot always be determined accurately using LA-ICP-MS and therefore OH content cannot be estimated. Without OH information, track retentivity can be underestimated, causing significant overlap of populations in kinetic parameter space, a result consistent with previous studies that emphasize the influence of OH on kinetic parameters (Ketcham et al.,

1999; Powell et al., 2018). In our experience, the following suite of elements are useful for identifying kinetic populations for most multikinetic samples: Fe, Mn, Mg, Na, Sr, La, Y, Ce, F, Cl, Ca, P, Si and S. Fe is very important because current kinetic model calibrations suggest it has a stronger influence on annealing than other elements and it is common in apatite recovered from Phanerozoic rocks of northern Canada. In addition, multikinetic detrital samples can have significant concentrations of elements such as Mn, Mg, Na and Sr. Anions (F, Cl, OH) are known to have a strong effect on annealing

behaviour and both F and Cl are essential for estimating OH (e.g., Barbarand et al., 2003; Carlson et al., 1999). Ca and P are useful for assessing the quality of probe analyses and ensuring stoichiometric calculations are accurate. Finally, Si and S have been observed in high abundance for some samples although their contributions to annealing cannot be properly



accounted for with existing models. These data may still be useful for assessing relative track retentivity among grains, particularly Si since there is experimental evidence to suggest it greatly enhances track retentivity (in wt % quantities) with respect to the Durango apatite age standard (Tello et al., 2006).

EPMA is undertaken in two passes per grain mount using the x–y coordinates to identify grains with age and length measurements (step 2, Fig. 1). As a result, there may be sets of replicate elemental analyses corresponding to grains with both types of measurements. Replicate analyses (both $D_{par}$ and elemental) are very useful for assessing the reproducibility and relative precision of conventional and elemental-based kinetic parameters (see below). Elemental weight % oxide values are converted to atoms per formula unit (apfu) based on an ideal apatite formula (step 3, Fig. 1). We used in house software (Probecal) that incorporates the stochiometric model of Ketcham (2015) to calculate apfu values (including estimation of OH content) for the Permian sample (Table 1). Elemental data were collected in 2011 for the Devonian sample and provided as apfu values directly from the laboratory. The apfu values are needed for calculating the empirical annealing kinetic parameter, $r_{mr0}$ (Carlson et al., 1999; Ketcham et al., 1999, 2007).

## 2.2 Kinetic parameter determination and precision

### 2.2.1 Calculation of $r_{mr0}$, eCl and $eD_{par}$ values

Composition-dependent AFT annealing was investigated using laboratory experiments (Carlson et al., 1999) and the observed difference in annealing behaviour between different compositional groups was approximated using the equation (Ketcham et al., 1999):

$$r_{lr} = \left( \frac{r_{mr} - r_{mr0}}{1 - r_{mr0}} \right)^{k} \qquad (1)$$

where $r_{lr}$ and $r_{mr}$ are the reduced fission-track lengths corresponding to apatite that is less resistant and more resistant to thermal annealing, respectively, and $k$ and $r_{mr0}$ are fitted parameters. The most resistant apatite in the experimental data set is B2 apatite from Bamble, Norway (highly enriched in Cl and OH) and it is the reference apatite used in equation (1). The parameter, $r_{mr0}$, is the reduced length of the more resistant apatite at the point in the thermal history when the less resistant apatite is totally annealed. From a computational standpoint, this means that annealing calculations are only required for the most resistant reference apatite and the degree of thermal annealing can be determined for any number of less resistant apatite populations using equation (1). Elemental apfu values can be used with an empirical multivariate equation (equation (6) in Carlson et al., 1999) to calculate $r_{mr0}$ values for each analysed apatite grain per sample (step 4, Fig. 1). Ketcham et al. (2007) proposed an alternate equation for $r_{mr0}$ that was derived from the analysis of the combined annealing experimental data of Carlson et al. (1999) and Barbarand et al. (2003). Either equation can be used but we prefer the original Carlson et al.





(1999) equation because it resolves kinetic populations better (less grain overlap) for the samples we have studied. Small values of $r_{mr0}$ represent highly track-retentive apatite ($r_{mr0} = 0$ for B2 apatite) whereas retentivity decreases with increasing $r_{mr0}$. Although endmember fluorapatite has a nominal $r_{mr0}$ value of 0.84 for the Ketcham et al. (1999) annealing model, we

have encountered less retentive apatite with higher $r_{mr0}$ values and the model can be extrapolated beyond this limit. Conversely, the model will not work properly in the unlikely event that an apatite is more retentive than the reference B2 apatite.

For most AFT studies, detailed elemental data are unavailable and $D_{par}$ or Cl content are used as kinetic parameters that are

converted to $r_{mr0}$ values in thermal-history models for the annealing calculations. Here, we do the opposite and convert element-based $r_{mr0}$ values for each apatite grain into "effective Cl" (eCl) values (apfu) (e.g., Issler et al., 2018; McDannell et al., 2019; Schneider and Issler, 2019; McDannell and Issler, 2021) (step 4, fig. 1) using an equation that relates $r_{mr0}$ to measured Cl (given in figure 7 of Ketcham et al., 1999):

$$r_{mr0} = 1 - \exp\left[2.107(1 - Cl^*) - 1.834\right] \tag{2}$$

where $Cl^* = \text{Abs}(Cl–1)$. Similarly, we can convert $r_{mr0}$ values to "effective $D_{par}$" (eD$_{par}$) values using the Ketcham et al. (1999) expression that relates $r_{mr0}$ to $D_{par}$:

$$r_{mr0} = 1 - \exp\left[0.647(Dpar - 1.75) - 1.834\right] \tag{3}$$


Equations (2) and (3) can be used to transform measured kinetic parameters (i.e., $D_{par}$ and Cl) to $r_{mr0}$ values or vice versa by rearranging the equations in terms of Cl and $D_{par}$. The Ketcham et al. (2007) multikinetic model has similar equations with slightly different coefficients. For example, the $r_{mr0}$ value for fluorapatite in this model is 0.83, which translates to an eCl value of ~0.03 apfu and an eD$_{par}$ of ~1.85 µm. The corresponding $r_{mr0}$, eCl and eD$_{par}$ values for the Ketcham et al. (1999)

model are 0.84, 0.0 apfu and 1.75 µm, respectively.

There are several advantages to converting nonlinear $r_{mr0}$ values to "linear" eCl values for data interpretation purposes. First, eCl values are more evenly distributed on linear x–y plots which enhances the visual display and interpretation of the data. Second, arithmetic averages of eCl values can be used to estimate representative eCl values for different kinetic populations.

Third, eCl values can be compared with the conventional parameter, Cl content, to show how other elements enhance AFT retentivity beyond what is expected from Cl alone. eCl represents the Cl concentration that would be required to yield an



equivalent $r_{mr0}$ value for the Ketcham et al. (1999) annealing model. The data transformation is temporary because eCl values revert back to $r_{mr0}$ values when used for thermal modelling.

### 2.2.2 Relative precision of $r_{mr0}$ versus conventional kinetic parameters

In order to investigate the relative precision of elemental-based $r_{mr0}$ values with respect to $D_{par}$ and Cl content, elemental data were compiled for fifty-two Phanerozoic AFT samples collected from northern Yukon and the Northwest Territories of northern Canada and then calculated $r_{mr0}$ values were converted to eCl and $eD_{par}$ values (step 4, Fig. 1) using equations (2) and (3). Replicate elemental and $D_{par}$ analyses from separate measurements on grains with both age and length data (step 2, Fig. 1) are very important for assessing the reproducibility of kinetic parameter values (Fig. 2). The eCl and $eD_{par}$ values

represent the actual Cl and $D_{par}$ measurements that would be required to produce the same $r_{mr0}$ value as derived from elemental data. Significant differences between these calculated and measured values imply that there are incompatibilities between $r_{mr0}$ and these conventional kinetic parameters. The data used for figure 2 are in the Supplement; these data will be published in more detail elsewhere after interpretation and modelling have been completed for these samples.

Figure 2a and 2b show a comparison of replicate eCl and Cl values, respectively, from single spot elemental analyses on apatite grains from fifty different samples. Both eCl and Cl are reproducible within ± 0.03 apfu for the majority of the data, representing a 5–10 % variation over the range of measured values. As expected, a somewhat higher percentage (91 % versus 84 %) of Cl values show closer agreement than the eCl values which were determined using multiple elements. Approximately 95 % of the eCl values are within ± 0.06 apfu and a small number of grains show differences as high as 0.24

apfu, likely representing chemical zoning within a sample. These results are encouraging because they suggest that chemical zoning is not a widespread problem and that single-spot probe analyses may be adequate to generate consistent eCl values for most apatite grains. In contrast, replicate $D_{par}$ analyses show much larger variation that represents uncertainties of up to ± 20–50 % of the typical measured range in a sample (Fig. 2c). Such large variations in $D_{par}$ could make it difficult to resolve different kinetic populations if they are present. Despite the relatively high precision of eCl and Cl measurements, there are

strong systematic differences between them (Fig. 2d). The eCl values are skewed to the right of the 1:1 line, indicating higher track retentivity than predicted by Cl alone due to the contributions of OH and elevated cation concentrations. Apatite with low Cl values can still have high eCl values as a result. $D_{par}$ is a function of apatite solubility, which in turn is related to mineral composition. The influence of apatite composition on $D_{par}$ may be the reason why data points are more evenly distributed on the $D_{par}$ versus $eD_{par}$ plot (fig. 2e). Large differences between $eD_{par}$ and $D_{par}$ seem to be explained mainly by

the imprecision of the $D_{par}$ measurements (fig. 2c).





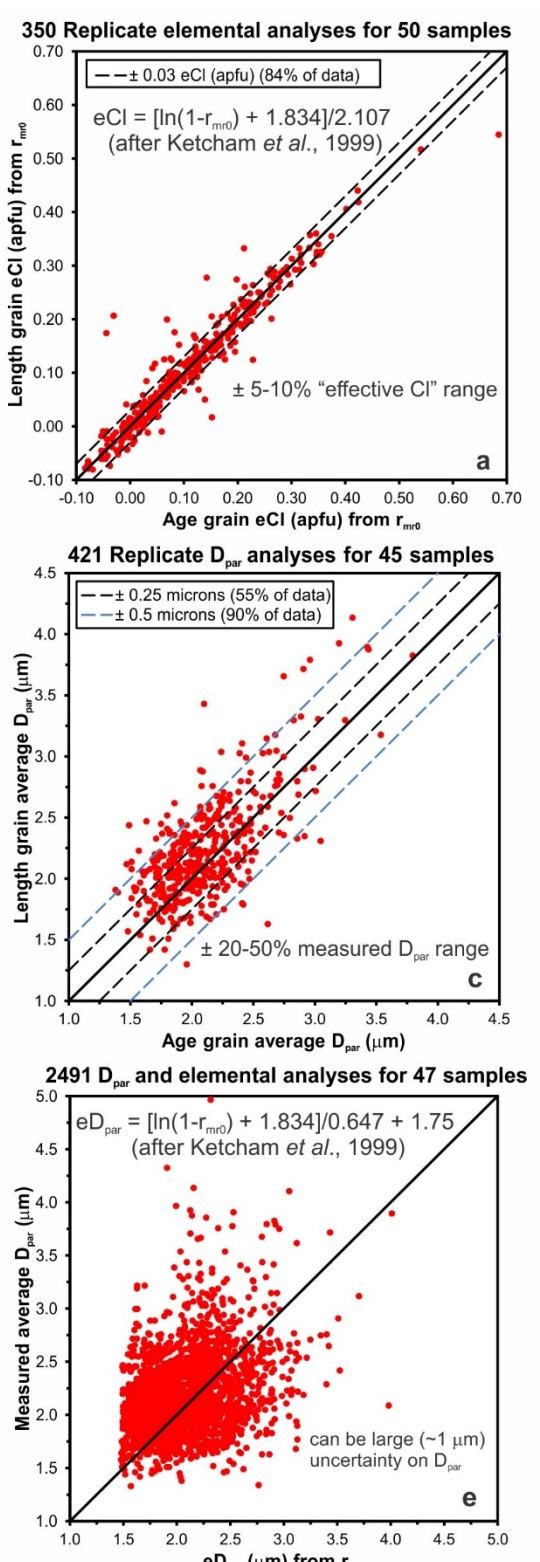

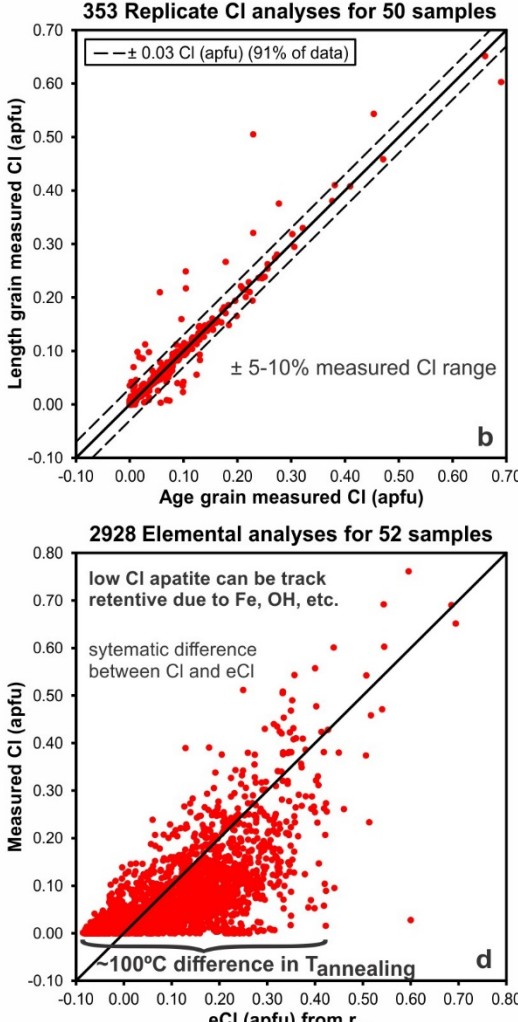

**Figure 2:** Relative precision of different AFT kinetic parameters based on data from Phanerozoic rocks of northern Canada. (a) eCl values from replicate elemental analyses for apatite grains with both age and length measurements. (b) Replicate Cl analyses for apatite grains with both age and length measurements. (c) Replicate $D_{par}$ measurements for apatite grains with both age and length measurements. (d) Comparison of Cl values with elemental-derived eCl values for the same apatite grains. Note systematic differences with eCl values tend to be greater than Cl values because multiple elements increase track retentivity. (e) Comparison of measured $D_{par}$ values with elemental-derived $eD_{par}$ values for the same apatite grains. Unlike in panel (d), data are more evenly distributed around the 1:1 line suggesting that low measurement precision may be a dominant influence.





In summary, the elemental-based $r_{mr0}$ parameter has significantly higher precision than $D_{par}$ and, unlike the single parameter Cl, it incorporates experimentally determined contributions of multiple elements to AFT annealing behaviour. Analysis of
Phanerozoic rocks in northern Canada indicates that heterogeneous apatite composition is widespread in sedimentary rocks. Therefore, $r_{mr0}$ is superior to $D_{par}$ or Cl for recognizing and characterizing multikinetic AFT populations in samples with variable apatite composition.

### 2.3 Kinetic population interpretation

Preliminary interpretation of the data gathered in steps 1 to 4 (Fig. 1) involves visual display of single grain AFT ages on a
radial plot (Galbraith, 1990) and age mixture modelling (Galbraith and Green, 1990; Sambridge and Compston, 1994) to investigate if the sample may have more than one age population (step 5; Fig. 1). We use the DensityPlotter software (version 8.4; Vermeesch, 2012) to plot single grain AFT ages and estimate age populations. Single grain ages and track length measurements are then plotted with respect to eCl values to determine whether discrete age and associated length populations can be identified as statistical kinetic populations (step 6; Fig. 1). Although boundaries separating interpreted
kinetic populations are set at a fixed eCl value, overlap of some population age and length data across boundaries is expected, given the documented uncertainties in eCl values (Fig. 2a). Other complications may include missing elemental data for some grains, compositional zoning, outlier ages associated with a different provenance, and poor-quality data. Some ambiguity in the interpretation can be reduced by including other data such as apatite U-Pb age data, $D_{par}$ values, replicate elemental analyses, and quality control procedures to refine the interpretations (step 7, Fig. 1). Kinetic population
interpretation is facilitated if all length measurements come from age grains; if not, length data must be sorted using eCl values and information from replicate measurements. Single grain ages can be coloured-coded with respect to the kinetic populations on a radial plot for comparison with the radial arms representing model age populations. There is strong evidence to infer multikinetic behaviour if age populations based on eCl values correspond closely with those determined from age mixture modelling, especially if there is other evidence (e.g. organic maturity) to indicate that burial temperatures
were hot enough for substantial AFT annealing.

Figure 3 shows radial plot results for the Permian (Jungle Creek Fm.) and Upper Devonian (Imperial Fm.) AFT samples listed in Table 1. Age mixture modelling yields three and two age populations for the Permian and Devonian samples, respectively. Single grain ages are colour-coded according to the interpreted kinetic populations in figures 4 and 5 and the
model peak ages (Fig. 3) are within one standard deviation of the kinetic population ages summarized in Table 2 (see Issler et al. (2021) for complete AFT data). All kinetic populations pass the $\chi^2$ test but the complete samples fail with high age dispersion (Fig. 3). Pooled ages are used for the kinetic populations if age dispersion is < 10 %; otherwise, the central age is used. Age and length data are well resolved for each kinetic population when plotted with respect to eCl (Fig. 4a, b and Fig.





5a, b) but show significant to complete overlap when plotted with respect to Cl content (Fig. 4c, d and Fig. 5c, d) and $D_{par}$

(Fig. 4e, f and Fig. 5e, f). Population boundaries were set at eCl values of 0.134 and 0.312 apfu for the Permian sample (Fig. 4a, b), and 0.725 apfu for the Devonian sample (Fig. 5 a, b). Percent vitrinite reflectance (%Ro; Table 2) measurements indicate that both samples exceed the threshold for full organic maturity (0.6 %Ro), implying significant AFT annealing.

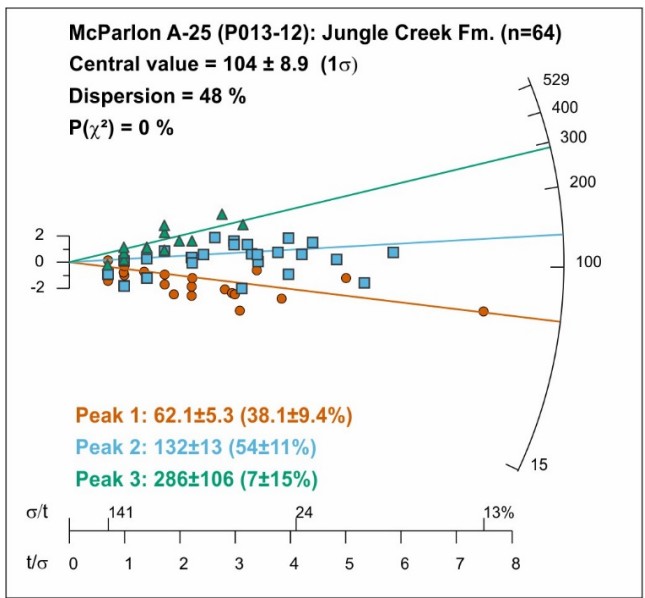

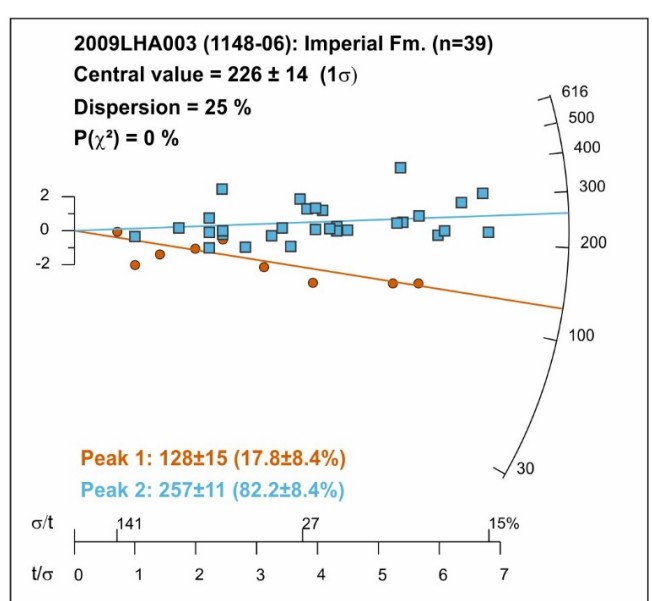

**Figure 3:** Radial plots of single grain ages for the multikinetic Permian (left) and Devonian (right) AFT samples. Points are colour-coded according to the kinetic populations determined in figure 4 and 5. Peak ages are from age mixture modelling and show good correspondence with kinetic population ages summarized in Table 2.


Average $D_{par}$ and Cl values are listed for each kinetic population in Table 2. Although average $D_{par}$ increases with population age, these values are not viable kinetic parameters due to the severe population overlap in $D_{par}$ space. Table 3 summarizes the range of measured elemental concentrations for the apatite grains with age and length measurements (see Issler et al. (2021) for elemental data) and it is clear that there are elevated cation and OH concentrations that can increase track retentivity. In

all cases, the average Cl value for each population is less than the average eCl values calculated using the Ketcham et al. (1999) or Ketcham et al. (2007) $r_{mr0}$-Cl equations (equivalent $r_{mr0}$ values are shown also). The last two columns in Table 2 show the eCl values (and equivalent $r_{mr0}$ values) that were used for thermal history modelling (see below).




**Table 2.** Summary apatite fission track data for Permian and Devonian samples, northern Yukon

| AFT kinetic pop. | n | $\Sigma N_s$ | $\Sigma P_i\Omega_i$ | $\Sigma\sigma_{Pi}^2\Omega_i^2$ | Pooled AFT age ± σ (Ma) | P($\chi^2$) (%) | Central age ± σ (Ma) | Age disp. (%) | MTL (nonproj.) ± σ (n) (μm) | MTL (proj.) ± σ (n) (μm) | Observed Ave. $D_{par}$ (μm) | Ave. Cl (apfu) | 1999 Ave. eCl (apfu) | $r_{mr0}$ | 2007 Ave. eCl (apfu) | $r_{mr0}$ | Model eCl (apfu) | $r_{mr0}$ |
|---|---|---|---|---|---|---|---|---|---|---|---|---|---|---|---|---|---|---|
| McParlon A-25 (1050-1100 m)  (Permian Jungle Creek Fm)   %Ro = 0.63±0.13 (±20%; n=30) |||||||||||||||||||
| 1 | 22 | 200 | 4.803E-05 | 1.663E-11 | 51.3±5.7 | 64 | 57.1±4.7 | 9 | 12.51±2.07 (19) | 14.04±1.20 | 1.90 | 0.02 | 0.05 | 0.822 | 0.05 | 0.819 | 0.05 | 0.822 |
| 2 | 26 | 285 | 3.197E-05 | 6.330E-14 | 109.2±6.8 | 22 | 117.3±7.8 | 11 | 12.04±2.26 (61) | 13.77±1.32 | 2.28 | 0.09 | 0.24 | 0.735 | 0.27 | 0.762 | 0.24 | 0.735 |
| 3 | 16 | 47 | 2.264E-06 | 6.164E-16 | 251.6±37.1 | 99 | 272.5±40.0 | 0 | 11.46±2.74 (14) | 13.62±1.37 | 2.45 | 0.21 | 0.37 | 0.652 | 0.42 | 0.713 | 0.55 | 0.491 |
| 2009LHA003C1 (U. Devonian Imperial Fm)   %Ro = 1.62±0.24 (±15%; n=21) |||||||||||||||||||
| 1 | 9 | 102 | 1.047E-05 | 4.404E-14 | 119.3±12.2 | 86 | 122.8±12.4 | 0 | 12.40±2.23 (63) | 13.97±1.35 | 1.94 | 0.02 | 0.03 | 0.830 | 0.05 | 0.819 | 0.03 | 0.830 |
| 2 | 30 | 572 | 2.746E-05 | 2.801E-14 | 252.4±11.6 | 31 | 258.3±11.6 | 7 | 12.84±2.03 (139) | 14.27±1.15 | 2.54 | 0.13 | 0.20 | 0.757 | 0.23 | 0.773 | 0.50 | 0.542 |

n is the number of age grains for each apatite fission track (AFT) statistical kinetic population per sample.

$\Sigma N_s$ is the sum of the number of counted spontaneous fission tracks for the age grains in each kinetic population per sample.

$\Sigma P_i\Omega_i$ is the sum of the product of individual $^{238}U/^{43}Ca$ ratios ($P_i$) and corresponding track count areas ($\Omega_i$) for each age grain per kinetic population.

$\Sigma N_s$ and $\Sigma P_i\Omega_i$ are used to calculate the pooled AFT age, and with $\Sigma\sigma_{Pi}^2\Omega_i^2$ (includes error on $P_i$), its standard deviation, σ.

Kinetic population grain ages pass the $\chi^2$ test for $\chi^2$ probability, P($\chi^2$) > 5%.

Central age ± standard deviation and corresponding age dispersion are shown alongside the pooled AFT age for each kinetic population per sample.

Conventional and c-axis projected mean track length (MTL) ± standard deviation (number of measured lengths in brackets) is shown for each kinetic population per sample.

The arithmetic average of age and length $D_{par}$ measurements (etch figure diameter parallel to c-axis) is shown for each kinetic population per sample.

The arithmetic average of age and length Cl measurements is shown for each kinetic population per sample.

Arithmetic average eCl and equivalent $r_{mr0}$ values from elemental analysis using the Ketcham et al. (1999, 2007) $r_{mr0}$-Cl equations, and values used for thermal modelling.

AFT ages are calculated using the LA-ICP-MS (ζ-calibration) method with modified ζ = 12.357, standard error (ζ̃) = 0.2251 and total decay constant of $^{238}U$ = 1.55125 x $10^{-10}$ yr$^{-1}$.

**Table 3.** Summary elemental data for AFT age and length measurements (atoms per formula unit)

| Type of analysis | # grains | Na | Mg | S | Mn | Fe | Sr | Y | La | Ce | Nd | Sm | Cl | OH |
|---|---|---|---|---|---|---|---|---|---|---|---|---|---|---|
| | | | | | | P013-12: Permian | | | | | | | | |
| age | 45 | 0 - 0.18 | 0 - 0.08 | 0 - 0.16 | 0.01 - 0.10 | 0.01 - 0.11 | 0 - 0.04 | 0 - 0.06 | 0 - 0.02 | 0 - 0.06 | | | 0 - 0.37 | 0 - 0.96 |
| length | 25 | 0 - 0.12 | 0 - 0.07 | 0 - 0.14 | 0.01 - 0.03 | 0.01 - 0.08 | 0 - 0.04 | 0 - 0.01 | 0 - 0.02 | 0 - 0.06 | | | 0.01 - 0.33 | 0.14 - 0.87 |
| | | | | | | 2009LHA003: Upper Devonian | | | | | | | | |
| age | 40 | 0 - 0.08 | 0 - 0.10 | 0 - 0.10 | 0 - 0.06 | 0.01 - 0.07 | 0 - 0.05 | 0 - 0.03 | 0 - 0.03 | 0 - 0.06 | 0 - 0.03 | 0 - 0.01 | 0 - 0.66 | 0 - 0.99 |
| length | 52 | 0 - 0.13 | 0 - 0.10 | 0 - 0.14 | 0 - 0.05 | 0.01 - 0.07 | 0 - 0.18 | 0 - 0.05 | 0 - 0.03 | 0 - 0.06 | 0 - 0.02 | 0 - 0.01 | 0 - 0.65 | 0 - 1.16 |

The Permian well cuttings sample illustrates some of the issues that can arise when dealing with natural multikinetic samples. It had modest apatite recovery and was processed in two aliquots to obtain 64 single grain ages and 94 track lengths

(Fig. 4). Sample drilling contamination does not appear to be a problem; the sandstone interval is overlain by a shale section and shallower Cretaceous AFT samples from the well bear no resemblance to it. Not all of the grains could be probed (approximately 30 % of the age and length data have no associated elemental data) but there is enough information to demonstrate that three populations are present. Data from the unprobed grains were sorted using age, $D_{par}$ and information from replicate analyses that link length data to age grains. Missing probe data is not a serious concern for this sample;

population ages are similar (within a few million years) if age grains without probe data are excluded. Sixteen of the length measurements without elemental data are associated with age grains that were sorted into various populations. Eleven



lengths with intermediate $D_{par}$ values were included with the dominant population two; the mean track length changes by < 0.3 μm if these data are excluded. Four age grains are interpreted to overlap with other populations in figure 4a. A single grain age in each of population two and three crosses the boundary dividing the populations but values are within the expected error range (Fig. 2a). Two relatively high precision ages (one with a single track length; Fig. 4b) plot within population two space but are assigned to population one based on their age. We have more confidence in the age (rather than eCl) because U concentration was determined at the point where the tracks were counted. Although elemental measurements can be made with high precision, they are the last step after the grains have been subject to multiple treatments (e.g., etching, laser ablation, Cf-irradiation and etching) and it may not be possible to find a clean spot for EPMA that is close to where the age was measured. Therefore, if compositional zoning is present, the grain may appear as an $r_{mr0}$ outlier. Other lower precision young ages were kept with population two because they had little effect on the population age even if they increase the age dispersion (Fig. 4a).

The Devonian outcrop sample is of high quality with 39 single grain ages and 202 track lengths that clearly define two robust kinetic populations in eCl-space (Fig. 5a, b); two single-grain ages from population two are inferred to overlap with population one but this is within measurement uncertainty (Fig. 2a). Both populations overlap completely with respect to Cl (Fig. 5c, d) and $D_{par}$ (Fig. 5e, f), indicating that multikinetic behaviour would be undetected or incorrectly interpreted for this sample using conventional kinetic parameters. Well-resolved populations with low age dispersion (Fig. 5a, b), close agreement between radial plot and kinetic population ages (Fig. 3 and Table 2), high thermal maturity, and abundant AFT data make this an excellent and unambiguous example for illustrating compositionally controlled multikinetic AFT annealing. The next step is to obtain thermal history information from the multikinetic samples using inverse modelling techniques (step 8, Fig. 1).



**Figure 4:** Plots of (a) single grain ages and (b) track lengths grouped into different colour-coded kinetic populations using eCl values for the Permian well sample. Boundary between populations is indicated by vertical black lines. Similar plots of population ages and lengths with respect to Cl concentration (c and d) and D$_{par}$ (e and f) using the same colour-coding as in (a) and (b).



**Figure 5:** Plots of (a) single grain ages and (b) track lengths grouped into different coloured-coded kinetic populations using eCl values for the Devonian outcrop sample. Boundary between populations is indicated by the vertical black line. Similar plots of population ages and lengths with respect to Cl concentration (c and d) and D$_{par}$ (e and f) with same colour scheme as (a) and (b).




## 3. Thermal history modelling of multikinetic AFT data

We use a newer version of the inverse thermal model AFTINV v. 6.1 (Issler, 1996; Issler et al., 2005) to illustrate how detailed thermal history information can be extracted from multikinetic AFT data. We refer the reader to McDannell and Issler (2021) for a recent application of the software to a synthetic AFT dataset. AFTINV uses the Ketcham et al. (1999)

multikinetic annealing formulation and can model up to four kinetic populations. Similar to the HeFTy model (Ketcham, 2005), thermal histories are generated randomly and fits between calculated and observed AFT ages and lengths are assessed using *p*-values. Temperatures are calculated at fixed, user-specified time points using randomly selected heating and cooling rates. Different thermal history styles (Tstyles) can be combined to create complicated thermal histories with multiple phases of heating and cooling. For example, Tstyles 4 (random heating, then cooling), 5 (random cooling, then heating) and 10

(cooling then two cycles of heating and cooling with randomly selected thermal minima) are used for modelling the two samples in this study (Table 4). Monte Carlo calculations terminate when a specific number of solutions (usually 300) exceed the 0.05 level of significance. Unlike HeFTy, a controlled random search (CRS; Price, 1977; Willett, 1997) learning algorithm is then used to try and improve the initial 0.05 level solution set to a higher significance level (typically 0.5 level). Calculations proceed iteratively until either all members of the solution set exceed the new significance threshold or no

further improvements can be made.

Table 4 shows the boundary conditions used to model the Permian (P013-12) and Devonian (2009LHA003) samples. The Permian sample was modelled with an initial pre-depositional cooling event followed by two cycles of random heating and cooling (Tstyle 10). The Devonian sample has an extra cycle of heating and cooling that was accommodated seamlessly by

adding an additional random cooling/heating (Tstyle 5) and heating/cooling (Tstyle 4) cycle. Each model starts at a high temperature (245–250 °C) and cools to surface temperatures (0–30 °C) within a time range for deposition that captures uncertainties in the stratigraphic age. The model uses static and dynamic temperature limits. Static limits define the entire model search space whereas dynamic limits are applied only at model inflection points to focus calculations into favourable regions of solution space. The static temperature limits (column 5, Table 4) were estimated based on maximum temperatures

inferred from organic maturity data, the general degree of annealing among the different kinetic populations, and regional maturity and stratigraphic trends. Dynamic limits are given for thermal minimum (column 6, Table 4) and thermal maximum (column 7, Table 4) inflection points and they are set equal to the static temperature limits where geological constraints are lacking. Heating and cooling rate limits are estimated based on regional geological information and model sensitivity tests. Vitrinite reflectance (%Ro) values are calculated for the entire post-depositional thermal history and for the last phase of

heating and cooling using the basin%R$_o$ model (Nielsen et al., 2017). Further details on model boundary conditions and supporting geological data are the subject of a future paper that deals with the regional thermal history and its geological implications.





**Table 4.** Model boundary conditions/constraints for two AFT samples

| Time range (Ma) | Thermal history style | Rate limits (°C/Myr) heat | Rate limits (°C/Myr) cool | Temperature limits (°C) Range | Temperature limits (°C) Tmin | Temperature limits (°C) Tmax | %Ro limit | Geological phase |
|---|---|---|---|---|---|---|---|---|
| P013-12 - Jungle Creek: cool/heat/cool/heat/cool history (Tstyle=10) | | | | | | | | |
| 600-295 | cool only | | 0-5 | 0-250 | | | | pre-deposition |
| 295-285 | random Tmin | 0.1-5 | 0.1-5 | 0-155 | 0-30 | | 0.63±0.13 | deposition |
| 285-115 | 1 rand heat/cool | 0.1-5 | 0.1-5 | 0-155 | | 130-155 | | burial/exhumation |
| 115-102.5 | random Tmin | 0.1-3 | 0.1-3 | 0-130 | 0-40 | | 0.60±0.15 | onset of reburial |
| 102.5-0 | 1 rand heat/cool | | | 0-130 | | 95-130 | | burial/exhumation |
| 102.5-5 | | 0.1-3 | 0.1-3 | | | | | burial/exhumation |
| 5-0 | | 0.1-3 | 0.1-20 | | | | | exhum./cooling |
| 0 | | | | 15-35 | | | | present |
| 2009LHA003 - Imperial Formation: cool/heat/cool/heat/cool + cool/heat + heat/cool history (Tstyle=10,5,4) | | | | | | | | |
| 700-377.5 | cool only | | 0-5 | 0-250 | | | | pre-deposition |
| 377.5-365 | random Tmin | 0.01-15 | 0.01-5 | 0-195 | 0-30 | | 1.62±0.24 | deposition |
| 365-240 | 1 rand heat/cool | 0.01-15 | 0.01-5 | 0-195 | | 155-195 | | burial/exhumation |
| 240-210 | random Tmin | 0.01-5 | 0.01-5 | 0-150 | 0-150 | | | onset of reburial |
| 210-115 | 1 rand heat/cool | 0.01-5 | 0.01-5 | 0-150 | | 0-150 | | burial/exhumation |
| 115-100 | random Tmin | 0.01-3 | 0.01-3 | 0-110 | 0-110 | | 0.50±0.20 | onset of reburial |
| 100-0 | 1 rand heat/cool | | | 0-110 | | 60-110 | | burial/exhumation |
| 100-5 | | 0.01-3 | 0.01-3 | | | | | burial/exhumation |
| 5-0 | | 0.01-3 | 0.01-20 | | | | | exhum./cooling |
| 0 | | | | 0-5 | | | | present |

Model sensitivity runs were undertaken to determine the style of thermal history and the suite of kinetic parameters that were
needed to obtain model solutions that closely fit the AFT data. Table 2 lists the model eCl values that yield a set of 300
solutions at the 0.5 significance level. For the Permian sample, the average eCl values of 0.05 and 0.24 apfu could be used
for kinetic populations one and two, respectively, but the eCl value for population three had to be increased from the average
of 0.37 to 0.55 apfu to obtain solutions that closely fit all three populations. For the Devonian sample, the average eCl value
(0.03 apfu) was used for population one but the model eCl value had to be increased substantially from the average value of
0.20 to 0.50 apfu for population two. A three-cycle heating model was attempted for the Permian sample but it cannot be
constrained by the data because the first thermal peak had a lower temperature than the second peak. Models using only two
cycles of heating (Paleozoic-Mesozoic burial/erosion and Late Cretaceous-Cenozoic burial/erosion) were unsuccessful in
finding solutions for the Devonian sample. Thin remnants (20 – 30 m) of Upper Triassic (Carnian-Norian) strata occur in
isolated places in the northern part of the study region (Norris, 1981) suggesting a burial event of unknown significance. The



three-cycle history was set up to allow for a reburial event starting in the upper Triassic — but it could occur anywhere between 0 °C and 150 °C (Table 4; i.e., no requirement to cool to near-surface temperatures prior to reburial). If a Triassic-Early Cretaceous burial/exhumation event were not required by the data, then only a minor inflection should appear on the thermal history over this time interval.

## 4. Model results

### 4.1 AFTINV

Figure 6 and 7 show AFTINV model results for the Permian and Devonian samples, respectively. The upper panels of both figures show acceptable solution space (≥ 0.05 significance level) defined by the light grey Monte Carlo solutions. The dark grey curves represent the "good" 0.5 level solutions obtained from updating the light grey solutions using the CRS algorithm. These results indicate that the different kinetic populations are mutually compatible and can be modelled with the same thermal history. The blue curve is the exponential mean of the 300 0.5 level solutions and it provides an excellent-fitting, smoothed thermal history. The green curve is the closest fitting minimum objective function CRS solution. The initial temperature search space (yellow shaded area) was estimated using regional geological information and model sensitivity analyses. It is larger than the acceptable solution space but small enough to limit the time spent interrogating unproductive regions of solution space. The initial temperature limits collapse to the bounds enveloping the acceptable thermal solutions when the model converges. The lower panels in Figure 6 and 7 display the model and c-axis projected measured track length distributions (coloured histograms) and model retention age distributions for each kinetic population. Model track length distributions are shown for the exponential mean (blue) and minimum objective function (green) solutions along with the regions encompassing track length distributions for the 0.05 (light grey) and 0.5 (dark grey) level solutions. Model retention ages represent the age of the oldest surviving track in each population (assumed to be ~ 2 μm; Ketcham et al., 2000) and they give an estimate of how much of the thermal history is preserved in each population.

Two cycles of heating and cooling were used to model the post-depositional thermal history of the Permian sample based on preserved stratigraphy in the well (Fig. 6). A broad range of solutions is permitted at the 0.05 significance level due to the generally low number of track lengths for this sample. Most of the lengths are in population two and therefore it has the strongest influence on the thermal history. The CRS algorithm defines a much narrower region of "good" solution space at the 0.5 level, but it allows for different modes. Most of the peaks for the first heating cycle occur between 160 Ma and 180 Ma but two higher temperature peaks at 192 Ma are associated with higher older temperature peaks at 70 Ma in the second heating cycle. The 70 Ma peaks are generated by higher Cretaceous heating rates that are permitted by the sparse length data for population one. Model retention ages were used to estimate total annealing temperatures of 115 °C, 160 °C and 245 °C for population one, two and three, respectively (Fig. 6). Retention ages for the exponential mean solution are plotted on the




thermal history for each kinetic population (coloured stars). Modelling suggests that population three has retained tracks from the initial phase of cooling at very high temperatures after approximately 540 Ma whereas population two records pre-depositional cooling at lower temperatures after 420 Ma. Any record of pre-depositional cooling is lost for population one due to thermal resetting by the first heating event; it records cooling after 140 Ma and therefore provides no constraint on the

magnitude and time of thermal peak one but it is sensitive to the second heating event. Population 2 is most sensitive to thermal peak one and it provides some constraint on the second thermal event. Joint modelling of all kinetic populations improves model resolution. The most robust result is the predicted > 30 °C difference in burial temperatures between the two heating cycles. The highly retentive population three has little influence on the post-depositional thermal history. Effective Cl values had to be adjusted higher (than calculated) for population three in order to obtain good solutions that improved the

model fit to population three AFT data without significantly changing model temperatures.

The Devonian outcrop sample is of much better quality in terms of data abundance and, although it has only two kinetic populations, it constrains a more complicated and well resolved thermal history with three cycles of heating and cooling of decreasing intensity with time (Fig. 7). The first phase of rapid Paleozoic heating is consistent with regional organic maturity and sedimentological evidence for rapid burial. Results show substantial cooling (> 100 °C) prior to reheating starting in the

Triassic. This scenario is permitted but not forced by the model which only requires that an inflection point occur somewhere within the initial temperature envelope over the interval, 240–210 Ma. The third heating/cooling cycle is consistent with the widespread occurrence of Cretaceous strata of generally low organic maturity across the region (Link and Bustin, 1989; Reyes et al., 2013). The three-cycle model works so well that the initial set of Monte Carlo solutions found thirteen 0.5 level

solutions (irregular dark grey curves) and the CRS algorithm had no problem updating the remaining solutions to the 0.5 level. Although a Triassic to Early Cretaceous burial/exhumation event was not expected for the area, newer results from the Permian sample to the west (Fig. 6) indicate that heating was sufficient at this time to strongly anneal AFT parameters and overprint Paleozoic thermal maturity at that location, providing further evidence that high-quality multikinetic data can preserve information on multiple heating events. Estimated total annealing temperatures for population one and two are 110

°C and 230 °C, respectively. The very old retention ages (generally > 500 Ma; Fig. 7) and high annealing temperature for population two suggest that it is most sensitive to the pre-depositional cooling history and the first high-temperature peak. Most of the retention ages for population one are older than, or similar to, the second thermal peak — indicating that it has been strongly annealed and retains no information from the first heating event and is most sensitive to the second thermal event. The larger uncertainty on the timing of the third lower temperature peak is consistent with the lower level of AFT

annealing.



**Figure 6:** Upper panel shows AFTINV thermal history results for the Permian sample. Light grey lines are statistically acceptable Monte Carlo solutions (≥ 0.05 significance level); dark grey lines are good solutions (≥ 0.5 level). The black curves bounding model solutions are not valid solutions. The blue curve is the exponential mean (EM) of the 300 good solutions; the green curve is the closest fitting minimum objective function (MOF) solution. Lower panels show model and observed track length distributions, and the distribution of model retention ages (age of oldest track) for the different kinetic populations. The goodness of fit (GOF) probability for the age and length data is given for the exponential mean solution. $T_{anneal}$ is the estimated total annealing temperature for each population. Uncertainties on average retention age, average peak temperature and average peak time are two standard deviations.



**Figure 7:** Upper panel shows AFTINV thermal history results for the Devonian sample. Light grey lines are statistically acceptable Monte Carlo solutions (≥ 0.05 significance level); dark grey lines are good solutions (≥ 0.5 level). The black curves bounding model solutions are not valid solutions. The blue curve is the exponential mean (EM) of the 300 good solutions; the green curve is the closest fitting minimum objective function (MOF) solution. Lower panels show model and observed track length distributions, and the distribution of model retention ages (age of oldest track) for the different kinetic populations. The goodness of fit (GOF) probability for the age and length data is given for the exponential mean solution. $T_{anneal}$ is the estimated total annealing temperature for each population. Uncertainties on average retention age, average peak temperature and average peak time are two standard deviations.



## 4.2 QTQt

Model results from the Bayesian QTQt software (Gallagher, 2012) are shown for comparison with model output from
AFTINV. General model setup was the same as the approach discussed in McDannell and Issler (2021), except here the
same geologic constraints and %Ro data that were used in the AFTINV models were used in QTQt as well. The differences
between the QTQt representative output models (i.e., maximum likelihood (ML), maximum posterior (MP), maximum mode
(MM), and expected (EX); see Gallagher and Ketcham, 2020) are discussed in McDannell and Issler (2021) and in Figure 8.
We primarily discuss the ML model since it provides the best fit to the observed data. We prevented more complex models
from being accepted during simulations unless they provided a better fit to the data — therefore unnecessary complexity was
prohibited and the ML and MP models often end up being similar, with LHA003 being an exception. The overall boundary
conditions and heating-style assumptions applied to the AFTINV inversions were absent for QTQt modelling; since the latter
relies on the data to directly inform the level of model complexity (i.e., t–T history style). We fixed the eCl kinetic parameter
for the well-determined kinetic population most similar to fluorapatite, which was the best constrained group in the
annealing experiments of Carlson et al. (1999). The remaining kinetic populations were allowed to vary within uncertainty
and underwent resampling during QTQt inversions.

The history for Permian sample P013-12 exhibits a two-pulse heating history in agreement with the AFTINV results.
Maximum temperatures of ~150 °C were achieved at ca. 135 Ma, followed by a second heating event to ~110 °C at ca. 60
Ma (Fig. 8). The QTQt results for the Devonian Imperial Fm. sample are notable, since preliminary AFTINV test
simulations investigated the possibility of a two or three-peak thermal history and the ability of the data to resolve the latter
scenario within the regional geologic context of preserved Mesozoic outliers. Other than the depositional age of the Imperial
Fm., a large constraint box was placed between 55 ± 55 °C to allow for a potential thermal minimum of unknown magnitude
at 107 ± 10 Ma (the approximate depositional age of the overlying Cretaceous rocks). The QTQt results demonstrate that a
three-peak history is the more likely scenario that provides the best fit to the AFT data (Fig. 9). The EX and ML models
suggest the first thermal maximum is at ca. 300 Ma and ~155–190 °C, the second peak is at ca. 155 Ma and ~100–115 °C,
and the third occurs near 60 Ma and 70–80 °C. The AFTINV and QTQt results are very similar, even with the subtle trade-
offs between the different thermal minima/maxima inflection points and preferred model population kinetic parameters.



**Figure 8:** QTQt model results for Permian detrital AFT sample P013-12 (Jungle Creek Formation). (a) Time-temperature plot of the histories retained post burn-in, coloured by relative probability, with warmer colours denoting higher probability. Individual models include the Maximum Likelihood (highest likelihood and best fit; red curve), Max. Posterior (preferred Bayesian model – simplest, balancing fit with complexity; gold curve), Max. Mode (peak of the marginal distribution at 1 m.y. intervals; white curve), Expected (weighted mean of the marginal distribution ± 95% credible interval; black curves). The latter two models are summaries of the posterior distribution and not directly sampled during the inversion. (b) Max. likelihood fit to the track length distribution and predictions for AFT age and MTL for population one using a fixed eCl value of 0.05 apfu. (c) Max. likelihood fit to the track length distribution and predictions for AFT age and MTL for population two, allowing resampling of the calculated eCl value. (d) Max. likelihood fit to the track length distribution and predictions for AFT age and MTL for population three, allowing resampling of the calculated eCl value. General information: Green box is depositional age of 290 ± 5 Ma (15 ± 15 °C). Yellow box is geologic constraint at 108.75 ± 6.25 Ma (20 ± 20 °C). Note: All populations utilize the central AFT age in QTQt, whereas the pooled age is used in AFTINV for samples with < 10% age dispersion. Model setup: 500,000 iterations (burn-in) and 500,000 iterations (post burn-in; shown). Prior model space: 300 ± 300 Ma and 125 ± 125 °C. Modern surface temperature of 25 ± 10 °C and maximum allowed $\partial T/\partial t$ of 20 °C/My. Proposal moves were rejected if proposed outside of the prior and more complex models were rejected for equivalent likelihood. See text for further discussion.



**Figure 9:** QTQt model results for Devonian detrital AFT sample LHA003 (Imperial Formation). (a) Time-temperature plot of the histories retained post burn-in, coloured by relative probability, with warmer colours denoting higher probability. Individual model descriptions are the same as those in Figure 8. (b) Max. likelihood fit to the track length distribution and predictions for AFT age and MTL for population one using a fixed eCl value of 0.03 apfu. (c) Max. likelihood fit to the track length distribution and predictions for AFT age and MTL for population two, allowing resampling of the calculated eCl value. Note that the MP model path differs from the ML model primarily because the large Cretaceous box allows a linear t–T segment between 280 Ma and 100 Ma that is simpler, but provides a poorer fit to the data. General information: Green box is depositional age of 371.25 ± 6.25 Ma (15 ± 15 °C). Yellow box is geologic constraint at 107 ± 10 Ma (55 ± 55 °C). Model setup: 500,000 iterations (burn-in) and 500,000 iterations (post burn-in; shown). Prior model space: 350 ± 350 Ma and 125 ± 125 °C. Modern surface temperature of 2.5 ± 2.5 °C and maximum allowed ∂T/∂t of 20 °C/My. Proposal moves were rejected if proposed outside of the prior and more complex models were rejected for equivalent likelihood.



## 5. Discussion

Our multikinetic data interpretation and modelling techniques are designed to improve thermal history resolution by exploiting compositionally controlled AFT annealing in samples with high age dispersion. Although numerous factors can
contribute to age dispersion, we conclude that multikinetic annealing is the dominant cause of dispersion for Phanerozoic detrital samples from broad geographic areas of northern Canada based on published (Issler et al., 2018; Powell et al., 2018, 2020; Schneider and Issler, 2019) and unpublished (e.g., Fig. 2) results. This result should apply to other areas that have experienced similar amounts of burial and exhumation and therefore it is of global significance. The method is not restricted to sedimentary rocks and has been applied to Precambrian basement and Proterozoic metasedimentary rocks as well
(McDannell et al., 2019; McDannell et al., *in press*). Heterogeneous apatite compositions are common for multikinetic detrital AFT samples and therefore kinetic populations are much better resolved using the multi-elemental $r_{mr0}$ parameter than the conventional kinetic parameters, Cl content or $D_{par}$. Although Cl content can be measured with sufficient accuracy and precision, it ignores the documented effects of cation and OH concentrations on track retentivity (e.g., Barbarand et al., 2003; Ketcham et al., 1999). $D_{par}$ is influenced by apatite composition but replicate $D_{par}$ analyses show that it has low
accuracy and precision (Fig. 2). The dearth of published multi-elemental data for AFT studies suggests that true multikinetic behaviour is underrepresented and underutilized in thermal history analysis.

Well characterized multikinetic samples can contain significantly more information than samples with a single AFT population. Therefore, more data and effort are required to interpret and model multikinetic samples, especially if they come
from areas with complicated tectonic histories. Generally, 40 age and 100–200 track length measurements are sufficient for typical multikinetic samples with two or three populations, depending on how the data are distributed among the populations. Samples with more populations are less common and may require additional processing to obtain sufficient data to better resolve each population. We can use most or all of the AFT measurements because our interpretations are constrained by multiple parameters (elemental, U–Pb age and $D_{par}$ data). In our experience, multikinetic detrital samples are
best interpreted as having discrete kinetic populations that are defined by grouping age and length data using eCl values. The close correspondence between population ages inferred from kinetic parameters and peak ages derived from age mixture modelling supports the discrete model approach and provides compelling evidence that differential AFT annealing is controlling age populations. A discrete population model is the simplest interpretation that is consistent with our data and requires less assumptions than a more continuous model that subdivides the data into finer groups using pre-determined
incremental kinetic parameter values. The latter model ignores the results of age mixture modelling and assumes that all measurements represent accurate kinetic parameter values over the full (kinetic) model range.





An important point is that successful modelling of multikinetic samples relies on the *relative* annealing behaviour that is implicit in the $r_{mr0}$ model(s) of Ketcham et al. (1999, 2007). The basic assumption is that the same annealing mechanism

applies to all apatite, but that composition controls the temperature at which annealing occurs. This reasonable assumption was used successfully by Ketcham et al. (1999, 2007) to account for experimental annealing data for apatite of variable composition. We extend this approach to natural multikinetic AFT samples and we are able to show that a common thermal history can reproduce AFT ages and lengths for the different kinetic populations. Ketcham et al. (1999) advised against general use of the Carlson et al. (1999) elemental-based multivariate equation for predicting $r_{mr0}$ values for unknown apatites

due to limited calibration data. We agree that this equation (or the equivalent equation in Ketcham et al., 2007) cannot predict $r_{mr0}$ values accurately over the full range of apatite compositions that are likely to be encountered in nature, and accurate prediction for poorly represented apatite may well be an unattainable goal. However, this should not be a deterrent to using a method that has the potential to yield better results than conventional approaches, keeping in mind that the conventional parameters are no more accurate and involve the same annealing model. In order to advance the field of

thermochronology, it is essential to pursue the logical consequences of the annealing experiments and show that it is possible to use existing techniques to better understand complicated AFT samples.

The examples in this paper, other published studies (Powell et al., 2018, 2020; Schneider and Issler, 2019) and unpublished results (Fig. 2) suggest that kinetic parameters can be accurately determined for eCl values of approximately 0 – 0.25 apfu

($r_{mr0}$ values of 0.73 – 0.84), a range that encompasses most of the data for the annealing experiments (Ketcham et al., 1999) and represents more typical apatite compositions. In our experience, the Carlson et al. (1999) multivariate equation overestimates track retentivity for endmember fluorapatite with $r_{mr0}$ values > 0.84 (requiring more negative eCl values than predicted), and underestimates retentivity for more exotic higher retentivity apatite with $r_{mr0}$ values < 0.73 (requires higher eCl values than predicted). A successful tactic we have used is to anchor model calculations on the kinetic population having

the best calibrated annealing behaviour within the above kinetic parameter range. In this case, the model eCl value can be fixed at the average eCl value for the anchor population and eCl values for atypical populations can be adjusted as required to obtain successful solutions that fit all populations. Issler et al. (2005; their Fig. 17) used this approach to investigate how the objective function value of the exponential mean solution changes as the kinetic parameter of one population is adjusted with respect to a population with a fixed kinetic value. It is possible to determine the optimal kinetic parameter offset that

yields the lowest minimum object function value (closest fit). Optimization generally occurs in a kinetic parameter interval where changing the kinetics has little effect on the thermal history solution (i.e., minor changes in annealing sensitivity), yet model misfit is minimized. For software like QTQt, manual adjustment is not required because the model resamples kinetic parameters in an attempt to fit the data. In addition to the uncertainty with $r_{mr0}$ calculations, high retentivity populations can span a broad range of kinetic parameter space and it can be difficult to obtain representative average eCl values for



undersampled populations (Fig. 4) or for populations with unevenly distributed data that cluster at one end of the range (Fig. 5). For the Permian sample, average eCl values for kinetic populations one and two are within the range of more typical apatite and require no adjustment whereas the population three eCl value had to be increased to obtain 300 CRS solutions that fit all three populations. For the Devonian sample, the eCl value was fixed for population one and the eCl value for population two was adjusted relative to population one until 300 solutions at the 0.5 significance level were obtained.

Preliminary investigations of resampling the kinetic parameter for *all populations* within QTQt, suggests that the 'fluorapatite' population often remains relatively stationary, whereas 'exotic' endmember kinetic populations are sampled outside of their calculated eCl range. This aligns with results from the annealing experiments and lends support for our AFTINV approach of relative kinetic adjustment during interpretation and modelling.

The allowable range for estimated eCl values for high retentivity populations depends on the degree of annealing. At low levels of annealing, there is lower sensitivity to the thermal history and a broad range of eCl values can yield good solutions. Numerous models were run for the Permian sample, but results are only presented for the model with an eCl value of 0.55 apfu for kinetic population three (Fig. 6) because it gave the highest number of solutions at the 0.5 significance level. Models with eCl values between 0.45 apfu and 0.55 apfu also gave a significant number of 0.5 solutions. The effect of changing eCl

was to improve the fit to AFT parameters while having a negligible effect on the thermal history which is largely constrained by populations one and two. For the Devonian sample, a range of eCl values is permissible as well for the higher retentivity population two but we only show results for an eCl value of 0.5 apfu (Fig. 7) because it gave 300 solutions at the 0.5 level with the broadest temperature envelope. The number of 0.5 level solutions decreased when eCl was increased to 0.55 apfu or decreased to 0.45 apfu. When eCl was increased from 0.5 apfu to 0.55 apfu, the region defined by the 0.5 level solutions

narrowed and the corresponding average peak temperature shifted from 173 °C at 341 Ma to 178 °C at 351 Ma with calculated average vitrinite reflectance increasing from 1.55 %Ro to 1.72 %Ro. The variation was much less (1 °C and 1 Ma) when eCl was decreased to 0.45 apfu. We do not consider these uncertainties in eCl values to be a problem for multikinetic modelling because the effects on the thermal history are minor and well within available geological constraints. In our experience, relative annealing can significantly limit the allowable offset in eCl values (incremental changes < 0.05

apfu) between lower retentivity populations that experienced strong annealing. In some cases, incremental changes must be < 0.05 apfu or solutions cannot be obtained at the 0.5 level. For our Permian sample, good solutions were obtained without adjusting the average eCl values for populations one and two.

Our modelling of natural and synthetic (McDannell and Issler, 2021) AFT data demonstrates that multikinetic samples can

retain a record of multiple heating and cooling events under suitable geological conditions due to the relative annealing behaviour of kinetic populations that are sensitive to different parts of the thermal history. The ability to recover multicycle



histories depends on many factors including sufficient and well distributed AFT and elemental data to constrain kinetic population interpretations, a favourable thermal history (decreasing thermal intensity with time) that caused significant annealing of multiple kinetic populations, and appropriate application of modelling strategies. Even if the first two
conditions are met, careful modelling is needed to obtain successful solutions. Nondirected Monte Carlo modelling software such as AFTINV and HeFTy depend on user-defined model boundary conditions and kinetic parameter values — often requiring iterative modelling to refine model parameters to obtain solutions. Overly simplistic thermal histories or unsuitable kinetic parameters may result in no or few solutions and this may be attributed incorrectly to problems with the data. For example, a two-cycle heating/cooling model failed to yield solutions for our high-quality Devonian sample. When boundary
conditions were adjusted to allow for the possibility of an extra heating event, the model converged on three well resolved heating cycles, a scenario that is compatible with available geological constraints and is independently supported by QTQt model results without user-imposed boundary conditions. The QTQt model allows for looser boundary conditions and greater uncertainty in the style of thermal history because it implements a reversible jump Markov chain Monte Carlo optimisation algorithm that automatically modifies kinetic parameters (within specified ranges) and the number of heating
cycles to try to fit the input data.

Clearly not all samples will be multikinetic, and not all multikinetic samples will share a common thermal history if there has been insufficient annealing to eliminate differences in provenance (i.e., inherited pre-depositional histories). Nevertheless, we have seen many cases where a common thermal history works for multikinetic samples. Possible
explanations for this include rapid exhumation of heterogeneous source areas, strong annealing/thermal resetting of some or all AFT populations, and mixing of detrital and syn-depositional volcanic components. If interpreted populations are incompatible, then solutions will not be obtained by adjusting kinetic parameters. Ultimately, the value of the approach will be judged on its ability to generate spatially coherent thermal histories over different stratigraphic intervals across study regions. An encouraging sign is that we have been able to use essentially the same kinetic parameters for samples from the
same stratigraphic units in the Yukon that have experienced different degrees of heating. We strongly recommend that elemental data be collected for detrital AFT samples from areas with complicated geological histories and for other samples with unexplained age dispersion. Age dispersion is a desirable characteristic of samples when viewed in a multikinetic framework due to the potential for enhanced thermal-history resolution.

**6. Conclusions**

It is common for detrital AFT samples of variable cation and anion composition to have significant age dispersion that is caused by multikinetic annealing. Under these conditions, AFT age and length data can be sorted into discrete kinetic populations with different annealing temperatures using eCl values (derived from $r_{mr0}$ values obtained using multi-elemental





data). In general, these kinetic populations are unresolved or poorly resolved using the conventional single kinetic parameters, $D_{par}$ or Cl content; $D_{par}$ has low precision and Cl alone neglects how other elements influence track retentivity.

Modelling of dozens of samples from northern Yukon, two of which are presented here, indicates that a complicated record of multiple heating and cooling cycles can be retained in multikinetic samples under certain geological conditions (heating cycles of decreasing intensity through time) due to the different relative annealing behaviour of the kinetic populations. Accurate prediction of elemental-derived kinetic parameters is unlikely and may not be attainable for all natural apatite populations, however this is not a requirement for using the method and the same problem exists with conventional kinetic

parameters. Absolute kinetic parameters are best constrained for eCl values within the range of 0–0.25 apfu ($r_{mr0}$ values of 0.73–0.84) which represents the more commonly encountered apatite compositions from published annealing experiments. Effective Cl values in this range can be fixed in the model and inaccurately predicted parameters for more exotic apatite compositions can be adjusted by exploiting the relative annealing inherent in the $r_{mr0}$ model. As expected, the uncertainty range on the less constrained higher eCl values increases as the degree of AFT annealing decreases. Overall, the model is

tolerant of these uncertainties and a range of eCl values for higher retentivity populations can still produce similar solutions at the 0.5 significance level. Considering these results, age dispersion in multikinetic samples should be viewed as desirable for enhancing thermal history resolution rather than as hindrance to data interpretation and modelling.

## 7. Supplement

The Supplement contains the data for figure 2.

## 660  8. Author Contributions

DRI developed the method, interpreted and modelled the data using AFTINV, and wrote the paper. KTM performed QTQt modelling, was involved in conceptual discussions, and contributed to writing and editing the paper. PBO did the AFT analysis (first at A to Z Inc. and then at GeoSep Services) and coordinated the elemental analyses at Washington State University. LSL collected the samples, arranged for the analyses, and provided geological context for the study.

## 665  9. Competing interests

The authors declare that they have no conflict of interest.

## 10. Acknowledgements

These ideas were developed over many years of working with detrital AFT samples and benefitted from many influences. Early on, Ray Donelick impressed upon DRI how widespread age dispersion is in detrital samples and his pioneering work

with the LA-ICPMS AFT method enabled us to acquire the data for this study. Richard Ketcham generously helped DRI to



better understand his annealing model so it could be incorporated into AFTINV. Careful AFT and multi-elemental analyses by Sandy Grist (formerly of Dalhousie University) for samples from the Mackenzie Delta region provided the first clear evidence for distinct kinetic populations in Canadian samples. We thank Chance Oil and Gas Ltd. (formerly Northern Cross (Yukon) Ltd.) for providing the well sample used in this study. Owen Neill (formerly of the Peter Hooper GeoAnalytical

Laboratory, Washington State University) and Ken Severin (Advanced Instrumentation Laboratory, University of Alaska Fairbanks) are recognized for EPMA analyses and their assistance in organizing and processing the data. This study has been supported through the Natural Resources Canada Geo-Mapping for Energy and Minerals Program Yukon Basins Project. NRCan contribution number ########.

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
