# Peer review of "Simulating sedimentary burial cycles — Part 2: Elemental-based multikinetic apatite fission-track interpretation and modelling techniques illustrated using examples from northern Yukon"

_Geochronology, 2021_

## Community Comment (CC1)

In his Review of the manuscript by Issler et al., Ketcham questions the costs vs the benefits of analysing compositions for the purpose of extracting thermal history information from apatite fission track data. Ketcham suggests that a major reason that this has been largely neglected to date is the "trouble and expertise" required to acquire compositional data, "since the rewards are unclear, especially since thermal history inversion software will often produce a result without it".

This suggests that compositional effects are regarded as a secondary issue, and perhaps a luxury that cannot be afforded. Possibly even a waste of time!

We find it frustrating (to say the least) that the importance of allowing for the influence of composition in extracting thermal history solutions from apatite fission track data should still be the subject of debate. We first demonstrated the systematic influence of chlorine content on annealing kinetics in apatite over 35 years ago, in a sample from a present day temperature of 95°C (Figure 1: from Green et al., 1986).

[Figure]

Figure 1:

Since then, a number of laboratory studies have provided further evidence in support of that conclusion. Carlson et. al. (1999) and Barbarand et al. (2003) explicitly downplayed the importance of Cl and highlighted other factors, but as shown by Green and Duddy (2012) and as illustrated here in Figure 2, both datasets clearly display the systematic influence of wt% Cl on annealing sensitivity (below), while other elements showed no systematic effects on annealing.

[Figure]

Figure 2:

In these plots, $F(t,T) = [\log t - \log t_0]/[(1/T) - (1/T_0)]$ with $\log t_0 = -10$, and $1/T_0 = 0.001$.

In considering these plots, it is important to realise that detrital apatites from most sandstone samples contain a spread of wt%Cl from 0 to ~0.5 or more.  In the appropriate temperature range the variation in annealing sensitivity over this range is pronounced.  Accessory apatites in crystalline rocks also often display significant variation in wt% Cl.

Green and Duddy (2012), following up one results shown in Figure 1, demonstrated the systematic influence of wt% Cl on annealing sensitivity in geological conditions, in a detailed study of data in core samples from  present-day temperatures between 95 and 124°C in the Flaxmans-1 well in the Otway Basin (Figure 3):

[Figure]

Figure 3:

In both the Carlson et al. (1999) and Barbarand et al. (2003) studies a small number of apatites analysed showed results that differ from the systematic trends dominated by wt% Cl (Fig. 2), suggesting that additional influences may exist.  In recent years, the effects  of multiple elements on annealing have been incorporated into kinetic descriptions via the $r_{mr0}$ term (Ketcham et al.1999, 2007), as adopted by Issler et al. in the paper under review  However, the relationship between $r_{mr0}$ and various elements is not well defined, leading Carlson et al. (1999) to urge caution in its application (caption to their Figure 5).

We have published a number of natural datasets in which fission track age varies systematically with wt% Cl in samples which have been heated to the appropriate temperature where differential annealing kinetics in apatites with different chlorine contents are sufficient produce a range of responses (varying degrees of age reduction).  These studies also highlight a number of outliers which may correspond to more unusual compositions similar to those highlighted by Carlson et al. (1999) and Barbarand et al. (2003).  Experience has shown that such anomalous data are rare, and can readily be identified as outliers in trends of age or length vs wt% Cl and eliminated prior to interpretation (Figure 4, from Green and Duddy, 2012).  In the vast majority of cases, variable levels of annealing in apatites from a single sample can be described solely by variation in wt% Cl.

[Figure]

Figure 4

These examples illustrate that failure to incorporate compositional influences on annealing kinetics in extracting thermal history must lead to inaccurate and misleading results.  To question the costs vs benefits of doing so is to accept a result that may not even represent a rough approximation to the true history, instead of performing the analysis in the most technically appropriate fashion.

In this respect, we find it amusing to contrast the negative attitude to compositional analysis evident in Ketcham's review with the willingness of many thermochronology labs to invest in expensive new machines to measure uranium for fission track studies by laser ablation, and to measure U-Th/He ages, when these methods are far from proven and have been shown to provide misleading results in many cases.

Issler et al. in the paper under review are to be commended for emphasising the importance of variation in annealing systematics between apatite grains within a single rock sample.  We do not find their approach convincing, based as it is on investigation of ages and lengths vs $r_{mr0}$ rather than wt% Cl.  But that is another issue which we are not concerned with here.  We simply ask why investigation of compositional effects in apatite fission track data, for which overwhelming evidence exists, is apparently considered an unnecessary luxury, especially when so much money has been devoted to investment in other aspects of thermochronology which are far from proven.

**References**

Barbarand, J., Carter, A., Wood, I. & Hurford, A.J. 2003: Compositional and structural control of fission track annealing in apatite. Chemical Geology 198, 107–137.

Carlson, W.D., Donelick, R.A. & Ketcham, R.A. 1999: Variability of apatite fission-track annealing kinetics: I. Experimental results. American Mineralogist 84, 1213–1223,

Ketcham, R.A., Donelick, R.A. & Carlson, W.D. 1999: Variability of apatite fission-track annealing kinetics: III. Extrapolation to geological timescales. American Mineralogist 84, 1235–1255.

Green, P.F., Duddy, I.R., Gleadow, A.J.W., Tingate, P.R., and Laslett, G.M. 1986 Thermal annealing of fission tracks in apatite 1. A qualitative description. Chemical Geology (Isotope Geoscience Section), 59, 237–253.

Green, P.F. & Duddy, I. R. 2013: Thermal history reconstruction in sedimentary basins using apatite fission-track analysis and related techniques, in Harris, N.D. & Peters, K. (eds): Analyzing the thermal history of sedimentary basins: methods and case histories. SEPM Special Publication 103, 65–104.

Ketcham, R. A., Carter, A. C., Donelick, R. A., Barbarand, J., and Hurford, A. J., 2007, Improved modeling of fission-track annealing in apatite: American Mineralogist, 92, 799-810.

Paul Green & Ian Duddy
8th October 2021

---

## Community Comment (CC3)

**Comment on:**

Issler, D.R., McDannell, K.T., O'Sullivan, P.B. and Lane, L.S., in review. Simulating sedimentary burial cycles – Part 2: Elemental-based multikinetic apatite fission-track interpretation and modelling techniques illustrated using examples from northern Yukon.

MS No.: gchron-2021-22

**Summary**

Serious problems with the quality of the EMPA and LAICPMS AFT data used in this paper irrevocably compromise the conclusions concerning the use and superiority of $rmr_0$ over chlorine (wt%) as a kinetic control on apatite fission track annealing. Because of these problems the subsequent thermal history modelling has no basis.

**EMPA data.**

Of the 92 apatite compositions reported for Devonian outcrop sample LHA003, 59 (~64%) have a total outside the 98-101 wt% range generally regarded as defining acceptable totals (as low as 82.4%). Poor EMPA analyses are alluded to when discussing sample PO13-12 (lines 336-340), but the scale of the issue was not mentioned. Further, Issler et al. violate their own workflow Step 7, in Lines 105-110:

 *"Step 7 involves assessing the interpretation by considering all available data in the context of measurement uncertainty and missing information. The goal here is to try to use all the available data except for obviously poor analyses."*

The majority of the EMPA data for LHA003 are clearly *"obviously poor"* and are thus unsuitable both for determining structural formulae and subsequently $rmr_0$ and eCl for defining kinetic populations.

The structural formulae calculations for LHA003 presented in Table D.2 ("assets link"; Issler et al., 2021) were apparently made using a non-standard procedure, markedly different to that described in Ketcham (2015), for example, which was used by the authors for sample PO13-12. It appears that the calculations for LHA003 have been made by assuming a stoichiometric ratio for the Ca and P sites, whereas the raw data show that most analyses have very low totals and the analyses are clearly not stoichiometric (Figure 1). Even though we show that most analyses for LHA003 are unsuitable for structural formula calculation, we have followed the authors in doing this, but using the same methods as they used for sample PO13-12, to demonstrate the effect that the poor totals have on subsequent calculations.

The relationship between EMPA total and the ratio (in apfu) of the Ca to P sites (stoichiometric ratio ~1.67) is shown in Figure 1. The obvious correlation between decreasing total and increasingly non-stoichiometric B/A ratio (i.e. Ca site/P site) provides a strong indication that there is a serious problem with the EMPA analyses with unequal effects on different elements. The potential effect on the reported Cl and F values cannot be determined. In other words, these analyses cannot be simply normalised to 100% and used to estimate the structural formula, as may have been done to produce the data listed in Table D.2.

We contend that the values of $rmr_0$ and eCl determined from most of these apatite analyses are inaccurate and unfit for purpose. This is confirmed with plots of these parameters in Figure 2, where increasingly lower totals correlate with lower $rmr_0$ and higher eCl values.

Poor totals will have a direct influence, for example, on the calculation of OH which is determined by difference, and this will feed through to $rmr_0$.

Thus, we see no justification for the conclusion stated in lines 344-345:

*"The Devonian outcrop sample is of high quality with 39 single grain ages and 202 track lengths that clearly define two robust kinetic populations in eCl-space (Fig. 5a, b)."*

It follows that relationships shown in Figure 5 of Issler et al. cannot be relied upon as showing the $rmr_0$ is a useful discriminator of AFT annealing behavior.

Is the EMPA data for Permian cuttings sample PO13-12 any better?

Thirty four of 78 EMPA analyses for PO13-12 (~44%) are less than 98% (as low as ~93.8%) and while the data set appears to be of higher quality than that presented for LHA003 (i.e. totals do not get extremely low), a similar trend been EMPA total and B/A ratio is displayed (Figure 3) with resulting question marks over the calculated $rmr_0$ and eCl values.

We conclude that the values of $rmr_0$ and eCl calculated by Issler et al for sample PO13-12 cannot be used to accurately characterize the properties of individual apatite grains and cannot therefore provide a reliable discrimination of differential AFT annealing.

Why are the EMPA analyses so poor and how can it be done better?

The analytical strategy described by the authors in Figure 1 of Issler et al appears to be the main reason their EMPA analyses are so poor, with probing relegated to the last step in the chain, despite apparently being aware of the problems resulting from this workflow for at least a decade (results for LHA003 were produced in 2011).

We are currently undertaking a research study investigating the usefulness of $rmr_0$ and LAICPMS for AFT U-determination and as part of this study we produced 1057 full EMPA analyses on apatite. Of these analyses, only 4, or <0.4 %, fell outside the range of acceptable totals noted above. This should be the norm for any apatite EMPA study.

We achieved this high quality data by simply carrying out EMPA after FT ages and lengths were measured and prior to ablation. In this way we were able to analyse the same area in which spontaneous tracks were measured and to put the ablation spot at the same location, avoiding the problems encountered in the MS under review.

**LAICPMS AFT Data**

The complete raw LAICPMS data for the two samples described in this study are not provided, only a reduced data summary available under the assets link (AFT_age_Tables B.1 and B.2 and AFT_length_Tables C.1 and C.2; Issler et al 2021). Note that key details such as the single ablation spot size (16 $\mu$m) and depth (16 to 18 $\mu$m), laser conditions, etc. are not provided in the MS (but can be found in Issler et al., 2021). In addition, the "primary" zeta value quoted is not enough by itself for calculating the AFT ages from the tabulated results.

The data for sample PO13-12 shows evidence for significant U-zoning in many grains although no mention is made of this in the paper under review. Two versions of uranium magnitude are tabulated: 1) the $^{238}U/^{43}Ca$ ratio used to determine the FT age and 2) U (ppm). These values are not equivalent, as apparently the former is a depth-weighted mean value and the latter a simple average over the full ablation depth (16-18 $\mu$m). A plot of one against the other for sample PO13-12 provided in Figure 4 reveals what we consider to be clear

evidence for U-zoning, with some grains showing extreme excursions from the one to one line. The possible effect of this zoning on the AFT ages and the trends with chemical composition should be discussed. From the unreviewed online assets link (Issler et al., 2021) high U ppm values are attributed to the presence of high U inclusions, but we feel U-zoning is more likely.

One important corollary of this observation is that the U, Th and Sm ppm values listed for these samples should not be used to calculate, for example, eU as a proxy for radiation damage, as they do not refer to the same sample volume. Some comments on the tabulated U, Th and Sm values are provided in Issler et al. (2021) to the effect that they are not absolute measurements due to changes in the U/Ca ratio between analytical sessions, referencing Cogne et al. (2020). In fact, Cogne et al. (2020) recommend against even reporting U ppm values (and presumably also Th and Sm) determined by LAICPMS as they are not accurate. This is a stark reminder that the generally increased precision of U-determination by LAICPMS over EDM is at the expense of accuracy. Here it is important to note that Seiler et al. (2013) found that U-determination by LAICPMS were systematically low at values less than ~5 ppm, while our own experience suggests this limit is at best 10 ppm. This leads to AFT ages which are too old, a major reason for the additional dispersion in LAICPMS AFT ages compared to EDM (e.g. Ketcham et al. 2018).

We are also concerned with additional unaccounted inaccuracies introduced by assuming that each of the analysed apatites are stoichiometric, with 53.454 wt% CaO assumed (McDannell et al 2019) in order to determine the U concentration of the ablated volume and calculate the AFT age. Considering only acceptable EMPA totals for both samples, CaO values in nearly all grains exceed this level (see Elemental wt% Oxide Tables in Issler et al, 2021).

We have some other concerns with the LAICPMS data integrity. For example, three grains from PO13-12, the two youngest and the oldest are rejected from the data set before assessment against eCl, on the grounds of poor U-analysis? (Table B.1). The oldest grain was also rejected from sample LHA003 on the same grounds.

The reader is entitled to ask why these should be regarded as poor U-determination, while the rest are considered OK. In this respect we note that in an earlier paper involving most of the same authors that attempts to resuscitate the concept of radiation enhanced annealing (McDannell et al., 2019) such data appear to have been accepted without question. Other grains have $rmr_0$ compositions that do not accord with the defined age populations and these are arbitrarily assigned to another population without adequate explanation (Lines 330-340). The same arbitrary reassignment was not carried out in relation to evaluating age trends with Cl apfu which the authors regard as inferior to $rmr_0$.

Further comments on this aspect of the MS are beyond the scope of this comment, but these examples highlight one of the major shortcomings of LAICPMS compared to EDM. With EDM the user has a permanent record of the U-distribution and magnitude in the mica detector and any anomalies can be checked and evaluated. With LAICPMS the evidence of U-distribution is difficult to obtain as the area ablated is generally much smaller than the area in which spontaneous tracks are measured (as applies to most grains in this study) and the ablation destroys the sample. We agree with reviewer Karl Lang that the supposed advantages of LAICPMS over EDM are overstated.

**$rmr_0$ calculations**

The authors prefer the use of the $rmr_0$ relationship of Carlson el at (1999) over the updated

version of Ketcham et al (2007) on the grounds that "it resolves kinetic populations better (less grain overlap) for the samples we have studied" (Line 212).

It is worthwhile to review the two relationships for $rmr_0$.

1)  Carlson el at (1999), equation 6:

$$rmr_0 = [0.027 + 0.431 \text{ ABS}(Cl - 1) + 0.107 \text{ ABS}(OH - 1) - 1.01 \text{ Mn} - 2.67 \text{ Fe} - 0.144 \text{ Others}]^{0.25.}$$

The "others" term refers only to the total (in apfu) of the cations measured in the original Carlson et al (1999) data set that substituted for Ca: Sr, Ce, La and Na.

We concur with the warning of Carlson et al (1999): Quote:

"The relative magnitudes of the coefficients can be used together with the data in Table 4 to estimate the relative significance of each of these compositional variables. However, in the absence of any physical understanding of why compositional variations impede or enhance annealing, we have little confidence that it can be used meaningfully to predict the annealing behavior of apatites not included in the experiments. For example, the effect of Fe concentration is determined entirely by the annealing behavior of a single extremely Fe-rich apatite. Nonetheless, these data appear to suggest that substituents for Ca tend to reduce rates of annealing, and that annealing kinetics depend upon the degree of mixing on the halogen site, in some still-concealed and probably complex way. "

It is unclear whether the $rmr_0$ calculations in the paper under review conform to the original equation. Ti, Zr, Al, As, Y, Sm, Nd, Mg, Ba and K were additionally measured for sample LHA003 and Mg and Y, not measured by Carlson et al (op.cit.), were additionally measured in PO13-12 but it is not stated whether any of these elements have been used to determine "others" for $rmr_0$. We have a suspicion that some may have been included since our own calculations of $rmr_0$ from these data (following the authors and ignoring the low EMPA totals) are a little different to those in the MS. Most are not greatly different, but in two grains with >2wt% Cl, our calculations give negative pre-power values, preventing calculation of $rmr_0$ and thus eCl, whereas finite values are listed for these grains in Supplementary Table S2 and are shown in the plots.

2)  Ketcham et al. (2007), equation 11:

$$rmr_0 = [-0.0495 - (0.0348 \text{ x F}) + 0.3528 \text{ ABS}(Cl - 1) + 0.0701 \text{ ABS}(OH - 1) - (0.8592 \text{ x Mn}) - (1.2252 \text{ x Fe}) - (0.1721 \text{ x Others})]^{0.1433.}$$

We concur with the further warning regarding use of $rmr_0$ of Ketcham et al (2007): Quote:

"Others is the sum of all other cation substituents aside from Mn and Fe. The relative magnitudes of the coefficients are broadly similar to those obtained by Carlson et al. (1999, Eq. 6), with the main difference being to de-emphasize Mn and OH. Attempts to fit different sets and combinations of compositional variables were less successful. The relationship between measured and estimated $rmr_0$ is shown in Figure 7; most estimates are within 0.02 of the correct figure, which translates into a roughly 5 to 10 °C uncertainty in closure temperature. However, we reiterate here the warning of Carlson et al. (1999) that this approach should only be used with caution, as the cation-based terms are poorly constrained and are likely to be oversimplified and have nonlinear effects. Furthermore, there is no physical basis for the form of Equation 11, and a different form could result in a more confident basis for extrapolation."

Issler et al. dismiss these warnings (Lines 555-561) and proceed without regard to the limited constraints on chemical composition available in the original Carlson et al (1999) annealing data set, for which no follow-up studies that might confirm or otherwise the importance of elements other than Cl have been undertaken in almost the last 2 decades.

A summary of the range of elements (apfu) in each sample is provided in Issler et al. (Table 3). We think it noteworthy that no element that is significant to the $rmr_0$ calculation is particularly abundant. Fe and Mn do not exceed 0.1 and 0.11 apfu (~0.41 and 0.5 wt%), respectively, in either sample while SrO is < 1.8wt%. No other cation exceeds 1 wt% oxide. SO3 reaches no higher than 1.2 wt% in PO13-12, but this element is not included in $rmr_0$. SiO2 reaches 1.17wt% in HA003, but again this element is not a part of $rmr_0$. These values are very much less than measured in the single high Fe, Mn and Sr apatites in the Carlson et al. (1999) data set.

What is controlling $rmr_0$ in the two samples?

The major control on $rmr_0$ is clearly the chlorine content above all other elements, as shown by a plot of wt% chlorine versus $rmr_0$ for PO13-12 and LHA003 provided in Figure 5.

This observation accords with the observation of Barbarand et al. (2003) , Quote:

**"for samples with Cl>0.1 apfu *(~0.35 wt%)*, Cl is the dominant control on track annealing, effectively masking any effect from Ce or other REE substitutions. The presence of large numbers of cations substituted for Ca exhibits some correlation with the annealing properties of F-apatites (Crowley et al., 1991; Carlson et al., 1999). In our study, little information about how Sr, Mn and Fe control mean track length was forthcoming because of the low concentrations of the elements in our samples (see Table 1). Seemingly very high concentrations are required to modify apatite track annealing properties; in the studies cited above, samples with 4.44 wt.% of SrO (Crowley et al., 1991), 7.04 wt.% of Mn and 9.15 wt.% of SrO (Carlson et al., 1999) were considered. For most apatites where Sr, Mn and Fe concentrations are low, the effect of these elements may be safely ignored".**

So why do the apatites in LHA003 have $rmr_0$ values that range down to 0.31 (in our calculations this value is also incorrect – see above) if none of the measured elements are outside the typical range of 'normal' apatite? The answer is that Cl is having the dominant effect (Figure 5) and major excursions from the trend probably reflect the low quality of the EMPA analyses.

**Multikinetic annealing**

We have investigated the effect of apatite composition of AFT annealing since the early 1980s and have incorporated the chlorine content in our work since for over 30 years. Despite the overwhelming evidence of the importance of Cl (e.g. Green and Duddy, 2012), we are bemused at the reluctance of the community to take it on board. While we support Issler et al. in attempting to promote the use of apatite composition, we believe their attempt is misguided, and that the approach of collecting "complete" compositional data and calculating $rmr_0$ is unnecessary. Simpler and better results can be achieved by determination of chlorine alone. The paper in review falls far short in demonstrating that $rmr_0$ is superior to Cl (wt%) alone.

**Further comments for the benefit of the authors**

Should the authors decide to continue with trying to demonstrate that $rmr_0$ is useful as a measure of annealing, we suggest that this can never be achieved with the analysis of further outcrop samples or well samples of the type used in this study (and apparently in a upcoming analysis of 50 further similar samples).

The only rigorous strategy is to use core samples from deep wells that *are currently at maximum temperatures at the present-day* (e.g. like those for the Flaxmans-1 well - Green and Duddy, 2012 – see also Geochron CC1 reply on RC1), or to subject outcrop samples to laboratory annealing. For example, sample LHA003 would be a good candidate for an annealing experiment, where both induced and spontaneous tracks could be used with a specific aim at investigating potential compositional controls in apatite with <0.4wt% Cl. Such a controlled experiment, and accurate EMPA analyses, would overcome the major shortcomings evident in the paper under review.

Further, the approach employed by Issler et al in defining age populations and then applying an arbitrary compositional boundary is backwards. If apatite composition is important, it is important in all samples, regardless of the range of AFT ages. Compositional boundaries should be defined based on the annealing kinetics of known compositions, and the ages in each group should be compared with model ages to derive a common thermal history. In our approach we have derived individual kinetic descriptions for chlorine (wt%) compositional groups at 0.1 wt% intervals up to 3 wt%. Typical quartzose-arkosic samples have Cl between 0 and ~0.6 wt% (6 compositional groups) while volcanogenic sandstones range up to ~3 wt% (30 groups) (e.g. Figures 18, 16 and others in Green and Duddy, 2013). Such an approach enables tighter constraints on maximum paleotemperature and time of cooling than available from assuming composition, or using an average value for a wide range of compositions as employed in the MS under review

In developing a more detailed understanding of compositional influences on FT annealing, cuttings samples should be avoided due to potential down hole or drilling additive contamination.

We also suggest that LAICPMS U analyses are avoided due to the current uncertainties surrounding the accuracy of U-determinations (Seiler et al 2013; Cogne et al 2020; Duddy and Green, in prep).

Ian Duddy and Paul Green

October 15[th] 2021

**REFERENCES**

Barbarand, J., Carter, A., Wood, I. and Hurford, T.: Compositional and structural control of fission-track annealing in apatite, Chem. Geol., 198(1–2), 107–137, 2003.

Carlson, W. D., Donelick, R. A. and Ketcham, R. A.: Variability of apatite fission-track annealing kinetics: I. Experimental results, Am. Mineral., 84(9), 1213–1223, 1999.

Cogne, N., Chew, D. M., Donelick, R. A. and Ansberque, C.: LA-ICP-MS apatite fission track dating: A practical zeta-based approach, Chem. Geol., 531, doi: 10.1016/j.chemgeo.2019.119302, 2020.

Crowley, K. D., Cameron, M. and Shaefer, R. I.: Experimental studies of annealing of etched fission tracks in fluorapatite, Geochim. Cosmochim. Acta, 705 55(5), 1449–1465, 1991.

Issler, D.R., McDannell, K.T., Lane, L.S., O'Sullivan, P.B. and Neill, O.K. 2021. A multikinetic approach to apatite fission-track thermal modelling using elemental data: data and model results for a Permian and Devonian sample from northern Yukon. Geological Survey of Canada, Open File 8821, 1 .zip file. https://doi.org/10.4095/328844

Issler, D.R., McDannell, K.T., O'Sullivan, P.B. and Lane, L.S., in review. Simulating sedimentary burial cycles – Part 2: Elemental-based multikinetic apatite fission-track interpretation and modelling techniques illustrated using examples from northern Yukon MS No.: gchron-2021-22.

Ketcham, R. A.: Calculation of stoichiometry from EMP data for apatite and other phases with mixing on monovalent anion sites, Am. Mineral., 100(7), 1620–1623, 2015.

Ketcham, R. A., Carter, A., Donelick, R. A., Barbarand, J. and Hurford, A. J.: Improved modeling of fission-track annealing in apatite, Am. Mineral., 92(5–6), 799–810, doi:10.2138/am.2007.2281, 2007.

Ketcham, R. A., van der Beek, P., Barbarand, J., Bernet, M. and Gautheron, C.: Reproducibility of Thermal History Reconstruction From Apatite Fission-Track and (U-Th)/He Data, Geochemistry, Geophys. Geosystems, 19(8), 2411–2436, doi:https://doi.org/10.1029/2018GC007555, 2018.

McDannell, K.T., Issler, D.R., O'Sullivan, P.B., Radiation-enhanced fission track annealing revisited and consequences for apatite thermochronometry, *Geochimica et Cosmochimica Acta* (2019), doi: https://doi.org/10.1016/j.gca.2019.03.006

Seiler, C., Gleadow, A. J. W. and Kohn, B. P.: Apatite fission track dating by LA-ICP-MS and External Detector Method: How do they stack up?, in American Geophysical Union, Fall Meeting 2013, pp. T42C-07, San Francisco, California. [online] Available from: https://ui.adsabs.harvard.edu/abs/2013AGUFM.T42C..07S/abstract, 2013.

[Figure]

**Figure 1:** Sample LHA003 apatite EMPA analysis total versus Ca site:P site ratio (B/A apfu). 59 of 92 analyses (64%) are outside the acceptable range. The strong correlation between low total and increasingly non-stoichiometric B/A ratio (i.e. Ca site/P site) is in stark contrast to the apatites used by Carlson et al (1999) to define $rmr_0$. (Note that the apatites used in annealing experiments by Barbarand et al 2003) fall in similar field to those from Carlson et al., 1999).

[Figure]

**Figure 2:** Sample LHA003 apatite EMPA analysis total versus Rmr0 and eCl.

The clear trend of decreasing $rmr_0$ (A.) and increasing eCl (B.) with decreasing EMPA total shows that the differences in fission track retentivity interpreted by the authors from these parameters is likely an artifact of the poor quality EMPA data and provides a strong indication that values of $rmr_0$ and eCl determined from most of these apatite are inaccurate, and unfit for purpose (also see Figure 3).

[Figure]

**Figure 3:** Sample PO13-12 apatite EMPA analysis total versus Ca site:P site ratio (B/A apfu). ~44% of EMPA totals are outside the acceptable range. The strong correlation between low total and increasingly non-stoichiometric B/A ratio observed for sample LHA003 is less stark for this sample, but still evident. Dashed field is the range of values for the apatites used by Carlson et al (1999) to define $rmr_0$.

[Figure]

[Figure]

[Figure]

**Figure 4:** Sample PO13-12 238U/43Ca ratio verus U (ppm) at three scales. Major excursions from a 1:1 trend at all U levels strongly suggests U-zoning.

[Figure]

**Figure 5:** A strong corrleation is seen between Chlorine wt% and $rmr_0$ indicating that variation in $rmr_0$ largely results from variation in chlorine content (values taken from Tables D1, D2 and Elemental oxide data tables of Issler et al, 2021).

Major excursions from the general trend, especially for sample LHA003, are attributed to the very poor EMPA analyses for this sample.

---

## Community Comment (CC4)

**Author response to community comment for preprint gchron-2021-22**

Issler, D. R., McDannell, K. T., O'Sullivan, P. B., and Lane, L. S.: Simulating sedimentary burial cycles – Part 2: Elemental-based multikinetic apatite fission-track interpretation and modelling techniques illustrated using examples from northern Yukon, Geochronology Discuss. [preprint], https://doi.org/10.5194/gchron-2021-22, in review, 2021.

We appreciate that Ian Duddy and Paul Green of Geotrack Intl. took the time to comment on our manuscript in review with GChron (doi: 10.5194/gchron-2021-22). The relationship between apatite composition and annealing is elegantly but incompletely captured by the single kinetic parameter $r_{mr0}$—we were fully transparent about this in our manuscript (lines 548–561 and 650–653). To simply dismiss this complexity as immaterial rather than to investigate it further (when in an obvious position to do so), is in our mind a disservice to the thermochronological community. While Carlson et al. (1999) cautioned the reader on use of the $r_{mr0}$ parameter outside of the examined apatite compositional range, this does not mean it should be disregarded entirely, as implied by Green and Duddy. It is therefore not surprising that the Geotrack authors "concur" with the cautionary statements in Carlson et al. (1999) and Ketcham et al. (2007) regarding $r_{mr0}$.

The initial Green and Duddy comment (https://doi.org/10.5194/gchron-2021-22-CC1) on R. Ketcham's review (https://doi.org/10.5194/gchron-2021-22-RC1) was not meant to address our manuscript under consideration, but to defend what they view as their methods (using only Cl content). We would ask, what are the motivations of the authors for making these claims? While we commend them for being pioneers in AFT methodology, they operate outside the sphere of academic research and benefit from marketing their in-house methods and data interpretations. Have the *details* of the Geotrack "multi-compositional" kinetic model, whose results are illustrated in Green and Duddy (2012) undergone formal peer review? The kinetic models in the aforementioned papers have met this condition.

Duddy asks (https://doi.org/10.5194/gchron-2021-22-CC3) *"what is controlling $r_{mr0}$ in the two* [Issler et al.] *samples?"*—then answers that it is *clearly* chlorine content. We would respond by saying that Cl is unquestionably significant and is a key element in the $r_{mr0}$ relation. It is obvious in our figures 4 and 5 that Cl is a dominant element and we do not deny the importance of Cl for track annealing. It is also evident that our interpreted kinetic populations overlap when considering only Cl, and better population definition occurs with $r_{mr0}$ (kinetic populations also align with radial plot mixture modelling age peaks). The Geotrack authors rely on an ad hoc approach where apatite grains are binned by Cl (wt%), which will not work if a sample contains heterogenous apatite compositions. We assert that because the AFT method is rooted in population statistics—if a sample is revealed to contain grains of different 'true' ages—that kinetic populations should have some quantitative basis relative to such component ages. This was conceptually discussed in Galbraith and Green (1990), among other publications and is a powerful tool used in many geochronological applications.

We demonstrated that *both* measured Cl and $D_{par}$ are inadequate for multikinetic interpretation (in our samples), so why do Green and Duddy only take issue with Cl? The answer may be that the kinetic parameter $D_{par}$ is low precision and often has poor resolving power for distinguishing kinetic populations and Green and Duddy agree with this statement (our Figs. 2, 4, and 5 and Green et al., 2005 comment on Barbarand et al., 2003; Green and Duddy, 2012; their Fig. 8). So why is Cl any different than $D_{par}$ in the results we discuss? Does Cl always work well for every sample? It will if Cl really is the only element in abundance. It can 'appear' to work if significant elements covary systematically with Cl. In other cases, where different elements are important, a Cl-only based framework could lead authors to dismiss data that do not conform to the expected trend as outliers. Imposing an interpretation framework onto which the data must conform is untenable. If the results of many annealing experiments show that elements other than Cl affect track retentivity, then the relationships exist—regardless of elemental preference or other yet unexplained phenomena that complicate the annealing dependence on apatite composition. These relationships are not negated just because the Geotrack authors prefer Cl. This is confirmation bias that disagrees with empirical evidence. We believe the multikinetic results of Carlson et al. (1999), Barbarand et al. (2003), and Ketcham et al. (2007) were compelling, albeit complicated, and should have led to a flourish of further AFT research into apatite composition/annealing, just like what was done with Cl in the late 1980s. This did not happen.

In their own published work, they show arbitrary designation of grains as "likely outliers" (as an example Fig. 4 of Green comment, https://doi.org/10.5194/gchron-2021-22-CC1). To quote Duddy, "The reader is entitled to ask why these [outliers] should be regarded as poor?" —this label does little to appease the reader or impart any confidence in this purported designation. Green and Duddy can assert that elements other than Cl are trivial, or argue about nuance

(i.e., specific elemental abundance thresholds for annealing or lack of a physical mechanism to explain other elemental effects on annealing behavior) because this fits their preconceptions and there are conveniently no definitive studies to prove otherwise.

Duddy in his comment summarizes with: *"Serious problems with the quality of the EMPA and LAICPMS AFT data used in this paper irrevocably compromise the conclusions concerning the use and superiority of rmr0 over chlorine (wt%) as a kinetic control on apatite fission track annealing. Because of these problems the subsequent thermal history modelling has no basis".* We find this statement interesting, considering that our data and figures clearly show trends that are not refuted by their discussion. In a literature survey of (only) the last 21 years, we cannot find a single instance of a peer-reviewed publication that contains all raw fission track or electron probe data produced by Geotrack. We leave open the rare possibility that we may be wrong in this appraisal but consider it unlikely given that the authors can claim data as proprietary without undergoing public scrutiny. Only reporting summary tables of AFT central ages, binned track length information, and single (or a range of) wt% chlorine values are in no way transparent for data quality assessment, adequate for high-quality peer review standards, and are certainly not reproducible. In addition, we can find no strong, available evidence for a positive assessment of, or comment on, any new fission-track research in the last 21 years either.

The chief focus on the electron microprobe (EPMA) data in Duddy's comment CC3 delves into minutiae to distract from the important matter at hand regarding the application of multikinetic AFT for thermal history analysis. Low elemental wt% totals can occur for a variety of reasons, namely grain damage from laser ablation, mount polish issues, electron beam damage during analysis, or the inability to fit good spots on small grains—but also from simply neglecting to measure an abundant element in the apatite. We made no claim that our data were perfect—of course, the preferred scenario is to have 100% elemental totals and ideal stoichiometric apatite—but real data are often messy and complex. All of the information necessary to evaluate EPMA data quality are in the assets included with the paper and low totals are flagged. Low totals or the absolute accuracy of the halogen measurements have little effect on the AFT data interpretations that involve *relative* annealing characteristics. We have other Imperial Fm. examples where elemental totals are ~97–101% with the same data patterns that are present in the LHA003 Imperial Fm. sample in our paper (Figure 1, see below). Suppose we have a sample that contains apatite with 0.0 wt% Cl but elevated concentrations of Fe, Mn, or Sr (elements that clearly enhance retentivity, e.g., Crowley et al., 1990; Carlson et al., 1999; Barbarand et al., 2003; Ravenhurst et al., 2003; Ketcham et al., 2007)—would the Geotrack authors be underestimating the retentivity of their sample by only considering Cl and would it matter for thermal history analysis? The discounting of elements where there is clear empirical evidence for the enhancement of FT retentivity would seem to be a gross oversight.

Duddy stated in his comment: *"We are currently undertaking a research study investigating the usefulness of rmr0 and LAICPMS for AFT U-determination and as part of this study we produced 1057 full EMPA analyses on apatite. Of these analyses, only 4, or <0.4 %, fell outside the range of acceptable totals noted above. This should be the norm for any apatite EMPA study."* Making this statement is meaningless without verifiable proof (i.e., data) and a full description of the microprobe analytical conditions, standards used, accuracy/precision of measurements, etc. We provided our data and analytical information openly, in support of our manuscript that is in a public forum and under consideration for publication in a peer-reviewed journal. We would ask, do Green and Duddy measure only Cl, and if so, what do the probe data and stoichiometry of their apatites look like? We do not know the answers to these questions because no complete data are made available or published.

The Geotrack authors also call into question the LAICPMS method, specifically the $^{238}U/^{43}Ca$ ratio and the reported U (ppm) in our AFT dataset. The mention of these details is an attempt at misdirection and has little bearing on the manuscript being evaluated. The U (or Th, Sm values in ppm) are reported in the supplementary data tables for completeness only—and are a simple ablation pit average—and nothing more, as we have previously mentioned in email correspondence with Duddy and Green and which they recounted in their comment CC3. As Duddy states *"These values are not equivalent"* —we know that. U (ppm) is not used for age calculation or data interpretation of any kind. Duddy discusses deviation of U and U/Ca values from a 1:1 line and shows plots of this in his Fig. 4. The fact that the U (ppm) and U/Ca value do not fall on a 1:1 line *could* indicate the presence of U zoning—or it could simply mean that the U (ppm) value averaged across the ablation-pit depth is not representative of the down-pit weighted U/Ca value that is used for age determination (e.g., hitting small inclusions with the laser)—one reason why the U (ppm) value is not used. We agree with the recommendation by Cogné and Gallagher (2021) that only U/Ca should be assessed for single-grain AFT ages. Effective uranium (i.e., eU) was only utilized in McDannell et al. (2019)

to facilitate comparison with apatite (U–Th)/He data (He reporting convention is U, Th, and Sm, or eU in ppm) and due to the non-intuitive nature of U/Ca values. The use of U/Ca values instead of eU in McDannell et al. (2019) only strengthens the discussed negative age-U trends for most examples. Regardless of these points, their lengthy discussion of eU is both incorrect and irrelevant to the current manuscript. Furthermore, all the required information to calculate AFT ages is available in the manuscript and accompanying (peer-reviewed) documents.

In their numerous publications and paper comments, the Geotrack authors frequently refer to apatite annealing behavior in Otway Basin of Australia, or specifically the Flaxmans-1 borehole (see comments and figures presented in CC1 and CC3 supplements). While we acknowledge that the data from Otway Basin were, and remain, integral results that helped to establish many of the empirical relationships between track annealing and apatite composition—in no way does this mean that the Otway Basin relationships are the universal rule for kinetic behavior and therefore should be applied to all natural apatites. On the contrary, applying a kinetic model based on a single locality or limited data (while better than nothing) could potentially be a hazardous assumption for assessing other apatites that do not adhere to the expected model. We could easily foresee said results being cast aside and considered "outliers." The Carlson et al. (1999), Barbarand et al. (2003), and Ketcham et al. (2007), work should be commended for considering a broad range of apatite species that are undoubtedly encountered in nature. The need for geological calibration is necessary to make reliable thermal history predictions, but those predictions are only as robust as the calibration data.

While the Geotrack authors may rely on their "experience" and "beliefs," in what can only be interpreted as attempts to impede any developments or advances in thermochronological research, we hold no such views and prefer to evaluate work based on data and modelling approaches that are available for public examination and peer review. To disregard this and proceed otherwise is not adhering to the rigours of the established scientific process, it is simply the manufacture of incensed prose disguised as objective criticism.

—Dale Issler, Kalin McDannell, Paul O'Sullivan, and Larry Lane

**References**

Barbarand, J., Carter, A., Wood, I. and Hurford, T.: Compositional and structural control of fission-track annealing in apatite, Chem. Geol., 198(1–2), 107–137, doi:10.1016/S0009-2541(02)00424-2, 2003.

Carlson, W. D., Donelick, R. A. and Ketcham, R. A.: Variability of apatite fission-track annealing kinetics: I. Experimental results, Am. Mineral., 84(9), 1213–1223, doi:10.2138/am-1999-0901, 1999.

Cogné, N. and Gallagher, K.: Some comments on the effect of uranium zonation on fission track dating by LA-ICP-MS, Chem. Geol., 573, 120226, doi:10.1016/j.chemgeo.2021.120226, 2021.

Crowley, K. D., Cameron, M. and McPherson, B. J.: Annealing of etchable fission-track damage in F-, OH-, Cl- and Sr-apatite. 1. Systematics and preliminary interpretations, Nucl. Tracks Radiat. Meas., 17(3), 409–410, doi:10.1016/1359-0189(90)90066-7, 1990.

Duddy, I.: Community comment on "Simulating sedimentary burial cycles – Part 2: Elemental-based multikinetic apatite fission-track interpretation and modelling techniques illustrated using examples from northern Yukon" by Dale R. Issler et al., Geochronology Discuss., https://doi.org/10.5194/gchron-2021-22-CC3, 2021

Galbraith, R. F. and Green, P. F.: Estimating the component ages in a finite mixture, Int. J. Radiat. Appl. Instrumentation. Part D. Nucl. Tracks Radiat. Meas., 17(3), 197–206, doi:10.1016/1359-0189(90)90035-V, 1990.

Green, P. F.: Community comment on "Simulating sedimentary burial cycles – Part 2: Elemental-based multikinetic apatite fission-track interpretation and modelling techniques illustrated using examples from northern Yukon" by Dale R. Issler et al., Geochronology Discuss., https://doi.org/10.5194/gchron-2021-22-CC1, 2021

Green, P. F., Duddy, I. R. and Hegarty, K. A.: Comment on "Compositional and structural control of fission track annealing in apatite" by J. Barbarand, A. Carter, I. Wood and A.J. Hurford, Chemical Geology, 198 (2003) 107-137, Chem. Geol., 214(3–4), 351–358, doi:10.1016/j.chemgeo.2004.10.010, 2005.

Green, P. F., and Duddy, I. R.: Thermal history reconstruction in sedimentary basins using apatite fission-track analysis and related techniques. Analyzing the thermal history of sedimentary basins: Methods and case studies, 103, 65-104, 2012.

Issler, D. R., McDannell, K. T., O'Sullivan, P. B. and Lane, L. S.: Simulating sedimentary burial cycles – Part 2: Elemental-based multikinetic apatite fission-track interpretation and modelling techniques illustrated using examples from northern Yukon, Geochronol. Discuss., 1–37, doi:10.5194/gchron-2021-22, 2021.

Ketcham, R. A.: Referee comment on "Simulating sedimentary burial cycles – Part 2: Elemental-based multikinetic apatite fission-track interpretation and modelling techniques illustrated using examples from northern Yukon" by Dale R. Issler et al., Geochronology Discuss., https://doi.org/10.5194/gchron-2021-22-RC1, 2021.

Ketcham, R. A., Carter, A., Donelick, R. A., Barbarand, J. and Hurford, A. J.: Improved modeling of fission-track annealing in apatite, Am. Mineral., 92(5–6), 799–810, doi:10.2138/am.2007.2281, 2007.

McDannell, K. T., Issler, D. R. and O'Sullivan, P. B.: Radiation-enhanced fission track annealing revisited and consequences for apatite thermochronometry, Geochim. Cosmochim. Acta, 252, 213–239, doi:10.1016/j.gca.2019.03.006, 2019.

Ravenhurst, C. E., Roden-Tice, M. K. and Miller, D. S.: Thermal annealing of fission tracks in fluorapatite, chlorapatite, manganoanapatite, and Durango apatite: Experimental results, Can. J. Earth Sci., 40(7), 995–1007, doi:10.1139/e03-032, 2003.

[Figure]

[Figure]

[Figure]

**Figure 1:** Imperial Formation multikinetic AFT sample, same unit as sample LHA003 in the gchron-2021-22 manuscript from northern Canada. **(A)** Radial plot showing two mixture model age peaks. **(B)** Multikinetic interpretation using $r_{mr0}$ or eCl (apfu), where the two populations agree with model age peaks. Only the pooled age is shown for pop. 2 because of low age dispersion (4%). Symbol overlays show EPMA wt % totals. All pop. 1 totals are 100% unless otherwise noted. Population 2 shows some variation between ~95–100%. There is no clear relationship between probe grain total wt % and eCl. **(C)** multikinetic populations using Cl only, showing complete population overlap. Q = $X^2$ probability.

---

## Author Comment (AC1)

**Response to Reviewer 1 (Richard Ketcham) Comment on gchron-2021-22**

We thank Richard Ketcham very much for his constructive review. His comments help to improve readability and clarify points that we are trying to make in the text. His questions are insightful and they provide us with the opportunity to elaborate on some important topics that are relevant to the research but hard to deal with in this focused contribution. Although the questions may seem straightforward, the responses are more complicated and must take into account multiple factors. Any additional information that encourages readers to try using a multikinetic approach is most welcome. We reply (blue text) to Richard's comments (black text) below.

Reviewer 1: Richard Ketcham

Multi-kinetic effects in AFT thermochronology have long been neglected by much of the community, I gather in large part because it entails more trouble and expense to acquire sufficient compositional data, and the rewards are unclear, especially since thermal history inversion software will often produce a result without it. Hopefully this paper, and others from this group, will bend the curve.

We understand the reluctance of much of the thermochronology community to acquire elemental data if they are uncertain that the extra cost will yield information that significantly improves data interpretation and modelling. Apatite chemistry, provenance, and thermal history interact in complex ways that result in natural samples that can have a bewildering range of characteristics. We believe that the method we present to define AFT kinetic populations is an improvement over conventional methods but do not claim that it will work for samples that do not exhibit clear multikinetic behaviour. For example, insufficient sample heating may mean that mixed provenance signatures are dominant and therefore incompatible thermal history information is retained within different components of a sample (this was discussed in paper 1, McDannell and Issler 2021, GChron v. 3, 321-335). However, the interpretation of partially annealed AFT samples of variable provenance may still be enhanced with multi-elemental data.

Our observations demonstrate that the rewards of collecting elemental data are worth the extra time and cost because we can better understand the cause of large age dispersion in sedimentary samples that experienced sufficient heating and long residence times in the partial annealing zone and use this information to extract valuable details concerning the thermal history. This improved understanding gives us increased confidence in our model results and new insights into the thermal history of our study areas. Furthermore, one compositionally diverse multikinetic sample has the potential to yield substantially more information than single 'monokinetic' samples, and therefore fewer samples may be needed to address specific scientific questions, which may lower the overall costs of some projects. It should be mentioned that elemental data can be beneficial even in the case of a single AFT age population. We have examples where single populations have high track retentivity based on their elemental composition. Without such data one may assume that they are F-apatite and model them that way. Why does this matter? The absence of elemental data can influence some interpretations. As an example, we have an AFT sample from a thrust sheet in western Canada that has long track lengths and an age somewhat younger than its stratigraphic age. This could be interpreted easily as evidence for thermal resetting and rapid cooling related to thrusting. That would not be correct because the sample is track retentive due to elevated Fe concentration (important for increasing retentivity in the Carlson et al., 1999 dataset) and it is best interpreted as a

volcanically-derived sample that underwent minimal annealing following deposition. Cretaceous volcanism is well documented in the Canadian Cordillera and we have numerous examples of minimally annealed retentive volcanic apatites in Cretaceous rocks of western and northern Canada. One advantage of the LA-ICPMS AFT method is that you can obtain apatite U-Pb ages to check for potential volcanic sources.

At the same time, to be effective in doing so (or at least transparent in trying), it would be good to better document the costs.

We are happy to discuss costs in our reply to the review but we don't think that costs should appear explicitly in the paper for the reasons given below. Certainly costs are one of many factors influencing a research program. However, prices are ephemeral and variable depending on who is doing the work and this not something that is normally included in a scientific paper. The purpose of our paper is to present a different approach to interpreting and modelling AFT data. We believe that the focus should be on the results achieved and not on the added cost of obtaining more data to constrain interpretations. Ultimately it is up to individual scientists to decide how they will allocate their resources. In our case, the benefits of obtaining elemental data are very clear and the adage that you get what you pay for is very applicable. EPMA adds a small premium to the cost but the added value is immeasurable because it enables a reliable differentiation of kinetic populations that may not be possible using conventional methods (Cl, Dpar).

It is not straightforward to predict total lab costs because these will vary among different labs. For example, we are dealing with two laboratories in the United States and costs will be influenced by the Canadian-US exchange rate. There can be separate rates for academic/collaborative research versus commercial work. Some university labs charge significant overhead rates for EPMA which can amplify costs and be a factor in deciding which lab to use. LA-ICPMS AFT analysis can be cheaper and certainly faster than EDM (depending on laboratory) because it avoids sample irradiation, cool down waiting periods, and extra counting of induced tracks. As a result, more grains can be analysed which is necessary for better characterization of different multikinetic populations. In general, AFT costs per sample are fixed, whereas EPMA costs vary with the number of grains and elements being analysed. This will also depend on apatite yield and the number of grains with AFT data that are suitable for probing. Currently we use a standard set of 14 elements that have been observed in variable abundance in Phanerozoic samples from western and northern Canada.

We have a good arrangement between GeoSep Services for LA-ICPMS AFT analysis and the Peter Hooper GeoAnalytical lab at Washington State University for EPMA. Lab costs are reasonable and both labs coordinate their activities. The cost of single-spot elemental analysis per grain is reduced as the number of analyses increases and can vary between US$3.50 (large batch) to US$5.50 (small batch). These costs may be higher if using some other labs that are less specialized for this process than WSU (GeoSep services provides a very efficient framework for selecting points for analysis). For our last contract with WSU, we had 1123 elemental analyses for 18 samples which took 6 days and cost US$3960, coming close to $3.50 per grain with an average cost of $220 per sample. The price of elemental analysis for each sample varies depending apatite recovery. On average, elemental analysis increases costs by approximately 20% relative to just LA-ICPMS AFT, Dpar and apatite U-Pb ages analysis. We believe this cost is easily justified for the complicated, chemically heterogeneous AFT samples we are working with.

For example, how long does the EMPA protocol take per spot?

We consulted with microprobe analyst colleagues, Dr. Owen Neill (formerly of WSU and now Manager, R.B. Mitchell Electron Microbeam Analysis Lab at University of Michigan) and Dr. Scott Boroughs, Peter Hooper Geoanalytical Lab, WSU) and obtained the following information. Probe routines can vary between labs - apatite can be a bit tricky to measure and different labs may optimize for different things, i.e. some labs might optimize for high-accuracy/high-precision F and Cl without bothering with the other trace elements, some labs might optimize for REE's at the cost of some accuracy/precision with the halogens, some might optimize for sulfur, and some labs may have different routines optimized for different things that they alternate between depending on the customer. So going to another lab besides WSU might mean that the probe procedure ends up being entirely different (it might not, but that's something that would be handled on a case-by-case basis). The raw analysis time for our samples at WSU is 3.7 minutes per spot (~16 grains per hour), but that doesn't include setup (1-2 hours), standardization (~8 hours no matter how many grains), and point picking (~150 grains per hour). Another complicating factor is that WSU bills by the hour ($55), but caps at 12 hours a day ($660). The GSC has almost always been able to take advantage of this, with large efficient batches that can be run for 48-72 hours straight, effectively cutting the hourly rate by 30-40%.

What considerations went into decision(s) of whether to do compositional analysis first versus laser ablation? Does doing laser ablation first save time, by figuring out which grains work and providing evidence of whether there is kinetic dispersion, and does this outweigh the disadvantage of not getting the analysis precisely where the tracks were measured?

We thank the reviewer for asking these questions. We present our current approach to doing the analysis but variations on this method can yield similar information and have other advantages. Our process considers and tries to balance: (1) efficiency and speed of analysis, (2) the need to maximize the amount of track length information, (3) minimizing selection bias for age grains, and (4) the desire to obtain replicate elemental data, all as part of routine sample analysis. Samples can be processed faster if all AFT analyses are done first before samples are sent to WSU for elemental analysis. Basically this is an extension of standard methods of AFT acquisition without elemental data. It avoids delays related to transmitting samples back and forth between labs and any delays related to conflicting lab schedules. In principle, doing AFT analyses first could also influence some researchers' decisions on whether to proceed with elemental analysis. For example, if AFT results are consistent with a single age population, some researchers may opt not to do EPMA. In our case, we proceed with elemental analyses for all our samples based on experience. The majority of our samples contain chemically variable detrital apatite with multiple age populations so AFT analysis followed by EPMA is part of our routine work flow. It should be pointed out that elemental data can be useful even for samples with single age populations, especially if track retentivity is much higher than expected for a F-apatite composition (see above discussion on volcanic apatite).

We believe that our procedures work well for many samples. Cf-irradiation is used to increase the number of length measurements in order to reduce the permissible range of thermal solutions that will fit the data during modelling. Length measurements are obtained from apatite grains with and without age information (some grains are suitable for length measurement but not age measurement) to maximize the amount of data. If the number of track lengths is low, then the sample has lower resolution and it becomes difficult to differentiate between simple and more complicated thermal solutions. Abundant length data for our LHA003 sample requires

three heating/cooling events to fit the data (this is observed in other Paleozoic samples in the area) but other samples with far less length data allow for simpler solutions with two cycles. An important issue is bias in age grain selection. LA-ICPMS AFT analysis allows for small grains to be analysed and this is important for proper characterization of single grain age distributions. Generally larger grain sizes may be preferred for EDM work but this type of preference may bias observations, resulting in poor representation of certain apatites or the failure to identify different kinetic populations within a sample. This is especially important when length measurements are not associated with age grains and must be assigned to a population using elemental data and/or other information such as Dpar. Our results from replicate elemental analyses on grains having age and length data indicate that zoning does not appear to be a major problem for many of the Phanerozoic samples we have analysed from northern and western Canada. Results indicate that zoning may contribute to some outlier grains that plot in the "wrong" region of kinetic space based on their age. We do not consider this a serious issue because the amount of kinetic population overlap is substantially reduced when using $r_{mr0}$/eCl in comparison to Dpar or Cl. We have samples that can show no obvious population overlap or that have some grains that cross kinetic population boundaries within the expected ±0.03 apfu uncertainty range. We have other data from crystalline basement samples with two probe spots per grain, one near the laser ablation pit and the other from a different grain location. These results suggest compositional zoning is minor, but in those few instances, does produce some variation in $r_{mr0}$ values.

Experienced microprobe analysts tell us that acceptable elemental totals are in the range, 97–100 wt %. We are dealing with detrital apatite grains of variable size and, for some samples, we can obtain lower than ideal elemental weight % oxide totals, generally for some of the smaller grains. The reasons for this are variable and sample-dependent (see reply to I. Duddy comment). Elements that were not analysed and the size and physical state of the grain likely play a role. While acquiring data for a variety of Phanerozoic samples from various regions of Canada over a number of years, we continued to refine data acquisition, interpretation methods and modelling procedures based on what we learned. We analysed various suites of elements and some elements were dropped because they occurred in extremely low abundance or were not present. For example, we discovered that elements such as S and Si could occur in high abundance for some samples but these elements were not included in all the analyses because they do not regularly occur in significant abundance nor do they appear as variables in the $r_{mr0}$ model calibration. We do not include these elements in the 'other' category of the $r_{mr0}$ calculation, unless an element was specifically discussed in the original papers. If missing elements contribute to lower totals then the effect should be to slightly increase apfu values for low abundance elements relative to calculations based on ideal elemental totals. The effect would be to slightly increase eCl values relative to those for an ideal wt % total. At some point you need to settle on a suite of the most common elements that have been encountered and this may not include all possible situations.

Another likely cause of lower elemental totals is a reduction in the EPMA measurement area due to imperfections on the polished mineral surface that are related to laser ablation and multiple etching treatments to reveal spontaneous and Cf tracks. Lower elemental totals generally occur with very small grains. Fortunately, a comparison of results for grains with ideal and less than ideal elemental totals shows that similar elemental proportions and $r_{mr0}$ values are obtained for both cases. Therefore, we believe that both sets of data are useful for qualitative binning of apatite grains into different statistical kinetic populations. Due to the statistical nature

of the problem and the uncertainty in absolute kinetics for retentive apatite grains, a pragmatic approach is to use as much of the data as you can to qualitatively assign grains to different kinetic populations and employ the relative annealing approach for modelling that we describe in the paper. We do not believe that avoiding measuring small grains (age bias) or rejecting "less-than-perfect" data in a multi-parameter data set is a better approach. Below we discuss alternative procedures that may improve elemental analyses for small grains. Although less common for our samples, high elemental totals can occur also. This is usually a sign that F/Cl ingrowth or excess oxygen from halogens are not properly dealt with. When analyses show excess halogens outside the limits of apatite stoichiometry, it is not possible to estimate OH content.

Are there cases where changing the order would be a good idea?

Yes. We discuss the procedures we have been using to acquire multikinetic data but the method can be adapted and customized to fit the requirements and preferences of individual labs. There are several factors that can influence whether or not one can obtain a representative elemental analysis. For example, if many of the grains have significant compositional zoning then it would be better to obtain probe data at the point where the age is measured. We have observed cases where replicate analyses with reasonable elemental wt % totals can yield eCl values that vary by > 0.2 apfu. This extreme variation within single grains is uncommon in our samples (usually only in one or two grains in a small number of samples) but has been observed for both EDM and LA-ICP-MS AFT samples that were analysed using different laboratories. There will always be trade-offs because no two multikinetic samples are alike. Generally, you won't know if zoning is a problem in advance of the elemental analysis.

A more common issue than zoning for our samples is related to multiple treatments on the grain mount that can reduce the suitability of smaller grains for probing (see comments above). In some cases, this could be mitigated by using narrower beam widths but it may also mean that an imperfect surface is measured or that no measurement is possible. These conditions may contribute to some reduced elemental totals. In our opinion, the added track length information from Cf-irradiation outweighs the potential lack of probe data for some grains which was the problem for some of the smaller grains in the Permian sample presented in the paper. Choosing to laser the grains as the last step in the process could allow for elemental measurement in the area where spontaneous tracks are counted. Doing elemental analysis prior to Cf-irradiation is not feasible and not cost effective when acquiring lengths off grains that are not used for age analysis. In this case, the results of Cf-irradiation are needed for selecting grains for track length measurement and a second round of elemental analyses would be necessary, adding more steps and reducing the benefits of running samples for EPMA in large batches.

Another beneficial change that could mitigate some of the uncertainty associated with the above issues is to (1) obtain many of the track lengths from age grains, or (2) only use age grains for length measurement. We have done the former for some samples and this can reduce ambiguity in cases where grains with track length measurements lack probe data, have poor probe data, or where zoning is an issue. At least measurements can be linked to age populations rather than solely relying on chemistry. However, this approach does slow down the AFT analysis so it has not been done routinely. We have found that information from replicate grains with both age and length information and two separate elemental analyses have been very helpful with data interpretation when zonation is an issue. In the latter case of only using age grains for length determination, it is possible to do elemental analysis prior to Cf-irradiation

and laser ablation. The success of this approach depends on obtaining enough lengths to provide well constrained thermal histories.

Ultimately there is a balance between efficiency of data acquisition and the amount of effort involved in data reduction and interpretation. The advantage of measuring lengths only on age grains is that probe costs are reduced because you only need EPMA data for up to typically 40 age grains and you can avoid the extra step of identifying grains with replicate analyses. The disadvantage is that you may obtain less track length information because undated grains or those that may be unsuitable for dating can yield abundant length information. Also relocating each age grain for length measurement is slower than just moving across the mount and measuring lengths for selected grains. Whatever the order of operations, we think that it is a good idea to use Cf-irradiation to increase the number of length measurements when track densities and U concentrations are relatively low in order to improve the resolution of model thermal histories.

[line 44] Justifying the 20°C bound seems to require citing Donelick et al (1990), and optionally Tamer and Ketcham (2020).

Thanks for pointing this out. These references have been inserted into the paper.

[line 243] Change to "thermal history modeling" (or "all model calculations").

The first recommended change has been made to the text.

[line 249] Although replicate values are indeed important for assessing the reproducibility of kinetic parameter values, they may also be taken as an indication of the presence of zoning. The authors do not specify how many spots they took per analysis, but I suspect the answer is one, and that it reflects the usual 2-µm activation zone for EMP; was this driven by the desire for a faster and/or less expensive analysis?

Our investigation of reproducibility of measurements encompasses zoning issues which are discussed in this section and elsewhere in the paper. The cause of divergent measurements may be due to different factors such as poor analyses or zonation. We mention that single spot elemental analyses are used in line 256 in the paragraph immediately below. To further clarify this point, we have modified the sentence on lines 248-250 (new insertion in red font) to be "Replicate elemental (a single EPMA spot per AFT analysis) and Dpar analyses from separate measurements on grains with both age and length data (step 2, Fig. 1) are very important for assessing the reproducibility of kinetic parameter values (Fig. 2)." A single EPMA spot is a very practical choice for minimizing the cost and time for analysis. Our results indicate that it works well for most cases and we feel it is not worth the cost or time to do multiple EPMA measurements to try and eliminate the occasional outlier points that inevitably appear in some data sets. The empirical $r_{mr0}$ model does not incorporate some elements such as S and Si that have been observed in abundance for some samples and thus some scatter can be related to incomplete model calibration. In such cases, additional probing may not help reduce scatter. For the case of difficult samples with lots of zonation, it may be worth redoing the elemental analysis in more detail if it can help improve the interpretation. However, one needs to acquire single spot EPMA data first to determine if more work is required.

Likewise, how many Dpar measurements are averaged for each Dpar determination? The usual procedure is to average four, which ought to make the reproducibility better than observed in Fig. 2c.

The LHA003 sample was analysed at AtoZ Inc. following protocols established by Ray Donelick and these procedures continued to be followed at GeoSep Services. The majority of Dpar measurements were derived by averaging four Dpar measurements per grain analysis. We modified a sentence (lines 152-155, p. 7; new insertion in red font) in section 2.1 to read, "Following standard mineral separation and grain mounting and etching procedures, spontaneous tracks are counted, Dpar is measured for individual apatite age grains (average of four Dpar measurements where possible), and grain x–y coordinates are recorded so that subsequent measurements can be linked to the age grains." Details such as this are also included in the separate GSC open file 8821 that contains the sample AFT and elemental data. Although most Dpar values are based on averaging four measurements it was not always possible to obtain four measurements from every grain. This is particularly true for age measurements where there may not be enough etched fossil tracks to get four values. In contrast, following Cf-irradiation, it is easier to get four values from freshly etched Cf tracks. Therefore, Dpar values associated with age measurements are from spontaneous tracks and Dpar values associated with length measurements are largely from Cf tracks. In principle, this is not supposed to matter because Dpar should depend on the bulk etching properties of the mineral and not on the type of track being used. For the AFT annealing experiments, Dpar measurements were taken from newly formed tracks induced by irradiation.

Also, it's a little unfortunate that the discussion of the downsides of this procedure (lines ~326-340; might not get a compositional analysis near the counting area, or for the grain at all, I gather partly due to the LAICPMS spot) is in the next section; the authors can probably clarify and condense things by briefly mentioning these here, and then referring to them in section 2.3.

On rereading this section of the paper we would prefer to leave it as written unless the Editor prefers that we make this change. It was written this way because we wanted to mention this point in the context of a real example. Single-spot elemental analysis is not a problem for all samples. For example, the LHA003 sample is well behaved in that each grain was probed and it shows minimal population overlap on the age versus eCl plot. Also missing probe data is not an issue for many samples. It happens to be an issue for the P013-12 well sample that has many small grains.

[line 269] Although Dpar imprecision is certainly responsible for a lot of the scatter in Fig. 2e, it's not clear it's the main reason; the authors might try only plotting the points within the 20% bars in Fig. 2c and seeing what the Dpar vs. eDpar scatter looks like. The even scatter might simply be an indication that the things that throw Dpar off are bidirectional; a little OH might increase resistance to annealing compared to no OH (i.e. F-apatite), and a lot of OH might decrease it (e.g., OH-apatite HS from Carlson et al. (1999), but the more OH you have the higher Dpar is.

[Figure 2] Maybe smaller symbols would be better to avoid some of the "solid cloud" effect; some "N =" annotations also would not hurt, and maybe correlation coefficients for d and e.

We have changed the symbols to try and reduce the point saturation. It is tricky to get around this when you are plotting thousands of points at a reduced scale. Given the very large scatter

of the data in Figs. 2c, 2d and 2e, we chose not to use linear regression but instead show how the data are distributed around a 1:1 line. We prefer not to include correlation coefficients because we don't think the relationships would be very meaningful, especially for Fig. 2d where there are strong systematic differences between eCl (a function of many elements) and Cl. Coefficients would change with the addition of more data from apatite of different elemental composition because eCl can vary considerably for a given value of Cl. The plot data are included in the supplement to the paper should someone wish to analyse it in a different way. The number of data points is shown in the title for each plot so adding "N=" would duplicate information that is already there. We have not made this change but if the Editor prefers this alternative we can remove the information from the title and add the notation.

We think that figure 2c and 2e are a fair representation of the uncertainty involving Dpar measurements when applied to natural samples and, from an applications viewpoint, it is unclear to us what can be gained by focusing on a subset of the data with less scatter. When dealing with natural samples, one does not have the luxury of only choosing data that appears to be "well-behaved." Nevertheless, we made some plots below of measured Dpar versus eDpar for various combinations of the grains with replicate analyses to illustrate our point. For some samples, Dpar may show low variability on a plot like Figure 2c, but there are two separate issues here. One is the ability to accurately repeat measurements on the same grain (Fig. 2c) and the other is to be able to resolve kinetic populations with these measurements (Figs. 4 and 5). One might expect reasonable reproducibility of Dpar measurements for the single crystals with induced tracks that have been used in annealing experiments. However, it would appear that Dpar measurements for detrital apatite grains show considerably larger uncertainty.

Figure 2c shows replicate Dpar analyses on the same apatite grain by the same analyst. It is difficult to separate out all the factors that may influence Dpar measurements such as grain chemistry, grain/track orientation, use of variably annealed spontaneous tracks versus fresh tracks formed by irradiation, the number of measured tracks for obtaining an average Dpar value, and analytical issues (etching, measurement, etc.). In principle, one might expect to see much better reproducibility in the measurements if the analyst revisited each mount and measured the same etch pits. However, in our case, the etch pits associated with the age measurements are from spontaneous tracks and those associated with length measurements are from Cf tracks. If much of the scatter is related to differences in etching between fossil and freshly formed tracks then this should have implications for using Dpar as a kinetic parameter because it is supposed to depend on mineral solubility and not the origin of the track being measured. We have noticed that plots of Dpar versus eCl show higher correlation coefficients for Cf tracks than for spontaneous tracks for sedimentary samples from the Mackenzie Delta region of northern Canada. If etching behaviour is the same for both types of tracks then other factors may be contributing to variability in Dpar measurements between age grain and length grain measurements. Under these conditions, four measurements may be insufficient to produce a representative average value. Although it is reasonable that compositional variation between grains within a sample can cause variation in Dpar values, it is less clear how variation in composition within individual grains with replicate analyses can lead to the large data scatter in Fig. 2c.

The two plots below are for Dpar measurements taken from age grains with replicate data that were used in Fig. 2c. The eDpar values were converted from elemental data gathered on the

age grains. The left panel is for grains plotting within the 0.25 μm contours in Fig. 2c and the right panel is for all the age grains in Fig. 2c that have elemental data.

[Figure]

[Figure]

Next we show the same type of plots but for Dpar values associated with length measurements taken after Cf-irradiation. The eDpar values are from elemental data gathered during probing of grains with length measurements.

[Figure]

[Figure]

We can see that both the age Dpar values (spontaneous tracks) and length Dpar values (Cf tracks) show very similar results (left panels above) for the grains with replicate Dpar analyses that are within ±0.25 μm in Fig. 2c. However, both the age and length data show significant scatter with respect to eDpar. This scatter is consistent with our observation that $r_{mr0}$/eCl/eDpar are much better at resolving kinetic populations than Dpar. The main problem is that many samples have a rather narrow range of Dpar values (~1.0 to 1.5 μm spread). Therefore population overlap is inevitable if uncertainties in measured Dpar are on the order of 0.25 to 0.5 μm. Measurement precision is simply too low to clearly resolve kinetic populations. The left panels are the optimistic case for the better fitting subset of data. Expanding the data to all the replicate samples (right panels) or to the thousands of analyses in Fig. 2e greatly increases the scatter and uncertainty in measured Dpar relative to eDpar.

Regardless of the cause of the scatter, we maintain that Dpar is a low resolution parameter in general. We have a small number samples where Dpar resolves different age populations but this is not the norm. Similarly, we have samples where measured Cl can resolve kinetic populations. For these simple cases, either Cl is the only significant element or Cl covaries

systematically with the other elements which makes it appear as if Cl is the only element controlling annealing. If these conditions are not present, then populations overlap with respect to measured Cl. Obviously the mentioned Cl cases will go unnoticed or there will be a level of 'positive reinforcement' regarding the importance of Cl, if only Cl is measured or considered (see Green and Duddy CC1 and CC3 comments). The whole reason for exploring $r_{mr0}$ in the first place was because a significant amount of multikinetic data from northern Canada could not be interpreted using Dpar or measured Cl.

[line 291] "colour-coded"

Quotations have been added.

[line 295] It may be worth noting that compositional populations may also be good candidates for shared inheritance. Although eCl is one such possibility, insofar as it combines a number of compositional variables into one number, apatites with similar eCl may get there via different compositional components, and thus not constitute a good candidate for shared inheritance. This is discussed further below.

This is an important point but a somewhat complicated and sample-dependent issue that we don't think should be discussed at this point in the paper because it doesn't apply to these samples which behave as kinetic populations without variable provenance signatures. We do touch on this aspect in the discussion. You can't properly assess the role of provenance until after you undertake thermal history modelling. If you can model all kinetic populations together using the same history then variable provenance is not an issue that will significantly affect model results. Either the sample thermal history has erased much of the provenance record or differences in provenance are too small to worry about. We are trying to keep the discussion focused and relevant to the data at hand. We have seen many different multikinetic samples and there are lots of different situations that could be discussed but we want to avoid going off on tangents. Our view is that AFT kinetic populations are dynamic and dependent on mineral chemistry and thermal history. Differential annealing and thermal resetting could homogenize formerly different provenance groups that share similar kinetics. Independent thermal maturation data show that this has happened for a large number of our samples. The elemental data we use to define the kinetic behaviour may not correlate with groups based on provenance information provided by other parameters such as U-Pb ages, REEs or other unique signatures. The relation between kinetic populations and populations based on provenance signatures is an interesting topic for future research as more detailed multi-elemental and age data are acquired. Some provenance information may be preserved throughout the thermal history but kinetic populations may change.

[line 333-337] Maybe here or elsewhere, discuss the choice between switching which bin a grain is in, versus leaving the grain out altogether.

Usually only a small percentage of AFT ages appear as outliers in kinetic space unless a sample shows significant elemental zonation. As we mentioned in the paper, our view is that age data are preferred over eCl data if there is a conflict—because laser spots coincide with track count areas, and EPMA spots depend on finding a 'clean' spot surface to get a good measurement. If ages have relatively high precision and plot well within one of the kinetic populations defined on the radial plot then we rely on the age information to determine which population it belongs to. Under these circumstances, age dispersion can be substantially reduced in one population by moving the grain to the other population while not significantly

affecting the age of the population to which the grain was reassigned. We believe this inference is easily justified for grains with replicate analyses where elemental analyses associated with the age cause the grain to be an outlier in kinetic space whereas the elemental analysis data associated with the length measurement for the same grain moves it to a population with similar ages. Other data can be used to help with interpretations. For example, we have some samples where U-Pb ages are distinct enough to identify a volcanic component as being separate from a detrital component and this can help if the volcanic component constitutes a separate kinetic population. Volcanic apatites can have a distinct and sometimes unusual chemistry that makes them stand out. It seems reasonable to use as much of the data as possible because age grains can have associated length information.

[line 438] The claim that population 3 has retained tracks from 540 Ma, or from about 245°C (Figure 6) is eye-catching, and probably overly optimistic about the ability of AFT to retain information about such high temperatures. It appears to stem from a difference in how AFTINV evaluates total annealing versus HeFTy's "oldest track". HeFTy assumes total annealing after reduced mean length falls below 0.4095 for non-projected lengths, corresponding to a mean length of just under 7 μm, whereas AFTINV appears to have total annealing correspond to a mean track length of 2 μm (line 419). This may be based on a slight misinterpretation of what's written in Ketcham et al. (2000); the 2 μm limit mentioned there corresponds to the smallest track that can appear in a track length distribution. However, such occurrences are due to including a population of tracks with a higher mean and large standard deviation. The 0.4095 value arises in part from the observation that no annealing experiments reported by Green (1988) or Carlson et al. (1999) had a mean length below 7 μm (although there are some 6's and 5's reported by Barbarand et al. (2003), and even an occasional 4 or 3 by their Analyst 3). Willett (1997) uses a similar value of 0.428 as the zero-density intercept of reduced length versus density reported by Green (1988). In other words, by the time a mean length falls below some limit, the track population becomes undetectable. I believe this provides a more realistic basis for evaluating total annealing and the oldest retained track. Using the revised criterion, the TA for the oldest track for an rmr0=0.491 apatite is closer to 200°C, which seems a lot more reasonable considering the closure temperature is 161°C. This is not the most crucial of issues, but it's prudent to avoid distracting claims.

This is a fair point to make. We realize that our choice of shorter track length for modelling retention ages gives an uppermost maximum limit on temperatures for track retention and we agree with the reviewer that we should not put too much credence in such theoretically-derived temperatures. This value was chosen because it represents the shortest track length ever measured by Ray Donelick out of many tens of thousands of analyses. We have some samples with measured track lengths between 2 and 4 μm but these are rare. The reduced mean length of totally annealed tracks is easy to change for the AFTINV model but doing so won't change the concepts illustrated by the theoretical retention ages described in the paper. We will modify the abstract to remove references to specific model-dependent annealing temperatures and update the text to better explain this point. From the reviewer's comments we can see that this value is not a precise parameter and it depends on our ability to observe and measure short tracks. As discussed by Ketcham et al. (1999), the concept of annealing temperature depends on how it is defined and it is influenced by factors such as annealing kinetics and the heating and cooling rates. However, the concept is still useful for estimating when track information may be retained in a sample. The retention age calculations still show how various populations have been annealed and they give an approximate oldest possible time limit from which AFT

populations can start to retain thermal history information. This is important for our samples because it shows that provenance information has been erased for some of the kinetic populations and therefore it is possible to model all the data using the same thermal history.

[Figure 6, 7] I appreciate the authors' efforts to incorporate the CRS method into AFTINV, and intrigued by the result – it looks to be a powerful addition. I have long been considering doing something similar myself, having dropped the CRS method when I converted my earlier program AFTSolve to HeFTy. However, one of the reasons I did so may still be evident in the model results here. The CRS method has a tendency to quickly converge to a relatively smooth solution that does not explore the solution space as well as the Monte Carlo method, and thus map out the range of solutions that fit well. In HeFTy results, this allows the resolving power of the data to be evaluated by looking at the width of the solution envelopes.

We thank the reviewer for drawing attention to this important point on clarifying the meaning of the thermal history results and the use of the CRS algorithm. The algorithm uses an expansion factor, alpha, that controls how aggressively the algorithm searches for new solutions by randomly recombining members of a fixed set of model histories to generate a new trial history. Earlier versions of AFTINV included a different implementation of the CRS algorithm but we stopped using it for the same reasons given by the reviewer. A critical consideration is that, previously, a constant alpha was used and the initial starting pool of random solutions were not acceptable solutions. Therefore, the starting point was far from acceptable solution space, it could take a long time to find solutions, and the algorithm could get stuck in a minimum and generate too narrow a region of acceptable solution space.

Two important changes were made. First, the initial pool of solutions is generated using the random Monte Carlo method and it converges when all solutions exceed the 0.05 significance level. The CRS algorithm is now able to draw from a set of acceptable 0.05 solutions and tries to improve the entire set to a higher significance level. The second change was to cycle through different values of alpha in order to help prevent the algorithm from getting stuck. If no solutions are forthcoming at a selected value of alpha, then alpha ramps up. When a solution is found, alpha drops and the cycle may be repeated. A more aggressive search is beneficial earlier in the calculations but it is not beneficial to stay at a high value of alpha. As the model evolves and generates many more solutions, then the range of alpha values decreases. There is a balance in selecting alpha values and iteration times. The algorithm we settled on is based on testing different schemes using different multikinetic samples to determine which gave the overall best performance in terms of maximizing the number of 0.5 solutions within reasonable model run times.

We believe that this new version of the CRS algorithm is a powerful addition to AFTINV because it enables us to discover the conditions required to obtain close fitting solutions. If variable provenance is not a factor, then multikinetic populations add very powerful constraints on thermal histories due to the requirement to fit the AFT age and length data of all populations with the same history. There are multiple reasons why it can be difficult to obtain answers. It is easy to determine if variable provenance is a factor by modelling individual populations and comparing the thermal histories. If pre-depositional thermal histories do not overlap, then this suggests different provenance information is preserved. For our samples, the main difficulty in finding solutions was to ensure we had the appropriate style of thermal history and compatible kinetic parameters. In the case of our LHA003 sample, three cycles of heating/cooling were required to fit the data. Early models had only two cycles and they were unsuccessful. Also, the

ability to closely fit age and length data depends on having the right separation in kinetic parameter values and this can be estimated using the CRS calculations. The number of 0.5 level CRS solutions is related to the relative differences in eCl values between different kinetic populations so we can determine the range in offset that produces the most close-fitting solutions.

We view the CRS smoothing of thermal solutions as a desirable effect for the temperature ranges that the AFT data are sensitive to, especially since the resulting CRS history pool is derived from the starting pool of histories that already fit at the 0.05 level. Unlike HeFTy and QTQt, AFTINV constructs thermal histories differently and uses many more time points on a fixed, user-specified grid. The Monte Carlo calculations use uniform random deviates to generate new temperature points. This creates more possibilities for small fluctuations and stair-step patterns in the thermal solutions. There is a trade-off between time and temperature so it is easy to overshoot or undershoot temperature due to this compensation effect. These fluctuations are not resolvable by the data and ragged looking thermal histories can yield just as good fits as smooth histories. We prefer the "averaging effects" of combining solutions to yield less extreme heating/cooling rates at the 0.5 level due to data resolution issues. We try to have the best of both worlds here. The 0.05 level Monte Carlo solutions map out a broad range of acceptable solutions and this envelope can be interpreted the same way as it is for the HeFTy model. The CRS solutions are then embedded in this envelope to show close-fitting, smooth solutions at the 0.5 level. The Monte Carlo solutions allow higher rates and therefore upper temperature limits and average temperatures are somewhat higher than the CRS values. For LHA003, thirteen 0.5 solutions generated by the Monte Carlo algorithm are more ragged looking and extend beyond the range of the CRS solutions because of that.

Although the CRS algorithm tends to yield smoother solutions, it can converge on multimodal solutions if they are permitted by the data. For P013-12, there is a minor solution mode with older peaks. The low number of track lengths in population one allows for a broader range of thermal peaks for the second heating/cooling event. The solutions with older Jurassic thermal peaks are associated with Cretaceous thermal peaks for the second heating event. Sample LHA003 also shows more than one solution mode. The number of permitted modes depends on the amount of length data available to constrain solutions and the complexity of the thermal history. We have samples with fewer tracks for which the CRS algorithm yields multimodal solutions. This indicates that the new version of the algorithm is less likely to be trapped in a thermal minimum, since multiple modes at the 0.5 significance level indicate no clear 'preference,' or alternatively a general inability to update the solution set and therefore preferentially select one of the modes as a result. The exponential mean solution generally provides an excellent fit to the data if there is a dominant solution mode. However it may fail if there are different modal peaks.

In the results here, what puzzles me for P013-12 is the relatively tight band of good solutions above 175°C from 600-450 Ma, and probably a fair bit younger/cooler than that. Given the 161°C closure temperature of the most resistant population, the idea that it would exert much constraint in the 175-250°C temperature range seems improbable, and is not reflected in the QTQt results either. This all is not necessarily a problem, but I think it should be discussed so people interpreting these results have a more complete knowledge of what they are looking at.

This is a good point to make. The model temperature at which tracks are retained will be a function of the annealing kinetics defined by eCl and the rate of temperature change and so

there will be a range of model temperatures corresponding to the estimated times of total track annealing. For example, very rapid heating (on the order of 10°C/My) following deposition for the LHA003 samples means that tracks will survive to higher temperatures than for more modest heating rates. This variability in rates and temperatures is reflected in the distribution of model retention ages so it is difficult to assign a single temperature for assumed total annealing. Using a higher value of reduced track length (~ 0.4) to represent total annealing will result in younger retention ages but the same concept applies. The ability of AFT data to constrain thermal histories diminishes as temperatures approach total annealing temperatures and the only way to estimate temperature ranges where AFT data can constrain the thermal history is to examine the model behaviour.

For the Devonian sample, LHA003, the most track retentive population has an eCl value of 0.5 apfu. The exponential mean solution shows a shift to more steady cooling rates at ~480 Ma and ~175°C. The average cooling rate for the exponential mean solution below 175°C is ~1.4°C/My between ~480-380 Ma. Above ~175°C, the cooling rate drops off. For the Permian sample, P013-12, the most retentive population has an eCl value of 0.55 apfu and steady cooling rates are achieved below a temperature of ~185°C at ~440 Ma. The average cooling rate below ~185°C is ~1.2°C/My from 440-295 Ma. Above ~185°C, the cooling rates decrease significantly. We concur with the reviewer that the AFT data do not constrain the thermal history at temperatures > ~175-185°C. Our interpretation is that the change in cooling rate marks the approximate point at which AFT data are able to constrain the temperature history. So why are the CRS temperature envelopes so narrow between 175-185°C and 250°C? This is related to the nature of the calculations. CRS solutions are required converge on a narrow temperature interval of 245-250°C, the same starting condition for the Monte Carlo calculations. These high temperatures were chosen to ensure that all populations were totally annealed at the beginning of the thermal history. Rate and temperature boundary conditions are enforced during CRS calculations to maintain consistency with the Monte Carlo calculations and solutions that fall outside of these ranges are not accepted. This tends to limit the spread of solutions in the region where AFT data provide no constraint. Probably the best way to deal with this is to annotate the figure to indicate where the thermal history is unconstrained by AFT data.

We believe the AFT data constrain temperatures at less than ~175-185°C for the samples used in the study. At these lower temperatures, each population retains some record of the thermal history. The lowest retentivity population is most sensitive to the later, lower temperature portion of the history whereas higher retentivity populations provide more constraints for the earlier higher temperature part. Thermal history resolution is enhanced because preserved tracks in each population contribute information on overlapping portions of the thermal history.

Along similar lines, did both the AFTINV and QTQt models assume that all apatites in each sample had the same inherited, pre-depositional history? If so, was the fact that they did so, and their success in fitting their models, and indication that there was shared provenance, or an indication that, for these samples, results are not terribly sensitive to the pre-depositional history?

We cannot determine from the available data whether each kinetic population has the same shared inheritance because this information has been degraded by thermal annealing. AFTINV modelling was done before QTQt modelling. Individual populations were run first to get an initial impression of the thermal history. For LHA003, the pre-depositional thermal record for the lower retentivity population one was erased completely by thermal annealing. Therefore, the predepositional history reflects information retained in the high retentivity population two and only this population is sensitive to this inherited history. This enabled us to run both populations together in order to enhance the resolution of the post-depositional thermal history which is the main focus of our northern Yukon study. It is notable that LHA003 shows more variability in the pre-depositional cooling history than P013-12 and this may be related to a higher degree of annealing associated with the higher temperatures (>30°C) encountered during the first thermal peak.

For P013-12, the pre-depositional thermal history has been erased for population 1 and model pre-depositional thermal histories overlap for population 2 and 3. Population 2 has also experienced significant annealing so the pre-depositional history is dominated by population 3. Again it is unclear whether there was shared provenance because thermal annealing has obscured the information. It should be noted that our multikinetic scheme can still be useful for interpreting and modelling samples where some populations preserve different pre-depositional histories. For example, we have a sample with four different populations, three of which can be modelled together. The fourth population was much older with a different pre-depositional history and was modelled on its own. The $r_{mr0}$ parameter still allowed us to define separate kinetic populations. Within QTQt, we did allow a variable pre-depositional history during modelling for both samples. This is apparent in the broad t-T envelopes for the Expected model. The results indicate that due to the strong annealing, the pre-depositional history does not greatly influence results, especially for sample LHA003.

Or, are the results sensitive – do the few earlier-cooling 0.5 paths for P013-12 corresponds to the earlier peaks T's at ~195 Ma and/or ~70 Ma?

The four earlier 0.5 level cooling paths correspond to a cluster of solutions with temperature peaks in the range, 141-144°C at a model time of 172.5 Ma. This is a solution mode determined by the CRS algorithm but it doesn't seem that there is much sensitivity to the pre-depositional cooling in terms of affecting later thermal events. The two outlier paths with temperatures in the range, 145-148°C at 192.5 Ma have associated thermal peaks at ~70 Ma but their pre-depositional cooling paths are within the region defined by the majority of CRS cooling paths. Therefore, for this sample, a pre-depositional history can be resolved but it has little impact subsequent thermal events.

The manual (AFTINV) and automatic (QTQt) raising of the rmr0 values for the most resistant populations in each sample is interesting. What seems to be going on is that the different populations need greater separation in their partial annealing zones to produce their respective divergent age and length distributions. It's further interesting that the higher resistance is corroborated by the vitrinite data for sample LHA003, though less so for P013-12. The authors recommended approach of "anchoring" on low-resistance kinetic seems like a good one. Another possible "advantage" of the Ketcham et al. (1999) model over the (2007) one beyond the different rmr0 equation is that it has a much higher temperature range, which these results may imply is necessary to create these divergent populations.

Yes we believe that relative annealing is a powerful approach to modelling multikinetic AFT samples when we lack the data to accurately predict kinetic parameters for some of the apatite populations that are likely to be encountered in nature. It takes advantage of the following: (1) the $r_{mr0}$ parameter is calibrated based on the relative annealing behaviour among apatites of different composition in the annealing experiments; (2) observed AFT data for different kinetic

populations within a sample must be accounted for within a shared post-depositional thermal history framework; (3) lower retentivity apatites within a specific compositional range are abundant in nature and are best represented in the annealing experiments and therefore provide a reasonable reference point for estimating kinetic parameters for less well understood compositions; and (4) thermal maturity data provide independent paleotemperature information for assessing kinetic parameter assignments and model thermal history predictions. Organic maturity is consistently high within Paleozoic strata across the northern Yukon study area and compatible with the eCl values required for the higher retentivity AFT kinetic populations. For the P013-12 sample, a single measured maturity value from a Permian cuttings sample is uncertain. It is similar to the overlying Cretaceous section and may be biased to a lower value. The Permian is only ~90 m thick in this well and the underlying Carboniferous units with more measurements have maturity values that are higher by ~0.1 %Ro. The model still fits the Permian measurements within two standard deviations.

Lastly, the comparison between AFTINV and QTQt results appears to gloss over their differences a bit. For P013-12, the first reheating peaks at ~168 Ma in AFTINV and could go as far back as 195 Ma, whereas QTQt appears to strongly say that it was at about 140 Ma. Similarly, AFTINV implies that the first peak reheating for LHA003 was at 345 Ma, compared to 300 Ma for QTQt. If you lay the models pairs on top of each other, they appear to exclude each other at these times. Is this because QTQt calculated different kinetics than the manually-shifted ones in AFTINV, or because of QTQt favoring simpler histories, or some combination of these and possibly other factors?

One of the main reasons that we ran the QTQt model was to show that the general style of thermal history determined for our samples using AFTINV could be reproduced using a different modelling approach. People who are not used to working with multikinetic data may be skeptical of the detailed thermal histories that we obtained for our samples. Prior to modelling the samples, we had a poor understanding of the regional thermal history. Much of the stratigraphic record has been eroded from the study region and simple model boundary conditions based on two major thermal events failed to yield satisfactory solutions for the LHA003 sample. Therefore, modelling investigations were undertaken to discover the conditions required for obtaining successful solutions. The approach was not to force any preconceived notion of the thermal history but to learn from the data. The QTQt results confirm that the multikinetic data retain a record of multiple heating events. AFTINV requires that the number of thermal events be specified by allowing for model inflection points to occur over specific time-temperature ranges. If the events aren't required then inflections will be minor and contribute nothing significant to the thermal history. QTQt was run with wide open boundary conditions and minimal constraints. It converged on a solution with three heating-cooling cycles for LHA003 without that condition being imposed.

AFTINV and QTQt models are constructed differently and calculations are undertaken in a significantly different way. Some differences are:

(1) The statistical methods are different. AFTINV uses a "frequentist" approach where the objective is to find a set of thermal solutions that exceed an absolute goodness-of-fit probability threshold. QTQt use a "Bayesian" approach that yields single model solutions (e.g. maximum likelihood or maximum posterior) from a large model ensemble that is a result of maximizing the probability (i.e., likelihood ratio) as a measure of fit to the data, or the 'acceptance' criterion. Therefore AFTINV (like HeFTy) relies on absolute values of

probability, whereas QTQt relies on the (likelihood) probability ratio. QTQt has a different acceptance criterion and lacks a specific 'threshold' for model path acceptance.

(2) AFTINV uses a grid with many time-temperature points whereas QTQt tends to use only the minimum number of points needed to fit the data at an adequate level (as modelled in our examples). The model adds and subtracts points as needed and tries to avoid too much complexity. Although AFTINV uses fixed time points, it imposes heating and cooling constraints consistent with specified thermal history styles to avoid saw tooth temperature fluctuations in thermal histories. QTQt is more tolerant of wide open boundary conditions, in part because it uses a minimal number of time-temperature points. For AFTINV, boundary conditions need to be more carefully specified. If they are too wide open, the much larger number of model points creates many more possibilities which could cause the model to spend too much time interrogating unpromising regions of solution space.

(3) Initial model conditions can vary. In typical applications of AFTINV for sedimentary samples, model thermal histories are started at high temperature at times much earlier than the depositional age. The goal is to try and model any inherited history as a pre-depositional cooling event. This allows the model to cool below the total annealing temperature when required by the data. If there is no inherited history, then a broad range of cooling curves will be generated. For QTQt applications, boundary conditions can be wide-open and it can generate a simple pre-depositional cooling history in an appropriate region of time-temperature space. Therefore, we specified a 'common' pre-depositional history in AFTINV (after trials suggested this was suitable), whereas we allowed different, albeit grossly simplified, pre-depositional histories for each kinetic population in QTQt (i.e., a single t-T point added prior to the depositional age).

(4) For AFTINV, kinetic parameters can be adjusted manually relative to a population with fixed kinetics to maximize the number of close-fitting solutions. In QTQt, kinetic parameters are adjusted automatically within ranges to improve model fits. The program may stop adjusting the parameters if it decides incremental changes in fitting the data are not significant enough. Thus, with many parameters allowed to vary (higher degrees of freedom), QTQt may provide marginal fits to ages or lengths, but those results are conditional on the input data and model prior assumptions—where in this case we rejected 'more complex' models for equivalent likelihood—which essentially provides a lower limit on the t-T complexity required to fit the data (possibly at the expense of 'perfect' fits to the data).

(5) AFTINV generates a set of acceptable (0.05 level) and good (0.5 level) solutions (typically 300 each). The AFTINV solution set maps out a broad range of possible times and temperatures for thermal peaks. The exponential mean of the CRS solutions generally provides a smooth, good-fitting solution. QTQt generates several types of single t-T solutions based on likelihood maximization and the posterior probability. The 95% credible interval around the expected model for example, summarizes (weighted mean or mode of the posterior peaks) all of the accepted post-burn solutions and often denotes regions of better or worse t-T resolution.

(6) Both models incorporate a relation between initial track length and kinetic parameters but they are different. QTQt uses relations published in Carlson et al. (1999) and AFTINV uses a newer unpublished relation provided by Ray Donelick that contains more data.

(7) QTQt specifies AFT central ages as input but converts LA-ICPMS data to EDM equivalent data. Although central ages are used for plotting model results, Ns and Ni count values are used in the QTQt calculations rather than ages. AFTINV uses central ages when there is significant age dispersion but otherwise uses pooled ages. It can model either EDM or LA-ICPMS AFT data and does not do conversion between these two different types of data.

The above factors can account for why there are differences in detail between the two models. For P013-12, AFTINV provides a broader range of good-fitting (0.5) CRS solutions with peak times extending from 150 Ma to ~195 Ma. The QTQt expected model is closer to this lower age limit but the 95% confidence region overlaps with AFTINV results. It is clear that QTQt does not provide as close a fit as AFTINV to the length data for population three. QTQt converged on eCl = 0.47 apfu for population three whereas AFTINV used 0.55 apfu. The most likely reason for the difference in model results are that AFTINV generates more solutions, uses more model points to construct thermal histories, and that observed ages are different for each model due to the conversion of LA-ICPMS FT data into EDM FT data by QTQt. This results in younger ages for the QTQt model because EDM and LA-ICPMS central ages are not equivalent. The difference between QTQt and AFTINV results for LHA003 is due to a number factors as well. These include that manual fine tuning of kinetic parameters allows AFTINV to more closely fit population two ages and lengths. QTQt converged on an eCl value of 0.44 whereas AFTINV used 0.5 which generated 300 0.5 level solutions. The difference in the age of the first peak is likely related to the difference in kinetics and differences in the pre-depositional cooling history. AFTINV shows cooling from higher temperatures later in the pre-depositional history. QTQt does prefer simpler histories and uses fewer points so that is definitely a factor as well.

---

## Author Comment (AC2)

**Response to Reviewer 2 (Karl Lang) Comment on gchron-2021-22**

We thank Karl Lang for his comments that help us to clarify issues concerning differences between the LA-ICPMS and EDM AFT analysis methods. It is not our intent to suggest that the LA-ICPMS AFT method is superior to the EDM method nor to promote the idea that people should stop using EDM in favour of LA-ICPMS. Indeed, we have plenty of quality EDM data for multikinetic AFT samples and AFTINV can accommodate either type of data. That said, it is equally important to note that the opposite is also true, that the EDM method is by no means superior to the LA-ICPMS approach that we chose to utilize. Based on the many thousands of analyses that our primary analyst has generated by both LA-ICPMS and EDM, it is clear that a properly-trained analyst can produce data of equal quality using either method.

Reviewer 2: Karl Lang

Use of "detrital" was a little confusing to me at first, since many applications of detrital thermochronology are now also focused on interpreting cooling histories of source rocks prior to deposition, and not simply the common cooling history of detrital minerals in a sedimentary rock after deposition. This is a semantic difference, but perhaps adding a sentence to state this explicitly at the beginning of the manuscript might clear up any confusion amongst readers.

We use detrital in the normal geological sense of "pertaining to particles of rock derived from the mechanical breakdown of preexisting rocks by weathering and erosion." We avoid the term, detrital thermochronology, because of the connotation it carries. From the title it should be evident that we are talking about sedimentary basin thermal histories. However, we could change the first sentence in the Abstract to read "…cause for AFT age dispersion in sedimentary samples" rather than "…cause for age dispersion in detrital AFT samples." Elsewhere in the text, detrital is used to describe the nature of the grain and so this should be clearly understood. If the confusion is related to the use of "detrital sample" then this could be replaced with sedimentary sample, for example.

Why does the manuscript include a vigorous preference of LA ICPMS over EDM approach? This seems unrelated to the central motivation of the paper and, in my opinion, is largely unsupported (see comments by line). The authors should explain why they chose to use LA-ICPMS instead of EDM, but they should avoid generalized claims about the relative efficacy of one method over the other (e.g. "The LA-ICP-MS method has some distinct advantages compared with EDM" [117]).

The manuscript does not include a "vigorous preference" of one method of data generation over another. Rather, the comments within the manuscript made in favour of the LA-ICPMS method used here provide exactly the requested "explanation" of why we chose to use LA-ICPMS instead of EDM. EDM is well established and has been around for many decades and its "perceived" advantages are well documented. For instance, as long as they had access to a research reactor and funding to pay for irradiations (heavily subsidized for academic labs), a trained analyst using EDM could create a fission track lab with not much more than an optical microscope and an automated stage. In contrast, the LA-ICPMS method is relatively new and requires access to more expensive and sophisticated equipment than EDM (not including the access to a research reactor). As a result: 1) only a limited number of labs have incorporated this newer approach, which directly measures parent U instead of using a proxy for U, and 2) the LA-ICPMS method is less understood within the thermochronology community.

This lack of familiarity with the LA-ICPMS AFT method has led some to question if the LA-ICPMS method provides data of the same quality (e.g., see community comment on paper). Comparisons of EDM and LA-ICPMS AFT data have shown that the latter method may yield data with more age dispersion. In some cases, $\chi^2$ statistics and age dispersion have been interpreted as data quality indicators to infer that there are analytical problems with LA-ICPMS data such as U zoning. Certainly zoning exists in nature but rigorous and consistent analytical practices can mitigate problems with zoning. If there are differences between EDM and LA-ICPMS AFT data, the tendency is to automatically look for problems in the latter data. However, there are multiple causes of age dispersion and these statistics must be used with caution because it can be misleading to infer data quality on that basis. There are other issues that may contribute to differences between EDM data such as grain selection and counting bias or differences in the precision of U values used for age calculation. High age dispersion is the norm for our multikinetic samples yet age dispersion is significantly reduced when grains are binned into different kinetic populations using elemental data.

In closing, we contend that both methods can yield data of similar quality but they are different methods with different advantages and disadvantages. In the paper, we are just pointing out some relative advantages of the LA-ICPMS method compared to EDM that are pertinent to this particular discussion. This doesn't preclude that EDM has other advantages relative to LA-ICPMS.

118. It has not been my experience that analytical costs are lower for LA-ICPMS than for the EDM when measured on a per grain basis. If you can measure 100 grains per mount and 50 mounts fit in a $1000 irradiation package, that's $5/grain. By comparison, LaserChron (probably cheapest option in US, at least) charges $9-16 per grain for 100 grain samples, not including costs for CL imaging. Also, throughput is not necessarily higher for LA if you have to wait several months for lab time to become available. In my experience the analytical time to produce a complete fission-track dataset is comparable regardless of the analytical approach. I worry that comments like this will gradually discourage scientists from using the EDM, which is a well established and data-rich method.

In terms of the wording within the manuscript, we offer our thanks for pointing this out. Lab costs are highly variable and depend on many factors, some of which are discussed in our reply to reviewer 1. We can say in our situation that overall costs were lower when we switched from using EDM to LA-ICPMS. However, that may not be the case for others so we will omit reference to costs. We maintain that AFT analysis is faster for the LA-ICPMS method versus EDM under normal conditions for the reasons given in the paper. Of course, there will be delays if equipment needs servicing and lab schedules are backed up but the same could be said for EDM if there are issues accessing a reactor and shipping samples. It is a safe bet that access to facilities would have been taken care of prior to any lab instigating the use of the LA-ICPMS approach; either by purchasing the necessary equipment (expensive), or having reached an agreement for access with an existing LA-ICPMS facility. It is an equally safe bet that any lab using the EDM method would have taken care of any questions surrounding access to irradiations.

The review comment above seems to be based on the costing related to the academic pricing for a ZrnUPb DZ analysis provided by LaserChron instead of a direct comparison to LA-ICPMS analysis for AFT (i.e., AFT does not require additional costs for CL imaging). If correct, and if LaserChron does charge between $900-$1,600 for 100 spots ($9-$16/spot), then this is not the

cheapest option in the US as the price we were charged for the use of a laser ablation and mass spec system to complete the AFT grain-age analyses presented in this manuscript was on the order of $2.12/grain. Even if the values presented by the reviewer were pertinent to an AFT analysis using the LA-ICPMS approach, the quoted cost of $5/grain would seem to be incorrect. Fifty mounts of 100 grains each (5,000 grains) should work out to $1,000/(50x100grains) = $0.2/grain. The subsidized $1,000/irradiation for 50 mounts makes it more affordable for users but this does not reflect the true cost of the service. Considering that most AFT labs provide analyses with up to 25 rather than 100 grains per mount, an analyst would produce data from a total of 1,250 age grains per $1,000 spent for each irradiation – at a cost of $0.8/grain or 4x the reviewer's estimate. It is evident that when looking at just analytical costs on a per-grain basis, subsidized irradiations are cheaper. However, simply not having to deal with all the rules and regulations associated with holding a radiation licence more than makes up the difference when evaluating the costs.

137-138. Wouldn't observer bias have a greater impact on age determination when it is only accounted for in spontaneous track counting? It seems to me that observer bias may actually be reduced when it is accounted for in both the spontaneous and induced track counts, rather than just in spontaneous counts. Either way, I don't consider it fair to say that LA-ICPMS is "more objective" if it still relies on user interpretation and collection of spontaneous track data.

Unconscious bias is an established phenomenon in cognitive research. The extent to which it may influence the selection and counting of grains in AFT analysis is unknown, but its existence has been established previously. We believe that it is valid to point out this fact and that it may have an influence on grain selection, especially if only a subset of potential age grains are counted. Under these circumstances, the age data may be of good quality but it may not be capturing the full range of compositional variation within a sample, which is the most important issue for multikinetic samples.

Unlike the LA-ICPMS FT method, all the information for U content and age determination is available at the time of measurement using EDM. The counting of spontaneous and induced tracks relies on the judgement and experience of the analyst to select which areas to count to yield reliable age data. An advantage of this approach is that areas with significant U zonation can be avoided. We accept that good data are obtained for the grains that are counted. However, Ns and Ni data are accumulating as tracks are being counted and the human brain is well adapted to pattern recognition. If multikinetic populations are present but cannot be resolved using the conventional kinetic parameters, Dpar and Cl, then there is not a strong incentive to count a large number of grains and add to the sample age dispersion. In this context, a dominant population may be counted with other grains viewed as outliers.

For the LA-ICPMS method, a measurement area is selected and spontaneous tracks are counted first. Subsequently, laser spot analysis within the predetermined track count area yields a U/Ca ratio for age determination. For our multikinetic samples, we tried to avoid grain selection bias by counting a broad range of grains including lower quality, more difficult to analyse smaller grains. From a multikinetic perspective, it is desirable to count enough grains to sample different age populations if they are present. We believe that EDM and the LA-ICPMS method will yield similar results for multikinetic samples if a large and diverse range of grains are analysed. Conversely, if too few grains are counted, then results will be more sensitive to the grain selection process and both methods may yield apparently different results.

140. This is not an inherent limitation of the EDM, simply a choice by the operator to count fewer grains. Many detrital studies regularly count more than 100 grains per sample with the EDM.

Yes. For thermal history studies involving sedimentary samples, we recommend that a more diverse and larger number of age grains be counted than is typically done for EDM, increasing the probability of higher age dispersion and $\chi^2$ failures. If the age dispersion is related to compositionally-controlled multikinetic annealing then more information will be obtained from a sample. We believe that EDM and LA-ICPMS will yield similar results for multikinetic samples if the goal is to count a large enough number of grains to represent the true age and compositional variation within a sample.

142-143. It is convenient to make this argument here, but one could also make an alternative argument that the induced track print actually allows for more robust data collection because you can avoid the zonation issues you mention to be a problem on line 128-130. I don't understand why this is cast as an example of making the EDM less objective.

We raise the issue because of past concerns regarding higher age dispersion with LA-ICPMS data. We just want to make the point that this is not necessarily a bad thing and that other factors may be contributing to apparent differences between EDM and LA-ICPMS data.

145. Again, this is not an inherent problem with EDM it is a choice by the EDM user.

Yes. Results obtained from both EDM and the LA-ICPMS methods will depend on choices made by the analyst. It is important that a large enough number of grains are analysed to better represent the true age variation within a multikinetic sample. Our point is that higher age dispersion is likely to be encountered using an LA-ICPMS approach that tries to capture variability in single grain ages by analysing a sufficiently large sample size. Many applications of EDM have analysed a smaller number of age grains and this can lead to lower age dispersion. With fewer grains counted and decisions made on what to count after all the analytical data has been gathered, there may be a tendency to choose similar grains, which can result in lower age dispersion for multikinetic samples. The solution is not to abandon EDM but to count more grains with a broader range of characteristics to better represent different age populations.

---

## Author Comment (AC3)

**Author response to community comment CC1 for preprint gchron-2021-22**

Issler, D. R., McDannell, K. T., O'Sullivan, P. B., and Lane, L. S.: Simulating sedimentary burial cycles – Part 2: Elemental-based multikinetic apatite fission-track interpretation and modelling techniques illustrated using examples from northern Yukon, Geochronology Discuss. [preprint], https://doi.org/10.5194/gchron-2021-22, in review, 2021.

Green and Duddy criticize reviewer 1 for what they perceive as his downplaying of the need for compositional data for AFT studies and use this as a segue to promote their Cl-based approach to interpreting and modelling AFT data that they developed and applied as consultants at Geotrack International. Their promotion of the Cl model is an integral part of their critique of our paper and therefore we are compelled to respond. They express frustration that apatite composition is not routinely included in AFT studies but are then highly critical (community comment CC3) when an alternate method that incorporates more detailed compositional data than they use is presented. The Green and Duddy Cl model rests on two key assertions. *(1) wt% Cl is the only important parameter controlling AFT annealing, and (2) other elements do not occur in significant abundance to influence annealing*, both of which have been definitively refuted (see below). Unfortunately, they refuse to publish complete sample and experimental annealing data, as well as descriptions of model calibration and modelling methods in sufficient detail for other scientists to replicate their results (unlike the 1999 benchmark papers of Carlson, Donelick, and Ketcham, for example). This lack of scientific standards for transparency and reproducibility means that members of the thermochronology community may remain unconvinced of the general value of their Cl-based model.

Fortunately, their method can be evaluated by trying to apply it to sedimentary rocks of variable stratigraphic age from different tectonic settings. We have done this for numerous Phanerozoic sedimentary rocks from various regions in northern and western Canada by acquiring detailed apatite elemental data to constrain AFT interpretations. We find that both $r_{mr0}$ and Cl can define AFT kinetic populations when Cl is the only element in significant abundance or when Cl varies systematically in abundance along with other elements. In both cases, kinetic population ages show a close correspondence to population ages on radial plots that are derived from age mixture modelling. However, for more chemically heterogeneous samples that do not meet these conditions, kinetic populations can be defined using the multi-element $r_{mr0}$ parameter, but they show partial to complete overlap when plotted with respect to Cl. We have numerous samples from different areas of Canada for which a Cl-based interpretation cannot adequately describe our EDM and LA-ICPMS AFT data. Our choices were then either to: (i) abandon the data, (ii) force an unsatisfactory interpretation by assuming the Cl model was right and ignoring important features in the data that were incompatible with it (i.e., Figure 4 of Green and Duddy), or (iii) to try other data interpretation and modelling approaches. We chose the latter path and invested years in developing alternate methods that build on the results of well-documented annealing experiments and modelling approaches. We already have published examples from the Mackenzie Delta (EDM AFT data; Schneider and Issler, 2019) and Mackenzie Corridor (LA-ICPMS AFT data; Powell et al, 2020) of the Canadian Northwest Territories where $r_{mr0}$ has proven to be better suited than Cl. We are preparing to publish results for many more samples once our method paper has been published.

Everyone agrees that Cl is a common constituent of apatite that has a significant influence on thermal annealing. Figure 1 of Green and Duddy shows AFT age varying with Cl content and this pattern makes sense for apatite grains with that particular Cl-dominant composition. Figure 2 of Green and Duddy is presented and discussed in a very misleading way. We don't understand how Green and Duddy infer that Carlson et al. (1999) and Barbarand et al. (2003) "explicitly downplayed" the importance of Cl. We take the opposite view. Their experimental data proved that Cl was insufficient to fully characterize apatite annealing behavior. They tried to accommodate the experimental data by expressing $r_{mr0}$ as a function of Cl—**but they were unable to account for all the annealing data because of apatite grains with significantly different chemistry**. Most of the relatively small number of apatite specimens used in the annealing experiments have variable Cl contents but low abundances of other elements and therefore they conform reasonably well to the predictions of the Cl-based $r_{mr0}$ model. Green and Duddy include these specimens in Figure 2 but they leave out key apatite specimens with high abundances of other elements that lead to different annealing behavior: HS – high OH; KP – high Sr; PC – high Mn, and TI – high Fe, Cl, OH and MIN – high OH, S, Si. Green and Duddy claim other elements show no systematic effects on annealing yet the multi-element-based $r_{mr0}$ parameter best describes the entire data set (Ketcham et al., 1999; Ketcham et al., 2007) compared to single-parameter methods. Although this is a small set of annealing data, these and other published annealing experiments (e.g.,

Ravenhurst et al., 2003; Tello et al., 2006) establish that elements other than Cl can have a significant effect on annealing, providing clear evidence that convincingly refutes assertion 1 above. Green and Duddy seem to be recommending that we ignore the science and focus on using a Cl-only scheme because we can safely omit the "few" occasional outliers that do not conform to their Cl-based model. The problem with this approach is that we do not know the true variation in apatite composition for natural samples and we never will if we don't measure it. Apatite with variable cation and OH concentrations constitute a minor component of the specimens used for the annealing experiments but this should not be construed as being representative of their occurrence in nature. After all, no one believes that the small number of specimens used in the experiments represent the full range of apatite compositions likely to be encountered in real world situations. In fact, our analyses of thousands of naturally occurring apatite grains in our projects supports that apatite compositions do vary considerably.

Figure 3 of Green and Duddy shows another example of how Cl influences annealing. If Cl is the only element of significance for the samples in the Flaxmans-1 well then you will get the kind of population resolution that we observe using the $eCl/r_{mr0}$ parameter. These data are similar to the Cl-dominant populations that comprise some of the specimens used in other annealing experiments (e.g., Carlson et al., 1999; Ketcham et al., 1999; Barbarand et al., 2003). Under these conditions, it is fine to use borehole and experimental annealing data to calibrate a kinetic model that applies to apatite with a Cl-dominant composition. However, it is not reasonable to assume that the composition of apatite in the Flaxmans-1 well applies to all natural apatite samples. Below is a published counterexample to the Flaxmans-1 well. The figures are from Schneider and Issler (2019) and they show EDM AFT data and results from core samples collected in a Mackenzie Delta well with temperature and thermal maturity data. This well-constrained example shows that the multi-elemental-based $eCl/r_{mr0}$ parameter defines two kinetic populations that cannot be resolved using Cl alone. The kinetic population pooled ages closely match (within one standard deviation) the populations ages on the radial plot that are derived from age mixture modelling. Two anomalously young low retentivity grains (dark blue solid triangles) have non-stoichiometric excess F values and no associated track lengths. Such young ages are not present in other samples from this well that display two kinetic populations.

[Figure]

The samples are from the Paleocene-Eocene Aklak and Taglu sequences which have remarkably uniform properties across the basin. The lower retentivity population (red symbols) is similar to other fluorapatites in nature with an eCl value of ~0.05 apfu. The more retentive older population (purple symbols) has high and variable amounts of Fe, Na, Mg, Ce, OH and Cl and it has an eCl of ~0.20 apfu. Six core samples were collected from this well and they show a consistent decrease in kinetic-population ages with increasing temperature and thermal maturity down the borehole.

The less annealed, higher retentivity population shows more variability in population age that is likely related to variability in provenance.

[Figure]

Independent AFTINV thermal models were run for each sample, yielding a coherent set of thermal histories (exponential mean histories are shown in the figure) that are compatible with geological constraints and the known very rapid burial rates in this deltaic succession. This published example provides additional evidence that assertion 2 of Green and Duddy is not supported and that elements beyond Cl occur in natural samples and they influence AFT annealing.

Green and Duddy contend that compositional outliers are rare and they use their Figure 4 to support their claim. This claim needs to be scrutinized in the context of how their analyses are undertaken. Their example (and other published examples by them) show that they measure approximately 25 age grains which is typical of many EDM AFT applications. This number of age grains may be adequate for single age populations but our experience shows that this is generally inadequate to properly characterize single-grain age distributions for the Phanerozoic multikinetic AFT samples that we have examined. Our recommendation is that 40 age grains are needed but, based on processing separate sample aliquots, even more may be necessary if three or more populations are present. It is not a surprise to us that they observe "rare" outliers if ages are undersampled. Also, as discussed in the reply to reviewer 2, unconscious bias related to EDM AFT age measurement may be a contributing factor to the apparent low frequency of "outliers." This can neither be proven nor disproven if populations are undersampled and the true variation in age is unknown. We know that Cl is a common and important element that influences annealing so, if we believe the Cl-based model to be correct, it should be possible to select a subset of grains that "confirm" this. Green and Duddy seemingly only measure wt % Cl and therefore as a result do not have the information to properly assess apatite composition and ensure that it conforms to their assumptions. We believe that the lack of elemental data and undersampling of age grains creates the potential for confirmation bias where the model is assumed to be correct, so there is no need for further investigation. Therefore, alternative methods involving newer technologies that yield results that are inconsistent with the Cl-only model are viewed with suspicion. This conviction seems to be well summarized in the statement by Green and Duddy, "*We find it amusing to contrast the negative attitude to compositional analysis evident in Ketcham's review with the willingness of many thermochronology labs to invest in expensive new machines to measure uranium for fission track studies by laser ablation, and to measure U-Th/He ages, when these methods are far from proven and have been shown to provide misleading results in many cases.*" We believe that science advances when new technologies are adopted and that challenges to conventional ways of thinking are a normal part of scientific endeavors.

References cited:

Barbarand, J., Carter, A., Wood, I. and Hurford, T.: Compositional and structural control of fission-track annealing in apatite, Chem. Geol., 198(1–2), 107–137, 2003.

Carlson, W. D., Donelick, R. A. and Ketcham, R. A.: Variability of apatite fission-track annealing kinetics: I. Experimental results, Am. Mineral., 84(9), 1213–1223, 1999.

Ketcham, R. A., Donelick, R. A. and Carlson, W. D.: Variability of apatite fission-track annealing kinetics; III, Extrapolation to geological time scales, Am. Mineral., 84(9), 1235–1255, doi:https://doi.org/10.2138/am-1999-0903, 1999.

Ketcham, R. A., Carter, A., Donelick, R. A., Barbarand, J. and Hurford, A. J.: Improved modeling of fission-track annealing in apatite, Am. Mineral., 92(5–6), 799–810, doi:10.2138/am.2007.2281, 2007.

Powell, J. W., Issler, D. R., Schneider, D. A., Fallas, K. M. and Stockli, D. F.: Thermal history of the Mackenzie Plain, Northwest Territories, Canada: Insights from low-temperature thermochronology of the Devonian Imperial Formation, Geol. Soc. Am. Bull., 132(3–4), 767–783, 2020.

Ravenhurst, C. E., Roden-Tice, M. K. and Miller, D. S.: Thermal annealing of fission tracks in fluorapatite, chlorapatite, manganoanapatite, and Durango apatite: Experimental results, Can. J. Earth Sci., 40(7), 995–1007, doi:10.1139/e03-032, 2003.

Schneider, D. A. and Issler, D. R.: Application of Low-Temperature Thermochronology to Hydrocarbon Exploration, in Fission-Track Thermochronology and its Application to Geology, edited by M. G. Malusà and P. Fitzgerald, pp. 315–333, Springer International Publishing, Cham., 2019.

Tello, C. A., Palissari, R., Hadler, J. C., lunes, P. J., Guedes, S., Curvo, E. A. and Paulo, S. R.: Annealing experiments on induced fission tracks in apatite: Measurements of horizontal-confined track lengths and track densities in basal sections and randomly oriented grains, Am. Mineral., 91(2–3), 252–260, 2006.

---

## Author Comment (AC4)

**Author response to community comment CC3 for preprint gchron-2021-22**

Issler, D. R., McDannell, K. T., O'Sullivan, P. B., and Lane, L. S.: Simulating sedimentary burial cycles – Part 2: Elemental-based multikinetic apatite fission-track interpretation and modelling techniques illustrated using examples from northern Yukon, Geochronology Discuss. [preprint], https://doi.org/10.5194/gchron-2021-22, in review, 2021.

Duddy and Green take an extreme view that alleged serious data quality issues "irrevocably" compromise the previously demonstrated (Schneider and Issler, 2019; Powell et al., 2020) conclusions that rmt0 is a more general parameter for characterizing the AFT annealing kinetic behaviour than wt% Cl. They conclude that the thermal history modelling therefore has no basis. They utilize select plots and impose arbitrary "eyeball" trends on the data to try and lead readers to the conclusion that both the AFT and elemental data sets are rife with errors while they ignore strong trends that link independently determined AFT age populations (from mixture modelling) with kinetic populations (determined using  $r_{mr0}$  values derived from elemental data). In an attempt to further cement their views, they suggest that it is not even appropriate to use the age mixture modelling the way we have done and therefore it is meaningless that AFT age populations just happen to coincide with kinetic populations. There is no scientific justification for this view. However, it conflicts with their Cl-based method—an approach that rests on assumptions that are not supported by published AFT annealing experiments and some field studies (see reply to their community comment CC1). Their willingness to infer trends in the data that do not exist touches on the subject of unconscious bias and the cognitive predisposition for pattern recognition with regard to the application EDM versus LA-ICPMS AFT methods (see reply to reviewer 2 comment RC2 and community comment CC1). They place a lot of restrictions on how they believe this, and future research should be undertaken and, conveniently, their recommendations conform to the way they have done it. There seems to be no room for new technological advances (LA-ICPMS) or a different way of looking at kinetic populations other than their Cl-based continuum model. The problem with their weak arguments is that they are based on opinions and a superficial examination of the data that do not hold up under closer scrutiny. We prefer not to follow their advice because it leads to confirmation bias where you only collect the minimum data needed to "confirm" a model rather than to test it. Duddy and Green either fail to recognize, or they ignore that multiple parameters are used to support our interpretation, unlike their single parameter Cl method. Below we respond to their various criticisms.

**EPMA data.**

Duddy and Green state categorically that only elemental data with totals between 98-101 wt% are acceptable and that data outside of this range are unfit for the purpose we are using them for. They show simple plots of wt% total versus (Ca site):(P site) and rmr0/eCl. This comparison is irrelevant because they fail to investigate the data deeply enough to determine whether their assertion-that the data provide no useful information-is correct. They suggest that we violate step 7 of our own workflow by not restricting ourselves to using only ideal data. We do not agree with their assessment of "obviously poor data" but it is clear that we need to elaborate on what we mean by this term. All the data and detailed model results are in our GSC Open File report (Issler et al., 2021; 161 p.) which is available for download and is included as an asset for the paper. All elemental totals

---

## Author Comment (AC5)

**Author response to community comment CC5 for preprint gchron-2021-22**

Issler, D. R., McDannell, K. T., O'Sullivan, P. B., and Lane, L. S.: Simulating sedimentary burial cycles – Part 2: Elemental-based multikinetic apatite fission-track interpretation and modelling techniques illustrated using examples from northern Yukon, Geochronology Discuss. [preprint], https://doi.org/10.5194/gchron-2021-22, in review, 2021.

Duddy and Green have taken a scattershot approach that attacks all aspects of our study. They conclude that both the elemental data and AFT data are unreliable but they do not seriously comment on why there is a strong correlation between independently determined AFT age populations and elemental composition for two different samples. Instead, they dismiss the correlation as something to be ignored. They interpret our reply (comment CC4) to their comment as a personal attack although it was not intended to be. Their early contributions to the field of AFT thermochronology are important and have had a strong influence on the research that we are doing today. However, their comment CC3 presents an unbalanced and biased view of our work that is based on inferring trends in our data that do not exist and therefore a firm response that questions their methods and logic is warranted (see our second reply AC4 to their comment CC3). Duddy and Green claim, "*Our initial comment was not written in defense of our methods*" and "*what we do is not at issue here*" yet they conveniently take the opportunity to promote and discuss their poorly documented model in two separate comments. Why is this even relevant if their focus is on "errors" in our study? All of the elements required for an objective scientific debate on competing methods (full disclosure of data including poor analyses and interpretation methods on both sides) are absent in this public discussion. The points we make are very relevant because the association between our "bad" data and their apparently better constrained model is an integral part of their argument.

The statement of Duddy and Green, "*That the EMPA data is too poor in quality to justify any conclusions regarding the utility, or otherwise, of $r_{mr0}$ as a superior kinetic measure to either chlorine alone, or Dpar, is incontrovertible on the basis of our comment (CC3)*" is a provocative claim, given that they invoke nonexistent trends in our data to discredit our study (comment AC4). Some significantly low elemental totals are included in the analysis, but they are not critical to the overall interpretation. They provide results that are consistent with the better quality EPMA data, AFT population ages, and the high Dpar values beyond the range of kinetic population overlap. The suboptimal EPMA data were retained as qualitative information that are consistent with kinetic population assignments. We use a multi-parameter approach that provides some cross-checks on data interpretations. This is not possible to do when relying on a single parameter such as Cl and ignoring independent information from age mixture modelling. Importantly, our method does not rely on the absolute accuracy of eCl/$r_{mr0}$ values for discriminating different kinetic populations. This is in recognition of the fact that natural samples will contain apatite with compositions that have not been encountered in the lab. All we require is the ability to identify different kinetic populations so we can apply the concept of relative annealing to model all the data based on principles that are already being used (i.e., the same annealing mechanism applies to all apatites but kinetic parameters vary with apatite composition).

---

## Author Comment (AC6)

**Author response to reviewer comment RC3 for preprint gchron-2021-22**

Issler, D. R., McDannell, K. T., O'Sullivan, P. B., and Lane, L. S.: Simulating sedimentary burial cycles – Part 2: Elemental-based multikinetic apatite fission-track interpretation and modelling techniques illustrated using examples from northern Yukon, Geochronology Discuss. [preprint], https://doi.org/10.5194/gchron-2021-22, in review, 2021.

Reviewer 1 (Rich Ketcham) has made useful suggestions on trying to assess the relative effects of OH, Cl and cations on annealing kinetics. We have already dealt with the effect of Cl alone in our response (comment AC4) to the comment CC3 of Duddy and Green but include some results here for reference. It is useful to examine the importance of OH on eCl/ $r_{mr0}$  values because, in the early stages of our study, we collected LA-ICPMS elemental data and had problems resolving kinetic populations. F could not be measured accurately using LA-ICPMS and therefore OH could not be calculated. The effect of neglecting OH is to decrease eCl values (increase  $r_{mr0}$ ) in ways that lead to population overlap. This is illustrated in the figures 1 and 2 below where we set OH values equal to zero and recalculated eCl/ $r_{mr0}$ values to simulate the effect of using LA-ICPMS elemental data.

Figure 1 shows various plots of AFT ages and lengths versus eCl (A and B), eCl excluding OH (C and D), and Cl alone (E and F) for the Devonian LHA003 AFT sample. AFT single grain ages are well resolved into two age populations using eCl values derived from  $r_{mr0}$  values calculated using multi-elemental data with the Carlson et al. (1999)  $r_{mr0}$  equation. The age populations closely match those derived from independent age mixture modelling as shown in preprint gchron-2021-22. Figure 1C and 1D show the same plots but exclude OH from the  $r_{mr0}$  calculations. This is analogous to the results we have for samples with LA-ICPMS elemental data that lack the F measurements required for estimating OH values. Both kinetic populations show a shift to lower eCl values (compare Figure 1C with 1A and 1D with 1B) but the change is larger for kinetic population two which has grains with the highest OH content (see table in author reply AC4), causing some grains to overlap with population one. The average eCl values for kinetic populations one and two are 0.03 apfu and 0.20 apfu, respectively, for the complete elemental data (Figure 1A and 1B), and 0.01 apfu and 0.15 apfu for eCl values excluding OH (Figure 1C and 1D). Thus, population two shifts to lower eCl values by 0.05 apfu on average although individual values can decrease by > 0.1 apfu. The effect is smaller for population one (~0.02 apfu decrease on average) with some eCl values becoming negative. For comparison, populations one and two overlap when the AFT data are plotted with respect to Cl alone (Figures 1E and 1F).

OH is clearly important but so are other elements, especially Fe, which has a larger affect than other cations on calculated rmr0 values. Fe is common in many Phanerozoic AFT sedimentary samples that we have examined from northern and western Canada. The LHA003 sample has quite a few grains with Fe contents in the range of 0.04–0.07 apfu and this can increase eCl values significantly as observed for apatite grains with high Fe and low Cl values. This is illustrated in the table below which shows elemental data (apfu) for apatite grains with good elemental totals. Cation abundances  $\geq -0.04$  apfu are highlighted in blue, Fe values  $\leq -0.02$  apfu and Cl values < 0.1 apfu are highlighted in yellow, and OH values > 0.5 apfu are highlighted in blue. Both Cl and eCl values  $\ge 0.1$  apfu are highlighted in green. It is difficult to isolate the contribution of each element to calculated eCl values, but it is clear from equation 6 of Carlson et al. (1999) that Fe has approximately double the effect of other cations. Fe concentrations in the range of 0.4 to 0.6 apfu increase eCl relative to Cl in apatite grains with relatively low Cl and OH contents (age grains 2, 19; length grains 5, 35, 36, 39 and 51). High OH also increases eCl values (age grains 30, 31, 39; length grain 34, 40). The effects are modulated by the addition of other elements (age grains 33, 35, 37; length grains 5, 6, 45 and 51). Age grain 32 has low Cl and OH, and high Na and Mn which increase eCl by 0.07 relative to Cl (with a small contribution from Fe as well). These results suggest that Fe has the strongest effect on track retentivity at a given apfu value with very significant contributions from OH due to its high abundance in many grains. Figure 2A and 2B show a significant reduction in eCl values for LHA003 when Fe is excluded from the rmr0 calculations. Average eCl decreases by ~0.05 apfu and >0.1 apfu for populations one and two, respectively, resulting in population overlap. The reduction in eCl values is less if Mn and other cations are excluded from the  $r_{mr0}$  calculations but Fe is retained (Figure 2C and 2D) – average eCl values decrease by ~0.02 and ~0.03-0.04 apfu for populations one and two. Even though the effect is smaller, neglecting other cations can still contribute to population overlap.

---

## Author Response (AR1)

**Changes in response to AE recommendations**

AE comments are in black font and specific author changes to the manuscript are in red font. Please also read the detailed explanation of changes in the final replies to the referees that accompany this document. Additional revisions were made in response to internal reviewer, Jeremy Powell.

One of the referee comments from Rich Ketcham suggests that you discuss the additional cost of elemental analysis in multi-kinetic AFT workflow. Your response contains a detailed, useful quantification of costs and EPMA analysis times, factors that influence their variability, and an assessment of whether these costs are justified by the data produced. I believe those who read only your article will share the same interest, even if it's only a snapshot of your costs at the present time. Please add a sentence or three on this topic to your revised manuscript.

See our final response to Ketcham's comments that describe the changes we made to the revised manuscript (next section below). We added three sentences that summarize EPMA analysis times, cost reduction strategies, and the relative additional cost of adding EPMA to the workflow.

Likewise, Ketcham's questions about the logistical considerations of first performing laser ablation vs. compositional analysis elicits four or five pages of explanation. Condensing and summarizing these remarks into a paragraph of discussion would benefit those considering whether and how to adopt or adapt your methods. I had the same inclination as Ketcham, around line 340, to add an inline reference to potential downsides of the procedure in section 2.3.

See our final response to Ketcham's comments that outline the changes we made to the manuscript. We have made significant additions to the text in section 2.1 to deal with these concerns.

First, in the second paragraph (line 103, p. 4) under "2. Multikinetic AFT methodology", we give an overall rationale for our chosen methods.

"*The order of steps is based on (1) efficiency and speed of analysis, (2) maximizing the number of track lengths, (3) minimizing selection bias for age grains, and (4) obtaining replicate elemental data. The method can be modified to optimize for other factors or to deal with particular sample conditions but this may increase the cost or the time for analysis.*"

The following text was also added in section 2.1 starting at line 190.

"*Results indicate that compositional zoning is not a common problem for the samples we have studied which is important because the grains are not probed at the exact point of age measurement (laser ablation precedes EMPA). This could contribute to the occasional compositional outlier in kinetic parameter space if elemental zoning is present but kinetic populations are still better resolved compared to when conventional parameters $D_{par}$ or Cl are used (see below). Changing the order of steps so that EPMA is done before laser ablation may help alleviate this problem and reduce the number of AFT grains without elemental data but it could delay sample analysis time by up to several weeks because samples must be transferred back and forth between labs and schedules need to be coordinated. Our current method is efficient and works well for the majority of our samples but it can be modified as needed to deal with more problematic samples. For example, if compositional zoning is a significant issue, it might be better to do laser ablation after EPMA and obtain track lengths from age grains only.*"

We added an inline reference to section 2.1 on line 337 in section 2.3 and eliminated the following two sentences:

"Although elemental measurements can be made with high precision, they are the last step after the grains have been subject to multiple treatments (e.g., etching, laser ablation, Cf-irradiation and etching) and it may not be possible to find a clean spot for EPMA that is close to where the age was measured. Therefore, if compositional zoning is present, the grain may appear as an $r_{mr0}$ outlier."

For Figure 2, reducing the marker size to reduce point saturation should suffice, and I also prefer not to see the correlation coefficient. Like Ketcham, I did not notice the "N =" in the plot titles and would suggest adding it to the statistical summary inset text, but I leave the final decision to the authors.

See our final response to Ketcham's comments. Points were reduced in size and "N=" was added to each panel in Figure 2.

The explanation of the implementation of the CRS method on pages 12-13 of the authors' response to Ketcham looks fine. I see broad parallels ramping alpha and simulated annealing algorithms, and similarities between its smoothing and other inverse problem approaches. As you mention on your related answer on page 14, though, please do make clear where you believe your AFT data do not constrain the thermal history.

See our final response to Ketcham's comments that outline the changes we made to the manuscript. We added some text and dashed lines to Figures 6 and 7 that indicate the approximate upper temperature limit that can be constrained by the AFT data.

Finally, a revised manuscript would benefit from a brief summary of the more detailed discussion around the AFTINV and QTQt differences on pages 16-18 of the responses to Ketcham's review. My feeling is that (4) plays the largest role here. The mechanics of the AFTINV and QTQt fits are discussed elsewhere in the manuscript, but a separate paragraph in the discussion would draw these threads together.

See our final response to Ketcham's comments that outline the changes we made to the manuscript. We added more text and a third paragraph in section 4.2 on QTQt to elaborate on factors that contribute to the differences in model results.

To address Karl Lang's initial comment, please seek maximum clarity about "detrital" vs. "sedimentary" nomenclature at the beginning of the paper. The use of "detrital sample" throughout the paper is ok with me.

See our final response to Lang's comment that outline the changes we made. We use "sedimentary" in place of "detrital" in several places near the beginning and retained "detrital" where we believe the meaning is clear.

As a larger point, I do share some of Lang's concern that this manuscript includes a "vigorous preference of LA ICPMS over [the] EDM approach." I understand that this is a current community-wide discussion and has been for some time now. This manuscript makes a meaningful contribution to this discussion in that it demonstrates the success of a workflow that includes LA-ICPMS in the AFT workflow. As such, I believe the paragraphs from lines 114 to 146 represent an out-of-place, discursive discussion in the Methods portion of the paper, and they should be omitted. These thoughts would be better included, I believe, in a separate manuscript that has space and scope for a more complete discussion of the "different methods with different advantages and disadvantages" described in the response to Lang. The advantages of the LA-ICPMS method in your workflow are well documented elsewhere throughout the paper. There is also plenty of additional discussion, outlined above, to add to a revised manuscript that is more germane to your novel method and its application to examples from the northern Yukon.

We have eliminated much of the text from the first two paragraphs in section 2.1 and replaced it with a single paragraph (see final response to Lang's comment). We agree that the subject is best left to a more focused paper in the future that deals with both LA-ICP-MS and EDM data. We have quality EDM data with clearly defined multikinetic populations and we believe that both methods can yield good results. As suggested by Karl Lang, if there are difference in results, it is more likely to be related to how users implement the methods rather than to intrinsic differences in the methods.

Finally, this paper received a series of contributed comments and replies from Ian Duddy and Paul Green, with a short clarifying response also from Rich Ketcham. The broad-ranging discussion includes questions about the LA-ICPMS method in general, use of the $r_{mr0}$ metric, and the quality of the EPMA analyses in this manuscript in particular. I appreciate the discussion and replies, which will remain available as part of the format of the GChron journal. I agree with Ketcham's overall assessment (RC3) of the EPMA data in that "I don't think perfect is necessary for this to be a useful contribution." I acknowledge the analytical difficulties in measuring compositions in heterogeneous apatite populations, and I find the discussion in AC6 of the low elemental totals, the effect of including vs. excluding OH in elemental totals to bring them up, and the effect of Fe on track retentivity interesting. A revised manuscript should include a sentence or two that summarize your reasoning that your conclusions are not adversely affected by low totals.

See our final response to Ketcham's comments that outline the changes we made to the manuscript. We added the following text to the beginning of the paragraph on lines 187-195 (p. 8) at the end of section 2.1:

"*We try to obtain EPMA measurements on a smooth "clean" surface but this is not always possible for small grains with many etched tracks and other imperfections. Missing elements and track void space in the electron beam excitation volume can result in elemental totals that are less than the 97-100 wt% expected for good analyses (see Issler et al. (2021) data tables) and this can happen for samples with AFT data acquired by LA-ICP-MS or EDM. Fortunately, we have observed that suboptimal analyses still yield an elemental signature that allows for discrimination of different kinetic populations and replicate elemental analyses with good and lower elemental totals yield similar results for the study samples.*"

**Review 1 (Richard Ketcham) Comments**

Reviewer comments are in black font and specific author changes to the manuscript are in red font.

At the same time, to be effective in doing so (or at least transparent in trying), it would be good to better document the costs.

We discussed this in detail in our reply (AC1) but we don't think that absolute costs should be mentioned in the paper because they depend on different factors such as the laboratory doing the work and the number of samples being processed at a given time among other things. However, we included information on relative costs for additional EPMA work (see point immediately below).

For example, how long does the EMPA protocol take per spot?

See detailed response (AC1). We added the following three sentences (in italics) (line 185) to provide some information on analysis time and relative cost with and without EPMA. "*The raw analysis time for our samples (Peter Hooper Geoanalytical Lab, WSU) is approximately 3.7 minutes per spot (16 grains per hour), excluding the time for setup (1-2 hours), standardization (8 hours) and point picking (150*

*grains per hour). Hourly billing is capped at 12 hours per day so it is advantageous to run samples in large batches for 48-72 hours straight which can reduce the hourly rate by 30-40 %. EPMA increases the average cost of AFT analysis by approximately 20% for our samples."*

What considerations went into decision(s) of whether to do compositional analysis first versus laser ablation? Does doing laser ablation first save time, by figuring out which grains work and providing evidence of whether there is kinetic dispersion, and does this outweigh the disadvantage of not getting the analysis precisely where the tracks were measured?

See detailed response (AC1). We added the following two sentences (in italics) after line 103 (below Table 1) to add some context without going into too many distracting details. "*The order of steps is based on (1) efficiency and speed of analysis, (2) maximizing the number of track lengths, (3) minimizing selection bias for age grains, and (4) obtaining replicate elemental data. The method can be modified to optimize for other factors or to deal with particular sample conditions but this may increase the cost or the time for analysis.*" We thought it was also important to modify another sentence in this paragraph (line 107 and 108) to read as, "The goal here is to try to use all the available data except for obviously poor analyses *that provide no useful information (i.e., inaccurate U measurements).*" This is in response to the community comment that we address in substantial detail (AC4). We strongly disagree with view of Duddy and Green that elemental data with suboptimal totals provide no useful information. Our rejection of obviously poor analyses are those where U measurement is clearly inaccurate as detailed in our comment AC4 (grains blown out during the initial stage of laser ablation and U values below ICP-MS resolution).

Are there cases where changing the order would be a good idea?

See detailed response (AC1). We are hoping the author replies to comments are an online resource for readers who want more detailed information.

We added the following text starting at line 190 to address this point.

"*Results indicate that compositional zoning is not a common problem for the samples we have studied which is important because the grains are not probed at the exact point of age measurement (laser ablation precedes EMPA). This could contribute to the occasional compositional outlier in kinetic parameter space if elemental zoning is present but kinetic populations are still better resolved compared to when conventional parameters $D_{par}$ or Cl are used (see below). Changing the order of steps so that EPMA is done before laser ablation may help alleviate this problem and reduce the number of AFT grains without elemental data but it could delay sample analysis time by up to several weeks because samples must be transferred back and forth between labs and schedules need to be coordinated. Our current method is efficient and works well for the majority of our samples but it can be modified as needed to deal with more problematic samples. For example, if compositional zoning is a significant issue, it might be better to do laser ablation after EPMA and obtain track lengths from age grains only.*"

[line 44] Justifying the 20°C bound seems to require citing Donelick et al (1990), and optionally Tamer and Ketcham (2020).

These references have been inserted into the paper.

[line 243] Change to "thermal history modeling" (or "all model calculations").

The first recommended change has been made to the text.

[line 249] Although replicate values are indeed important for assessing the reproducibility of kinetic parameter values, they may also be taken as an indication of the presence of zoning. The authors do not specify how many spots they took per analysis, but I suspect the answer is one, and that it reflects the usual 2-µm activation zone for EMP; was this driven by the desire for a faster and/or less expensive analysis?

Efficient and cost-effective analysis is an important factor. We believe that the ability to define kinetic populations is related to the good reproducibility of eCl values from single spot analyses for samples with replicate measurements (shown in our Fig. 2) and this implies zoning is not a pervasive problem. We mention that single spot elemental analyses are used in line 256 in the paragraph immediately below. To further clarify this point, we have modified the sentence on lines 248-250 (new insertion in italic) to be "Replicate elemental *(a single EPMA spot per AFT analysis)* and Dpar analyses from separate measurements on grains with both age and length data (step 2, Fig. 1) are very important for assessing the reproducibility of kinetic parameter values (Fig. 2)."

Likewise, how many Dpar measurements are averaged for each Dpar determination? The usual procedure is to average four, which ought to make the reproducibility better than observed in Fig. 2c.

The LHA003 sample was analysed at AtoZ Inc. following protocols established by Ray Donelick and these procedures continued to be followed at GeoSep Services. The majority of $D_{par}$ measurements were derived by averaging four $D_{par}$ measurements per grain analysis. We modified text (line 153, p. 7; new insertion in italic) in section 2.1 to read, "…$D_{par}$ is measured for individual apatite age grains *(average of four $D_{par}$ measurements where possible)* …." Details such as this are also included in the separate GSC Open File 8821 that contains the sample AFT and elemental data.

Also, it's a little unfortunate that the discussion of the downsides of this procedure (lines ~326-340; might not get a compositional analysis near the counting area, or for the grain at all, I gather partly due to the LAICPMS spot) is in the next section; the authors can probably clarify and condense things by briefly mentioning these here, and then referring to them in section 2.3.

We decided to address this point in section 2.1 (all changes in italics). We also added comments concerning elemental wt% totals because the issue was raised in the community comments. We made additions and modifications to the text in several paragraphs to clarify the order of steps for acquiring AFT data. Sentences were restructured and reordered in the third paragraph of section 2.1 (lines 151-165, p. 7) to make it clear that all track count and length measurements are done before laser ablation to obtain U concentration. The first two sentences duplicated information already presented and were eliminated. The relevant changes to the paragraph are:

"Following standard mineral separation, and *grinding, polishing and etching of apatite crystals to expose spontaneous tracks, grain mounts are typically $^{252}Cf$ –irradiated to increase the number of confined tracks for length measurement (this may not be necessary for samples with high track densities such as Precambrian samples). Then,* spontaneous tracks are counted, $D_{par}$ is measured for individual apatite age grains *(average of four $D_{par}$ measurements where possible),* and grain x–y coordinates are recorded so that subsequent measurements can be linked to the age grains *(Step 1, Figure 1). The sample is re-etched to reveal horizontal confined tracks and their lengths, angles with respect to the mineral c-axis, and $D_{par}$ are measured and x–y coordinates are recorded for the measured grains. Finally,* the sample is analysed using LA-ICP-MS to obtain U, Th, Sm, U–Pb age, *and trace element (as an option)* data for the AFT age grains, ensuring that the laser spot coincides with the track count area to minimize any potential problems with inhomogeneous U distributions. *Jepson et al. (2021) discuss how U-Pb age, trace element, and AFT*

*data can be used to enhance thermal history interpretations.* As an *additional* option, *U-Pb age*, U, Th, Sm, *and other trace element* data can be acquired for the length grains as well."

We changed line 169-171 to read as, "*Currently*, we recommend that elemental data be acquired using electron probe microanalysis (EPMA) rather than by LA-ICP-MS even if the latter method *allows for elemental data to be acquired at the point of age measurement and it is* more convenient to integrate in the workflow." Also the paragraph starting on line 167 was split into two paragraphs to accommodate requested changes by GSC internal reviewer, Jeremy Powell (see document outlining changes to the manuscript in response to his comments).

We added the following two sentences to the beginning of the paragraph on line 187 ("EPMA is undertaken…"). "*We try to obtain EPMA measurements on a smooth "clean" surface but this is not always possible for small grains with many etched tracks and other imperfections. Missing elements and track void space in the electron beam excitation volume can result in elemental totals that are less than the 97-100 wt% expected for good analyses (see Issler et al. (2021) data tables) and this can happen for samples with AFT data acquired by LA-ICP-MS or EDM. Fortunately, we have observed that suboptimal analyses still yield an elemental signature that allows for discrimination of different kinetic populations and replicate elemental analyses with good and lower elemental totals yield similar results for the study samples.*"

We added the following sentence after the text on replicate analyses on line 190. "*Results indicate that compositional zoning is not a common problem for the samples we have studied which is important because the grains are not probed at the exact point of age measurement (laser ablation precedes EMPA). This could contribute to the occasional compositional outlier in kinetic parameter space if elemental zoning is present but kinetic populations are still better resolved compared to when conventional parameters $D_{par}$ or Cl are used (see below). Changing the order of steps so that EPMA is done before laser ablation may help alleviate this problem and reduce the number of AFT grains without elemental data but it could delay sample analysis time by up to several weeks because samples must be transferred back and forth between labs and schedules need to be coordinated. Our current method is efficient and works well for the majority of our samples but it can be modified as needed to deal with more problematic samples. For example, if compositional zoning is a significant issue, it might be better to do laser ablation after EPMA and obtain track lengths from age grains only.*"

Finally, we added an inline reference to section 2.1 on line 337 in section 2.3 and eliminated the following two sentences:

"Although elemental measurements can be made with high precision, they are the last step after the grains have been subject to multiple treatments (e.g., etching, laser ablation, Cf-irradiation and etching) and it may not be possible to find a clean spot for EPMA that is close to where the age was measured. Therefore, if compositional zoning is present, the grain may appear as an $r_{mr0}$ outlier."

[line 269] Although Dpar imprecision is certainly responsible for a lot of the scatter in Fig. 2e, it's not clear it's the main reason; the authors might try only plotting the points within the 20% bars in Fig. 2c and seeing what the Dpar vs. eDpar scatter looks like. The even scatter might simply be an indication that the things that throw Dpar off are bidirectional; a little OH might increase resistance to annealing compared to no OH (i.e. F-apatite), and a lot of OH might decrease it (e.g., OH-apatite HS from Carlson et al. (1999), but the more OH you have the higher Dpar is.

We have given a lengthy reply in our initial author response (AC1). However, we don't see how any of this would fit well in the paper without increasing its length. We think it might best serve as an online reference as is.

[Figure 2] Maybe smaller symbols would be better to avoid some of the "solid cloud" effect; some "N =" annotations also would not hurt, and maybe correlation coefficients for d and e.

We have decreased the plot symbol size to try and reduce the point saturation. It is tricky to get around this when you are plotting thousands of points at a reduced scale. Given the very large scatter of the data in Figs. 2c, 2d and 2e, we chose not to use linear regression but instead show how the data are distributed around a 1:1 line. We prefer not to include correlation coefficients because we don't think the relationships would be very meaningful, especially for Fig. 2d where there are strong systematic differences between eCl (a function of many elements) and Cl. Coefficients would change with the addition of more data from apatite of different elemental composition because eCl can vary considerably for a given value of Cl. The plot data are included in the supplement to the paper should someone wish to analyse it in a different way. We removed the number of data points from the title for each plot and added "N=" as inset text for each panel.

[line 291] "colour-coded"

Quotation symbols have been added.

[line 295] It may be worth noting that compositional populations may also be good candidates for shared inheritance. Although eCl is one such possibility, insofar as it combines a number of compositional variables into one number, apatites with similar eCl may get there via different compositional components, and thus not constitute a good candidate for shared inheritance. This is discussed further below.

See comment AC1. We don't think it should be mentioned at this point in the paper for the reasons given. The topic of inheritance is mentioned in the discussion.

[line 333-337] Maybe here or elsewhere, discuss the choice between switching which bin a grain is in, versus leaving the grain out altogether.

See detailed reply in AC1.We added the following two sentences at the end of the paragraph to emphasize the choices we make and the reasons for doing so. "*In our experience, the LA-ICP-MS method produces consistent and reliable AFT ages and therefore we use as much of the data as possible. If a higher precision AFT age matches an existing age population but plots as an outlier in kinetic space, we prefer to reassign it to the matching population rather than omitting the grain.*"

[line 438] The claim that population 3 has retained tracks from 540 Ma, or from about 245°C (Figure 6) is eye-catching, and probably overly optimistic about the ability of AFT to retain information about such high temperatures. It appears to stem from a difference in how AFTINV evaluates total annealing versus HeFTy's "oldest track". HeFTy assumes total annealing after reduced mean length falls below 0.4095 for non-projected lengths, corresponding to a mean length of just under 7 µm, whereas AFTINV appears to have total annealing correspond to a mean track length of 2 µm (line 419). This may be based on a slight misinterpretation of what's written in Ketcham et al. (2000); the 2 µm limit mentioned there corresponds to the smallest track that can appear in a track length distribution. However, such occurrences are due to including a population of tracks with a higher mean and large standard deviation. The 0.4095 value arises in part from the observation that no annealing experiments reported by Green (1988) or Carlson et al. (1999) had a mean length below 7 µm (although there are some 6's and 5's reported by Barbarand et al.

(2003), and even an occasional 4 or 3 by their Analyst 3). Willett (1997) uses a similar value of 0.428 as the zero-density intercept of reduced length versus density reported by Green (1988). In other words, by the time a mean length falls below some limit, the track population becomes undetectable. I believe this provides a more realistic basis for evaluating total annealing and the oldest retained track. Using the revised criterion, the TA for the oldest track for an rmr0=0.491 apatite is closer to 200°C, which seems a lot more reasonable considering the closure temperature is 161°C. This is not the most crucial of issues, but it's prudent to avoid distracting claims.

See detailed reply AC1. Short tracks exist but are rarely observed, in part because they are less likely to be intersected by an etchant pathway than longer tracks. The issue of model retention ages was also raised by internal reviewer, Jeremy Powell, and we made changes in response to his comments. We changed the range of annealing temperatures from ~100-245°C to ~110-185°C (Abstract, line 16 and Introduction, line 84) based on the ability of the model to resolve temperatures rather than on predictions using retention ages for very short tracks. For figures 6 and 7, we added a dashed line and arrow to show the model temperature range that is unconstrained by the AFT data and we removed the reference to total annealing temperatures for the ~2 μm tracks used to calculate retention ages. The figure captions have been updated accordingly. We also removed reference to total annealing temperatures linked to retention ages in the text (lines 429-430 and 454-455).

[Figure 6, 7] I appreciate the authors' efforts to incorporate the CRS method into AFTINV, and intrigued by the result – it looks to be a powerful addition. I have long been considering doing something similar myself, having dropped the CRS method when I converted my earlier program AFTSolve to HeFTy. However, one of the reasons I did so may still be evident in the model results here. The CRS method has a tendency to quickly converge to a relatively smooth solution that does not explore the solution space as well as the Monte Carlo method, and thus map out the range of solutions that fit well. In HeFTy results, this allows the resolving power of the data to be evaluated by looking at the width of the solution envelopes.

We think our detailed response (AC1) is a useful online reference but we don't think this material would fit into the paper.

In the results here, what puzzles me for P013-12 is the relatively tight band of good solutions above 175°C from 600-450 Ma, and probably a fair bit younger/cooler than that. Given the 161°C closure temperature of the most resistant population, the idea that it would exert much constraint in the 175-250°C temperature range seems improbable, and is not reflected in the QTQt results either. This all is not necessarily a problem, but I think it should be discussed so people interpreting these results have a more complete knowledge of what they are looking at.

See detailed response AC1. We updated figures 6 and 7 to indicate the parts of the thermal history that are resolved by the AFT data and we added some explanatory text. We added the sentence, "*The dashed line (upper panel, Fig. 6) coincides with a change to a steady cooling rate (~1.2 °C/My) below 185 °C at ~440 Ma and marks the upper temperature limit that can be resolved from modelling the AFT data*" on line 437 for the P013-12 sample. On line 460 (end of paragraph), we add the sentence, "*Model results suggest that the AFT data can resolve the thermal history below 175 °C after ~480 Ma (dashed line in upper panel, Fig. 7)*" for the LHA003 sample.

Along similar lines, did both the AFTINV and QTQt models assume that all apatites in each sample had the same inherited, pre-depositional history? If so, was the fact that they did so, and their success in fitting

their models, and indication that there was shared provenance, or an indication that, for these samples, results are not terribly sensitive to the pre-depositional history?

See detailed response AC1. We added the following paragraph after line 460 at the end of section 4.1 AFTINV.

"*The P013-12 and LHA003 samples were modelled assuming a common inherited, pre-depositional history for each kinetic population in order to better resolve the post-depositional thermal history by taking advantage of relative annealing. We cannot determine whether the kinetic populations within each sample have a shared inheritance because this information has been degraded by thermal annealing of the less track-retentive populations. For LHA003, the pre-depositional thermal record for the lower retentivity population one was erased completely by thermal annealing and therefore pre-depositional cooling is only constrained by population two. For P013-12, pre-depositional thermal history has been erased for population one and population two experienced significant post-depositional annealing. Pre-depositional cooling is dominated by population three which can easily overlap with any residual cooling record for population two. Overall, the post-depositional thermal histories are not very sensitive to the pre-depositional cooling for these samples*."

Or, are the results sensitive – do the few earlier-cooling 0.5 paths for P013-12 corresponds to the earlier peaks T's at ~195 Ma and/or ~70 Ma?

We answered this specific question in our reply (AC1) but we don't see how to directly address this in the paper without it looking like we were answering a reviewer question. See response immediately above.

The manual (AFTINV) and automatic (QTQt) raising of the rmr0 values for the most resistant populations in each sample is interesting. What seems to be going on is that the different populations need greater separation in their partial annealing zones to produce their respective divergent age and length distributions. It's further interesting that the higher resistance is corroborated by the vitrinite data for sample LHA003, though less so for P013-12. The authors recommended approach of "anchoring" on low-resistance kinetic seems like a good one. Another possible "advantage" of the Ketcham et al. (1999) model over the (2007) one beyond the different rmr0 equation is that it has a much higher temperature range, which these results may imply is necessary to create these divergent populations.

Our reply (AC1) adds some information that may be useful as a reference but doesn't seem to be necessary to include in the paper.

Lastly, the comparison between AFTINV and QTQt results appears to gloss over their differences a bit. For P013-12, the first reheating peaks at ~168 Ma in AFTINV and could go as far back as 195 Ma, whereas QTQt appears to strongly say that it was at about 140 Ma. Similarly, AFTINV implies that the first peak reheating for LHA003 was at 345 Ma, compared to 300 Ma for QTQt. If you lay the models pairs on top of each other, they appear to exclude each other at these times. Is this because QTQt calculated different kinetics than the manually-shifted ones in AFTINV, or because of QTQt favoring simpler histories, or some combination of these and possibly other factors?

We give a very detailed response (AC1). We added more text to the second paragraph of section 4.2 on QTQt to discuss the differences in model results and the probable factors responsible for this.

We inserted the following sentence after the second sentence in the second paragraph of section 4.2 QTQt (line 480). "*These temperatures are at the upper end of the range defined by the CRS solutions in Figure 6 (132–147 °C between 150–195 Ma and 102–110 °C between 45–70 Ma) but the 95% confidence region overlaps with the AFTINV results.*"

The following two sentences were added after the sixth sentence (line 487). "*QTQt predicts a younger time for the first thermal peak than the AFTINV CRS solutions (Fig. 7; ~320–360 Ma) but the temperatures and times for the other peaks show good agreement and both models overlap in the 95% confidence region. AFTINV uses larger eCl values and more closely fits the AFT parameters for the most retentive population in both samples.*"

We started a new paragraph beginning at the last sentence (line 487) and added the following five sentences at the end of the QTQt section to summarize reasons why the QTQT and AFTINV model results differ in some details.

"*Overall, the AFTINV and QTQt results are very similar, even with the subtle trade-offs between the different thermal minima/maxima inflection points and preferred model population kinetic parameters. Model results differ in detail for a number of reasons. Compared with QTQt, AFTINV uses more model points, constructs thermal histories differently, allows for manual fine tuning of kinetic parameters, and generates a much larger set of "acceptable" and "good" solutions. QTQt generally prefers simpler histories and there is a trade-off between the number of time-temperature points and data fit. QTQt converts LA-ICP-MS AFT data to EDM AFT data and uses Ns and Ni count data rather than ages for modelling whereas AFTINV models either EDM or LA-ICP-MS AFT data using central or pooled ages, depending on $\chi^2$ and age dispersion statistics. This difference is most evident for sample P013-12 where QTQt uses younger observed population ages for model input than AFTINV (compare Figs. 6 and 8).*"

**Review 2 (Karl Lang) Comments**

Reviewer comments are in black font and specific author changes to the manuscript are in red font.

Use of "detrital" was a little confusing to me at first, since many applications of detrital thermochronology are now also focused on interpreting cooling histories of source rocks prior to deposition, and not simply the common cooling history of detrital minerals in a sedimentary rock after deposition. This is a semantic difference, but perhaps adding a sentence to state this explicitly at the beginning of the manuscript might clear up any confusion amongst readers.

We changed the first sentence in the Abstract to read "…cause for AFT age dispersion in sedimentary samples" rather than "…cause for age dispersion in detrital AFT samples." We changed "…detrital AFT samples with apatite of variable cation and anion composition to have significant age dispersion…" to "…sedimentary samples with apatite of variable cation and anion composition to have significant AFT age dispersion…" in the first sentence of the Conclusions. We retained detrital elsewhere in the text.

Why does the manuscript include a vigorous preference of LA ICPMS over EDM approach? This seems unrelated to the central motivation of the paper and, in my opinion, is largely unsupported (see comments by line). The authors should explain why they chose to use LA-ICPMS instead of EDM, but they should avoid generalized claims about the relative efficacy of one method over the other (e.g. "The LA-ICP-MS method has some distinct advantages compared with EDM" [117]).

We did not intend to indicate a strong preference for LA-ICP-MS with respect to EDM. We have excellent EDM AFT data that show clearly defined multikinetic AFT populations that can be modelled successfully. We anticipated that there may be critical community comments concerning the LA-ICP-MS AFT method (we surmised correctly in this case) and so we inserted some text to emphasize that the two methods are different in some respects but that these differences may have different advantages. However, if these comments are being viewed as an assault on the EDM method then we appreciate this

being pointed out. On rereading this section, we agree that the text is peripheral to the main point of the paper and, as suggested by the AE, these ideas are better left to a future study that compares both methods. Therefore, we have replaced the first two paragraphs in section 2.1 (lines 114 to 146) with the following paragraph.

"This section discusses the type of data required for multikinetic AFT thermochronology; more details on sample analysis are in Issler et al. (2021). Our AFT data were acquired using the LA-ICP-MS method (Chew and Donelick, 2012; Cogné et al., 2020; Donelick et al., 2005; Hasebe et al., 2004) although the technique works equally well using the older external detector method (EDM; Hurford and Green, 1982). A key difference is that the U data needed for AFT age determination are acquired for the spontaneous track count area after counting is completed (LA-ICP-MS method) whereas counting of spontaneous and induced (proxy measure for U from sample irradiation) tracks are done at the same time (EDM). Typically, 40 single grain AFT ages and 100–200 track lengths are obtained per sample, depending on apatite yield. Generally, this amount of data is sufficient for most multikinetic samples with two or three kinetic populations, but more data may be required for samples with unevenly distributed populations or with more than three populations. In contrast, many EDM AFT studies have used a lower number of age grains per sample (usually ≤ 20) for thermal history studies. The greater number of counted age grains naturally increases the statistical probability of $\chi^2$ failure that may complicate mixture model interpretation (McDannell, 2020; Vermeesch, 2019)."

118. It has not been my experience that analytical costs are lower for LA-ICPMS than for the EDM when measured on a per grain basis. If you can measure 100 grains per mount and 50 mounts fit in a $1000 irradiation package, that's $5/grain. By comparison, LaserChron (probably cheapest option in US, at least) charges $9-16 per grain for 100 grain samples, not including costs for CL imaging. Also, throughput is not necessarily higher for LA if you have to wait several months for lab time to become available. In my experience the analytical time to produce a complete fission-track dataset is comparable regardless of the analytical approach. I worry that comments like this will gradually discourage scientists from using the EDM, which is a well established and data-rich method.

Our reply (AC2) provides some background reference pertaining to cost. We removed "and costs are lower" from line 118. Although costs are lower for us, it is difficult to generalize when different labs and different methods are being used.

137-138. Wouldn't observer bias have a greater impact on age determination when it is only accounted for in spontaneous track counting? It seems to me that observer bias may actually be reduced when it is accounted for in both the spontaneous and induced track counts, rather than just in spontaneous counts. Either way, I don't consider it fair to say that LA-ICPMS is "more objective" if it still relies on user interpretation and collection of spontaneous track data.

We removed the text but we believe that the issue of objectivity and the potential for grain selection bias should not be dismissed categorically. We maintain that LA-ICP-MS is more "objective" in that analysts must count track areas without advance knowledge of U concentration. They may be able get some assessment of potential U zonation from the distribution of spontaneous tracks which can be helpful for avoiding areas with strong zonation but all track counting is done prior to U measurement. With EDM, there is the potential for track counting to be influenced by accumulating Ns/Ni count data, especially if there is a desire to have samples with minimum age dispersion. We know in the real world that outcomes can change when people have additional information and that the "Monty Hall effect" is a real phenomenon. We do not know how significant a factor this is for real measurements but we point out that

the potential is there for it to be a factor that may lead to underrepresentation of multikinetic populations. Nevertheless, this discussion belongs elsewhere because we are not dealing with EDM data in this study.

140. This is not an inherent limitation of the EDM, simply a choice by the operator to count fewer grains. Many detrital studies regularly count more than 100 grains per sample with the EDM.

Yes, but in practical applications of EDM for thermal history analysis, most studies have counted a lower number of grains compared to what we measure using LA-ICP-MS.

142-143. It is convenient to make this argument here, but one could also make an alternative argument that the induced track print actually allows for more robust data collection because you can avoid the zonation issues you mention to be a problem on line 128-130. I don't understand why this is cast as an example of making the EDM less objective.

This point is generally overlooked as a possible factor when comparing differences in LA-ICP-MS and EDM AFT data so we think it is worth mentioning in future work that compares the two methods. Zonation is a possible reason for data discrepancies, but if lab work is done in a careful and consistent manner, we believe that it is not a widespread problem based on the success we have had with analyzing Phanerozoic sedimentary rocks in northern Canada.

145. Again, this is not an inherent problem with EDM it is a choice by the EDM user.

We agree that user choices can influence results and that experienced users who are aware of these issues can mitigate these problems.

**Jeremy Powell (GSC Internal Review) Comments**

Reviewer comments are in black font and our replies are in red font.

Abstract, line 12. Consider "we present an interpretation and modelling strategy that exploits multikinetic AFT annealing kinetics of samples with compositional variability. These multikinetic thermal histories provide more detail and better resolution compared to conventional methods"

We inserted extra text (in italics) in the sentence so it reads as, "We present an interpretation and modelling strategy *for samples with variable apatite composition* that exploits multikinetic AFT annealing to obtain thermal histories that can provide more detail and better resolution compared to conventional methods".

Abstract, line 52, p2. Substitute "regions" for "areas"?

We retained areas because both words are synonymous. Seems to be a matter of personal preference.

Line 84, p. 3. Should specify that these are based on the thermal history models and not independent annealing studies. That's not clear in this sentence.

We changed the total annealing temperature range from ~110 °C to 245 °C to ~110 °C to 185 °C based on the range over which the model can resolve thermal histories. Sentence has been modified to say "The AFT kinetic populations of this study have a wider range of total annealing temperatures (~110 °C to 185 °C *based on model thermal histories*) than typical fluorapatite (~110 °C)…"

Table 1 – should include which NAD is referenced.

NAD83 datum was added to the bottom of Table 1.

Multikinetic AFT methodology section, line 90-100. I agree with not discussing the geologic setting or providing a map in this manuscript, but consider referencing other publications or geologic maps that the readers can look up if interested.

We updated a sentence (new text in italics) to read "Table 1 summarizes basic sample location and stratigraphic information *and a geological map with plotted sample locations is available in Issler et al. (2021)*."

Multikinetic AFT methodology section, line 90-100. Available thermal maturity data could be discussed here as well – both for the samples, and for the overlying Cretaceous section.

We added the sentence, "*Percent vitrinite reflectance (%Ro) data (Issler et al., 2021) indicate that both samples experienced paleotemperatures that were high enough (~135 – 175 °C) to cause substantial AFT annealing*." We also modified a sentence (new text in italics) to say, "Approximately 1 km of Cretaceous strata *(≤ 0.6 %Ro)* overlie the unconformity…"

Figure 1, Step 4: shouldn't the referenced equation be Carlson et al., 1999?

Yes, Ketcham et al. (1999) was changed to Carlson et al. (1999) for step 4 of flowchart.

Figure 1, Step 6: the grey kinetic boundary is hard to see. Might not show up online. Maybe make it darker?

The light grey dashed lined was replaced with a dark grey.

Line 135. Rework this sentence. It's a bit of a mouthful, which makes it hard to understand the point you're making.

We have divided the sentence into two sentences. It now reads as, "*We think that both the LA-ICP-MS and EDM methods can yield equally good results for the age grains that are measured. This has been demonstrated repeatedly in publications where EDM and ICP-MS methods were compared or where ICP-MS AFT dates were referenced to samples with well-determined absolute ages (e.g., Ansberque et al., 2021; Cogné et al., 2020; Hasebe et al., 2004; Iwano et al., 2019; Seiler et al., 2013; Soares et al., 2014)*."

Line 150-151. This section feels a bit redundant given the description in lines 103-112. Is one of these sections the figure caption?

We removed the first two sentences, "Figure 1 illustrates the key steps in our multikinetic workflow. Step 1 summarizes the procedures needed for acquiring AFT and related data using the LA-ICP-MS method" to avoid duplication of the text above. We modified the next sentence by inserting "*(Step 1, Figure 1)*" at the end of the sentence. This should convey the essential information.

Paragraph on lines 150-164. Consider expanding the utility of AFT + UPb by referencing Jepson et al 2020 and their work on double dating.

Modified line 153-155 (new text in italics) to read, "The sample is analysed using LA-ICP-MS to obtain U, Th, Sm, U–Pb age, *and trace element (as an option)* data for the AFT age grains, ensuring that the laser spot coincides with the track count area to minimize any potential problems with inhomogeneous U distributions." Added the following sentence, "*Jepson et al. (2021) discuss how U-Pb age, trace element, and AFT data can be used to enhance thermal history interpretations.*" Modified line 158-159 to read,

"As an option, *U-Pb age,* U, Th, Sm*, and other trace element* data can be acquired for the length grains as well."

Line 170-175. Provide a comment on the challenges of measuring F using EPMA as well. It would be useful if you recommended analytical conditions for measuring apatite composition via EPMA – are you doing two runs and changing the operating conditions for halogens vs majors? Is there a processing software that the lab uses to mitigate beam migration, etc.

We split the paragraph into two paragraphs and added the following two sentences at the end of the first paragraph. *Analytical conditions for elemental analysis are summarized in the elemental data files included with the Issler et al. (2021) sample report. For the sake of efficient sample processing, a single setup was used for elemental analysis and time-dependent corrections were used to deal with halogen migration (similar to Nielsen and Sigurdsson, 1981), with the knowledge that crystals oriented with their c-axis parallel to the electron beam could yield some inaccurate results.*

Line 224, equation 2. I think eCl is a really confusing concept to many readers, so it's good to be crystal clear in your description of it. For example, It might be a bit tough for readers to connect that Cl\* of Ketcham et al 1999 is the eCl of this paper, and that Abs(Cl-1) is the measured Cl component.

We have chosen to use the same equations as given in Ketcham et al. (1999) and all parameters are defined. For those less comfortable with the mathematical aspects, we cited equation 1 of McDannell and Issler (2021) which shows the final transformed equation for eCl. The sentence on lines 229 and 230 now reads as (new text in italics), " Equations (2) and (3) can be used to transform measured kinetic parameters (i.e., Dpar and Cl) to $r_{mr0}$ values or vice versa by rearranging the equations in terms of Cl and Dpar *(see equation (1) of McDannell and Issler (2021) for eCl).*"

Line 239, change "eCl" to "The eCl parameter.

Done.

Line 253. Can't quite tell, but are their two ranges indicated around the 1:1 line? The figure says 5-10% but I only see one set of dashed lines.

No. The dashed lines represents ±0.03 apfu. This value is typically 5-10% of the range of eCl and Cl values present in a sample.

Line 253. Also, consider adding a figure showing replicate analyses of F (maybe in the supporting information document). I think that the EPMA composition could be a sticking point for some reviewers. The description of EPMA data acquisition is limited, and some reviewers might suspect that the Cl vs eCl difference could partially be due to poor F measurements resulting in changes in stoichiometric OH. I know that's almost certainly not the case, but it might help this manuscript to be a bit more descriptive in how you acquire F and how replicate measurements look.

We added F data to table S1 in the supplement and included a plot showing the results of replicate F analyses with the table. We added the following two sentences after the discussion of the replicate results for Cl and eCl. *A similar plot of 322 replicate results for F is included in the supplement with Table S1 and it shows that 92% of the measurements are within ±0.2 apfu with larger variations associated with zoning and nonstoichiometric F values. Overall, the accuracy of the halogen measurements is sufficient for estimating OH contents and calculating $r_{mr0}$ values.*

Line 258. Of major and minor elements – might be worthwhile specifying, as readers could relate this to your discussion of U-Th-Sm zonation and LAICPMS vs EDM above.

We think this may not be necessary as it should be implicit that chemical zoning refers to major and minor elements.

Line 298, Fig 3 doesn't show chi square values for individual population.

These are listed in Table 2 for the kinetic populations. DensityPlotter only shows the $\chi^2$ probability value for the whole sample. The $\chi^2$ numbers are more relevant for the interpreted kinetic populations which are used for thermal modelling.

Also, Fig. 3 caption needs to mention what the values in brackets next to the peak ages are.

Added the following to the caption, "*(estimated percentage of grains per population in brackets)*."

Line 373-374. Maybe mention that dynamic limits are determined iteratively and only applied to make the modelling more efficient? Some might misinterpret this as forcing the model.

Modified sentence to read (new text in italics), "Static limits define the entire model search space whereas dynamic limits are applied only at model inflection points to focus calculations into favourable regions of solution space *to improve model efficiency*."

Line 388. As well as dynamic temperature limits?

We consider determining the style of thermal history to encompass any adjustment to rate/temperature limits.

Line 417. This description of retention ages could be more clear. The retention age concept is something that many people seem to struggle to wrap their head around, so consider adding an extra sentence or two describing it.

Replaced the existing sentence with the following two sentences. *Model retention ages represent a theoretical age for the oldest (shortest) track in each population (assumed to be ~ 2 μm based on the shortest track ever measured; Ketcham et al., 2000) and provide an uppermost temperature and time limit for track survival. However, very short tracks are rarely observed and maximum temperatures constrained by the AFT data may be significantly lower.*

Line 531. 'may yield significantly more thermal history information'

Changed "can contain" to "may yield."

Line 557. This is an important paragraph of the discussion, as it develops an argument as to why $r_{mr0}$ is a valid kinetic parameter despite the warning of Ketcham et al 1999. I think that the last sentence loses the point a little, and could be reworded to better emphasize these points. In my opinion, the important stuff here isn't that you're 'advancing the field of thermochronology', it's that you're revisiting existing experimental data in a novel way to improve how we apply thermochronology methods to complex geological problems. Maybe that's pedantic, but some might read that sentence and think "advancing the field of thermochronology would be fully understanding these complex annealing behaviours that aren't predicted by Ketcham/Carlson."

Changed the last sentence of the paragraph to, "*Our method pursues the logical consequences of the annealing experiments and shows that it is possible to use existing techniques in a novel way to improve how we apply thermochronology methods to complex geological problems.*"

Line 566. In between these sentences might be a good place to mention that the <0.73 $r_{mr0}$ range also corresponds with where we have no constraints in the annealing datasets, so the discrepancy b/w measured $r_{mr0}$ values and those required by the modelling is expected.

Added the sentence, "*There is a dearth of experimental annealing data for $r_{mr0}$ values < 0.73 so the discrepancy between $r_{mr0}$ values calculated using elemental data and those required for modelling is expected.*"

Line 598. Just a thought as I review – but did you mention that you used Basin%Ro to model vitrinite? It would be good to reference that model and why you use it instead of EasyRo (or reference our Mackenzie Plain paper), given that this paper is emphasizing how we can extract complex thermal histories from sedimentary rocks with multiple burial cycles.

Yes, the basin%Ro model is cited in the section, "3. Thermal history modelling of multikinetic AFT data." I modified the second last sentence of the second paragraph (lines 379-380) to be (new text in italics), "Vitrinite reflectance (%Ro) values are calculated for the entire post-depositional thermal history and for the last phase of heating and cooling using the basin%Ro model (Nielsen et al., 2017) *which provides better fits to observed maturity profiles in northern Canada than the Sweeney and Burnham (1990) EASY%Ro model (e.g., Issler et al., 2016; Powell et al., 2020).*"

Line 625. Another point is that multikinetic samples may produce similar thermal histories when modelled, but have very different age populations. It would be a useful thought experiment (maybe a future paper?) to test what increasing or decreasing the temperature of one thermal event does to the pooled age and track length distribution of these populations – presumably, lower Cretaceous temperatures = more effect of pre-Cretaceous thermal history on the youngest population and a partially annealed age. Whereas, higher Cretaceous temperatures might lower the age of the Mesozoic and Paleozoic population. Judging by the samples that we've worked on, I suspect that it doesn't take much of an increase/decrease in temperature to produce noticeable effects. I mention this, because as people begin to work with multi-kinetic samples they will inevitably be frustrated/sceptical when proximal samples yield different age populations. Many will skim over your point on "value of the approach judged on the ability to generate spatially coherent thermal histories", which is a very important conclusion of your multikinetic work.

We added the following two sentences to the last paragraph of the Discussion section. "*Furthermore, multikinetic samples with different age populations may produce similar thermal histories, depending on how differences in provenance and composition interact with the thermal history. Therefore, proximal samples may not necessarily have the same age populations.*"